# On Computing Probabilistic Explanations for Decision Trees

**Marcelo Arenas**[1,2,4]**, Pablo Barceló**[2,4,5]**, Miguel Romero**[3,5]**, Bernardo Subercaseaux**[6]

Authors are listed in alphabetical order.
Correspondence at `bsuberca@cs.cmu.edu`.
[1] Department of Computer Science, PUC Chile
[2] Institute for Mathematical and Computational Engineering, PUC Chile
[3] Faculty of Engineering and Science, UAI Chile
[4] Millenium Institute for Foundational Research on Data, Chile
[5] CENIA Chile
[6] Carnegie Mellon University, Pittsburgh, USA

## Abstract

Formal XAI (explainable AI) is a growing area that focuses on computing explanations with mathematical guarantees for the decisions made by ML models. Inside formal XAI, one of the most studied cases is that of explaining the choices taken by decision trees, as they are traditionally deemed as one of the most interpretable classes of models. Recent work has focused on studying the computation of *sufficient reasons*, a kind of explanation in which given a decision tree $T$ and an instance $\boldsymbol{x}$, one explains the decision $T(\boldsymbol{x})$ by providing a subset $\boldsymbol{y}$ of the features of $\boldsymbol{x}$ such that for any other instance $\boldsymbol{z}$ compatible with $\boldsymbol{y}$, it holds that $T(\boldsymbol{z}) = T(\boldsymbol{x})$, intuitively meaning that the features in $\boldsymbol{y}$ are already enough to fully justify the classification of $\boldsymbol{x}$ by $T$. It has been argued, however, that sufficient reasons constitute a restrictive notion of explanation. For such a reason, the community has started to study their probabilistic counterpart, in which one requires that the probability of $T(\boldsymbol{z}) = T(\boldsymbol{x})$ must be at least some value $\delta \in (0, 1]$, where $\boldsymbol{z}$ is a random instance that is compatible with $\boldsymbol{y}$. Our paper settles the computational complexity of $\delta$-sufficient-reasons over decision trees, showing that both (1) finding $\delta$-sufficient-reasons that are minimal in size, and (2) finding $\delta$-sufficient-reasons that are minimal inclusion-wise, do not admit polynomial-time algorithms (unless PTIME = NP). This is in stark contrast with the deterministic case ($\delta = 1$) where inclusion-wise minimal sufficient-reasons are easy to compute. By doing this, we answer two open problems originally raised by Izza et al., and extend the hardness of explanations for Boolean circuits presented by Wäldchen et al. to the more restricted case of decision trees. On the positive side, we identify structural restrictions of decision trees that make the problem tractable, and show how SAT solvers might be able to tackle these problems in practical settings.

## 1 Introduction

**Context.** The trust that AI models generate in people has been repetitively linked to our ability of *explaining* the decision of said models (Arrieta et al., 2020), thus suggesting the area of *explainable AI* (XAI) as fundamental for the deployment of trustworthy models. A sub-area of explainability that has received considerable attention over the last years, showing quick progress in theoretical and practical terms, is that of *local* explanations, i.e., explanations for the outcome of a particular input to an ML model after the model has been trained. Several queries and scores have been proposed to

specify explanations of this kind. These include, e.g., queries based on *prime implicants* (Shih et al., 2018) or *anchors* (Ribeiro et al., 2018), which are parts of an instance that are sufficient to explain its classification, as well as scores that intend to quantify the impact of a single feature in the output of such a classification (Lundberg and Lee, 2017; Yan and Procaccia, 2021).

A remarkable achievement of this area of research has been the development of *formal* notions of explainability. The benefits brought about by this principled approach have been highlighted, in a very thorough and convincing way, in a recent survey by Marques-Silva and Ignatiev (2022). A prime example of this kind of approach is given by *sufficient reasons*, which are also known as prime implicant explanations (Shih et al., 2018) or abductive explanations (Ignatiev et al., 2021).

Given an ML model $\mathcal{M}$ of dimension $n$ and a Boolean input instance $\boldsymbol{x} \in \{0,1\}^n$, a sufficient reason for $\boldsymbol{x}$ under $\mathcal{M}$ is a subset $\boldsymbol{y}$ of the features of $\boldsymbol{x}$, such that any instance $\boldsymbol{z}$ compatible with $\boldsymbol{y}$ receives the same classification result as $\boldsymbol{x}$ on $\mathcal{M}$. In more intuitive words, $\boldsymbol{y}$ is a sufficient reason for $\boldsymbol{x}$ under $\mathcal{M}$ if the features in $\boldsymbol{y}$ suffices to explain the output of $\mathcal{M}$ on $\boldsymbol{x}$. In the formal explainability approach, one then aims to find sufficient reasons $\boldsymbol{y}$ that satisfy one of the following optimality criteria: (a) they are *minimum*, i.e., there are no sufficient reasons with fewer features than $\boldsymbol{y}$, or (b) they are *minimal*, i.e., there are no sufficient reasons that are strictly contained in $\boldsymbol{y}$.

**Problem.** The XAI community has studied for which Boolean ML models the problem of computing (minimum or minimal) sufficient reasons is computationally tractable and for which it is computationally hard (see, e.g., (Barceló et al., 2020; Marques-Silva et al., 2020, 2021)). It has been argued, however, that for practical applications sufficient reasons might be too *rigid*, as they are specified under worst-case conditions. That is, $\boldsymbol{y}$ is a sufficient reason for $\boldsymbol{x}$ under $\mathcal{M}$ if *every* "completion" of $\boldsymbol{y}$ is classified by $\mathcal{M}$ in the same way as $\boldsymbol{x}$. As several authors have noted already, there is a natural way in which this notion can be relaxed in order to become more suitable for real-world explainability tasks: Instead of asking for each completion of $\boldsymbol{y}$ to yield the same result as $\boldsymbol{x}$ on $\mathcal{M}$, we could allow for a small fragment of the completions of $\boldsymbol{y}$ to be classified differently than $\boldsymbol{x}$ (Izza et al., 2021; Wäldchen et al., 2021; Wang et al., 2021). More precisely, we would like to ensure that a random completion of $\boldsymbol{y}$ is classified as $\boldsymbol{x}$ with probability at least $\delta \in (0,1]$, a threshold that the recipient of the explanation controls. We call $\boldsymbol{y}$ a $\delta$-*sufficient reason for* e *under* $\mathcal{M}$.

The study of the cost of computing minimum $\delta$-sufficient reasons for expressive Boolean ML models based on propositional formulas was started by Wäldchen et al. (2021). They show, in particular, that the decision problem of checking if $\boldsymbol{x}$ admits a $\delta$-sufficient reason of a certain size $k$ under a model $\mathcal{M}$, where $\mathcal{M}$ is specified as a CNF formula, is $\mathrm{NP^{PP}}$-complete. This result shows that the problem is very difficult for complex models, at least in theoretical terms. Nonetheless, it leaves the door open for obtaining tractability results over simpler Boolean models, starting from those which are often deemed to be "easy to interpret", e.g., *decision trees* (Gilpin et al., 2018; Izza et al., 2020a; Lipton, 2018). In particular, the study of the cost of computing both minimum and minimal $\delta$-sufficient reasons for decision trees was initiated by Izza et al. (2021, 2022), but nothing beyond the fact that the problem lies in NP has been obtained. Work by Blanc et al. (2021) has shown that it is possible to obtain efficient algorithms that succeed with a certain probability, and that instead of finding a smallest (either cardinality or inclusion-wise) $\delta$-sufficient reason, find $\delta$-sufficient reasons that are small compared to the *average* size of $\delta$-sufficient reasons for the considered model.

**Our results.** In this paper we provide an in-depth study of the complexity of the problem of minimum and minimal $\delta$-sufficient reasons for decision trees.

1. We start by pinpointing the exact computational complexity of these problems by showing that, under the assumption that PTIME $\neq$ NP, none of them can be solved in polynomial time. We start with minimum $\delta$-sufficient reasons and show that the problem is hard even if $\delta$ is an arbitrary fixed constant in $(0,1]$. Our proof takes as basis the fact that, assuming PTIME $\neq$ NP, the problem of computing minimum sufficient reasons for decision trees is not tractable (Barceló et al., 2020). The reduction, however, is non-trivial and requires several involved constructions and a careful analysis. The proof for minimal $\delta$-sufficient reasons is even more difficult, and the result more surprising, as in this case we cannot start from a similar problem over decision trees: computing minimal sufficient reasons over decision trees (or, equivalently, minimal $\delta$-sufficient reasons for $\delta = 1$) admits a simple polynomial time algorithm. Our result then implies that such a good behavior is lost when the input parameter $\delta$ is allowed to be smaller than 1.

2. To deal with the high computational complexity of the problems, we look for structural restrictions of it that, at the same time, represent meaningful practical instances and ensure that these problems can be solved in polynomial time. The first restriction, called *bounded split number*, assumes there is a constant $c \geq 1$ such that, for each node $u$ of a decision tree $T$ of dimension $n$, the number of features that are mentioned in both $T_u$, the subtree of $T$ rooted at $u$, and $T - T_u$, the subtree of $T$ obtained by removing $T_u$, is at most $c$. We show that the problems of computing minimum and minimal $\delta$-sufficient reasons over decision trees with bounded split number can be solved in polynomial time. The second restriction is *monotonicity*. Intuitively, a Boolean ML model $\mathcal{M}$ is *monotone*, if the class of instances that are classified positively by $\mathcal{M}$ is closed under the operation of replacing 0s with 1s. For example, if $\mathcal{M}$ is of dimension 3 and it classifies the input $(1, 0, 0)$ as positive, then so it does for all the instances in $\{(1, 0, 1), (1, 1, 0), (1, 1, 1)\}$. We show that computing minimal $\delta$-sufficient reasons for monotone decision trees is a tractable problem. (This good behavior extends to any class of monotone ML models for which counting the number of positive instances is tractable; e.g., monotone *free binary decision diagrams* (Wegener, 2004)).

3. In spite of the intractability results we obtain in the paper, we show experimentally that our problems can be solved over practical instances by using SAT solvers. This requires finding efficient encodings of such problems as conjunctive normal form (CNF) formulas, which we then check for satisfiability. This is particularly non-trivial for probabilistic sufficient reasons, as it requires dealing with the arithmetical nature of the probabilities involved through a Boolean encoding.

**Organization of the paper.** We introduce the main terminology used in the paper in Section 2, and we define the problems of computing minimum and minimal $\delta$-sufficient reasons in Section 3. The intractability of the these problems is proved in Section 4, while some tractable restrictions of them are provided in Section 5. Our Boolean encodings can be found in Section 6. Finally, we discuss some future work in Section 7.

## 2 Background

An *instance* of dimension $n$, with $n \geq 1$, is a tuple $\boldsymbol{x} \in \{0, 1\}^n$. We use notation $\boldsymbol{x}[i]$ to refer to the $i$-th component of this tuple, or equivalently, its $i$-th feature. Moreover, we consider an abstract notion of a model of dimension $n$, and we define it as a Boolean function $\mathcal{M} : \{0, 1\}^n \rightarrow \{0, 1\}$. That is, $\mathcal{M}$ assigns a Boolean value to each instance of dimension $n$, so that we focus on binary classifiers with Boolean input features. Restricting inputs and outputs to be Boolean makes our setting cleaner while still covering several relevant practical scenarios.

A *partial instance* of dimension $n$ is a tuple $\boldsymbol{y} \in \{0, 1, \bot\}^n$. Intuitively, if $\boldsymbol{y}[i] = \bot$, then the value of the $i$-th feature is undefined. Notice that an instance is a particular case of a partial instance where all features are assigned value either 0 or 1. Given two partial instances $\boldsymbol{x}, \boldsymbol{y}$ of dimension $n$, we say that $\boldsymbol{y}$ is *subsumed* by $\boldsymbol{x}$, denoted by $\boldsymbol{y} \subseteq \boldsymbol{x}$, if for every $i \in \{1, \ldots, n\}$ such that $\boldsymbol{y}[i] \neq \bot$, it holds that $\boldsymbol{y}[i] = \boldsymbol{x}[i]$. That is, $\boldsymbol{y}$ is subsumed by $\boldsymbol{x}$ if it is possible to obtain $\boldsymbol{x}$ from $\boldsymbol{y}$ by replacing some undefined values. Moreover, we say that $\boldsymbol{y}$ is *properly subsumed* by $\boldsymbol{x}$, denoted by $\boldsymbol{y} \subsetneq \boldsymbol{x}$, if $\boldsymbol{y} \subseteq \boldsymbol{x}$ and $\boldsymbol{y} \neq \boldsymbol{x}$. Notice that a partial instance $\boldsymbol{y}$ can be thought of as a compact representation of the set of instances $\boldsymbol{z}$ such that $\boldsymbol{y}$ is subsumed by $\boldsymbol{z}$, where such instances $\boldsymbol{z}$ are called the *completions* of $\boldsymbol{y}$ and are grouped in the set $\text{COMP}(\boldsymbol{y})$.

A *binary decision diagram* (BDD) of dimension $n$ is a rooted directed acyclic graph $\mathcal{M}$ with labels on edges and nodes such that: (i) each leaf (a node with no outgoing edges) is labeled with **true** or **false**; (ii) each internal node (a node that is not a leaf) is labeled with a feature $i \in \{1, \ldots, n\}$; and (iii) each internal node has two outgoing edges, one labeled 1 and the other one labeled 0. Every instance $\boldsymbol{x} \in \{0, 1\}^n$ defines a unique path $\pi_{\boldsymbol{x}} = u_1 \cdots u_k$ from the root $u_1$ to a leaf $u_k$ of $\mathcal{M}$ such that: if the label of $u_i$ is $j \in \{1, \ldots, n\}$, where $i \in \{1, \ldots, k-1\}$, then the edge from $u_i$ to $u_{i+1}$ is labeled with $\boldsymbol{x}[j]$. Moreover, the instance $\boldsymbol{x}$ is positive, denoted by $\mathcal{M}(\boldsymbol{x}) = 1$, if the label of $u_k$ is **true**; otherwise the instance $\boldsymbol{x}$ is negative, which is denoted by $\mathcal{M}(\boldsymbol{x}) = 0$. A BDD $\mathcal{M}$ is *free* if for every path from the root to a leaf, no two nodes on that path have the same label. A *decision tree* is simply a free BDD whose underlying directed acyclic graph is a rooted tree.

# 3 Probabilistic Sufficient Reasons

Sufficient reasons are partial instances obtained by removing from an instance $x$ components that do not affect the final classification. Formally, fix a dimension $n$. Given a decision tree $T$, an instance $x$, and a partial instance $y$ with $y \subseteq x$, we call $y$ a *sufficient reason* for $x$ under $T$ if $T(x) = T(z)$ for every $z \in \text{COMP}(y)$. In other words, the features of $y$ that take value either $0$ or $1$ explain the decision taken by $T$ on $x$, as $T(x)$ would not change if the remaining features (i.e., those that are undefined in $y$) were to change in $x$, thus implying that the classification $T(x)$ is a consequence of the features defined in $y$. We say that a sufficient reason $y$ for $x$ under $T$ is *minimal*, if it is minimal under the order induced by $\subseteq$, that is, if there is no sufficient reason $y'$ for $x$ under $T$ such that $y' \subsetneq y$. Also, we define a *minimum* sufficient reason for $x$ under $T$ as a sufficient reason $y$ for $x$ under $T$ that maximizes the value $|y|_\perp := |\{i \in \{1, \ldots, n\} \mid y[i] = \perp\}|$.

It turns out that minimal sufficient reasons can be computed efficiently for decision trees with a very simple algorithm, assuming a sub-routine to check whether a given partial instance is a sufficient reason (not necessarily minimal) of another given instance. As shown in Algorithm 1, the idea of the algorithm is as follows: start with a candidate answer $y$ which is initially equal to $x$, the instance to explain, and maintain the invariant that $y$ is a sufficient reason for $x$, while trying to remove defined components from $y$ until no longer possible. It is not hard to see that one can check whether a partial instance $y$ is a sufficient reason for an instance $x$ in linear time over decision trees (Audemard et al., 2021). This algorithm is well known (see e.g., (Huang et al., 2021)), and relies on the following simple observation, tracing back to Goldsmith et al. (2005).

**Observation 1.** *For any class of models $\mathfrak{C}$, if a partial instance $y$ of dimension $n$ is a sufficient reason for an instance $x$ under a model $\mathcal{M} \in \mathfrak{C}$, but not a minimal sufficient reason, then there is a partial instance $\hat{y}$ which is equal to $y$ except that $\hat{y}[i] = \perp, y[i] \neq \perp$ for some $i \in \{1, \ldots, n\}$ which is also a sufficient reason for $x$ under $\mathcal{M}$.*

---

**Algorithm 1:** Minimal Sufficient Reason

**Input:** Decision tree $T$ and instance $x$, both of dimension $n$
**Output:** A minimal sufficient reason $y$ for $x$ under $T$.

1  $y \leftarrow x$
2  **while** *true* **do**
3      reduced $\leftarrow$ **false**
4      **for** $i \in \{1, \ldots, n\}$ **do**
5          $\hat{y} \leftarrow y$
6          $\hat{y}[i] \leftarrow \perp$
7          **if** *CheckSufficientReason*$(T, \hat{y}, x)$ **then**
8              $y \leftarrow \hat{y}$
9              reduced $\leftarrow$ **true**
10             **break**
11     **if** *($\neg$reduced) or* $|y|_\perp = n$ **then**
12         **return** $y$

---

The following theorem shows a stark contrast between the complexity of computing minimal and minimum sufficient reasons over decision trees.

**Theorem 1** (Barceló et al. (2020))**.** *Assuming* PTIME $\neq$ NP*, there is no polynomial-time algorithm that, given a decision tree $T$ and an instance $x$ of the same dimension, computes a minimum sufficient reason for $x$ under $T$.*

Arguably, the notion of sufficient reason is a natural notion of explanation for the result of a classifier. However, such a concept imposes a severe restriction by asking all completions of a partial instance to be classified in the same way. To overcome this limitation, a probabilistic generalization of sufficient reasons was proposed by Wäldchen et al. (2021) and Izza et al. (2021). More precisely, this notion allows to settle a confidence $\delta \in (0, 1]$ on the fraction of completions of a partial instance that yield the same classification.

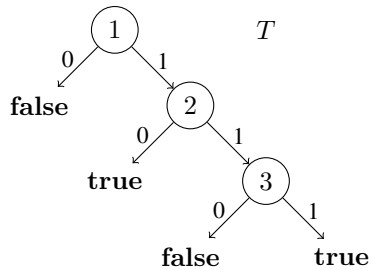

$$p(\emptyset) = \tfrac{3}{8} \qquad p(\{1,2\}) = \tfrac{1}{2}$$
$$p(\{1\}) = \tfrac{3}{4} \qquad p(\{1,3\}) = 1$$
$$p(\{2\}) = \tfrac{1}{4} \qquad p(\{2,3\}) = \tfrac{1}{2}$$
$$p(\{3\}) = \tfrac{1}{2} \qquad p(\{1,2,3\}) = 1$$

Figure 1: The decision tree $T$ and the values $p(X)$ from Example 1.

**Definition 1** (Probabilistic sufficient reasons). *Given a value $\delta \in (0,1]$, a $\delta$-sufficient reason ($\delta$-SR for short) for an instance $\boldsymbol{x}$ under a decision tree $T$ is a partial instance $\boldsymbol{y}$ such that $\boldsymbol{y} \subseteq \boldsymbol{x}$ and*

$$\Pr_{\boldsymbol{z}}[T(\boldsymbol{z}) = T(\boldsymbol{x}) \mid \boldsymbol{z} \in \text{COMP}(\boldsymbol{y})] := \frac{|\{\boldsymbol{z} \in \text{COMP}(\boldsymbol{y}) \mid T(\boldsymbol{z}) = T(\boldsymbol{x})\}|}{2^{|\boldsymbol{y}|_\perp}} \geq \delta.$$

Minimal and minimum $\delta$-sufficient reasons are defined analogously as in the case of minimal and minimum sufficient reasons.

**Example 1.** Consider the decision tree $T$ over features $\{1,2,3\}$ shown in Figure 1 and the input instance $\boldsymbol{x} = (1,1,1)$. Notice that $T(\boldsymbol{x}) = 1$. For each $X \subseteq \{1,2,3\}$, we show the probability $p(X) := \Pr_{\boldsymbol{z}}[T(\boldsymbol{z}) = 1 \mid \boldsymbol{z} \in \text{COMP}(\boldsymbol{y}_X)]$, where $\boldsymbol{y}_X$ is the partial instance that is obtained from $\boldsymbol{x}$ by fixing $\boldsymbol{y}_X[i] = \perp$ for each $i \notin X$. We observe, for instance, that $\boldsymbol{x}$ itself is neither a minimum nor a minimal 1-SR for $\boldsymbol{x}$ under $T$, as $\boldsymbol{y}_{\{1,3\}} = (1, \perp, 1)$ is also a 1-SR. In turn, $\boldsymbol{y}_{\{1,3\}}$ is both a minimal and a minimum 1-SR for $\boldsymbol{x}$ under $T$. The partial instance $\boldsymbol{y}_{\{1,3\}}$ is not, however, a minimal or a minimum $^3/_4$-SR for $\boldsymbol{x}$ under $T$, as $\boldsymbol{y}_{\{1\}} = (1, \perp, \perp)$ is also a $^3/_4$-SR. $\qquad\square$

**Example 2.** Consider the decision tree $T$ over features $\{1,2,3\}$ shown in Figure 2 and the input instance $\boldsymbol{x} = (1,1,1)$. Notice that $T(\boldsymbol{x}) = 1$. Exactly as in Example 1, we display as well the probabilities $p(X)$ for each $X \subseteq \{1,2,3\}$. Interestingly, this example illustrates that Observation 1 does not hold when $\delta < 1$. Indeed, consider that $\boldsymbol{y}_{\{1,2,3\}}$ is a $^5/_8$-SR which is not minimal, as $\boldsymbol{y}_\emptyset$ is also a $^5/_8$-SR, but if we remove any single feature from $\boldsymbol{y}_{\{1,2,3\}}$, we obtain a partial instance which is not a $^5/_8$-SR. $\qquad\square$

As illustrated on Example 2, it is not true in general that if $\boldsymbol{y}' \subset \boldsymbol{y}$ then

$$\Pr_{\boldsymbol{z}}[T(\boldsymbol{z}) = T(\boldsymbol{x}) \mid \boldsymbol{z} \in \text{COMP}(\boldsymbol{y}')] \leq \Pr_{\boldsymbol{z}}[T(\boldsymbol{z}) = T(\boldsymbol{x}) \mid \boldsymbol{z} \in \text{COMP}(\boldsymbol{y})],$$

which means that standard algorithms for finding minimal sets holding monotone predicates (see e.g., (Marques-Silva et al., 2013)) cannot be used to compute minimal $\delta$-SRs.

The problems of computing minimum and minimal $\delta$-SR on decision trees were defined and left open by (Izza et al., 2021, 2022). These problems are formally defined as follows.

| PROBLEM: : | Compute-Minimum-SR |
|---|---|
| INPUT : | A decision tree $T$ of dimension $n$, an instance $\boldsymbol{x}$ of dimension $n$ and $\delta \in (0,1]$ |
| OUTPUT : | A minimum $\delta$-SR for $\boldsymbol{x}$ under $T$ |

| PROBLEM: : | Compute-Minimal-SR |
|---|---|
| INPUT : | A decision tree $T$ of dimension $n$, an instance $\boldsymbol{x}$ of dimension $n$ and $\delta \in (0,1]$ |
| OUTPUT : | A minimal $\delta$-SR for $\boldsymbol{x}$ under $T$ |

## 4 The Complexity of Probabilistic Sufficient Reasons on Decision Trees

In what follows, we show that neither Compute-Minimum-SR nor Compute-Minimal-SR can be solved in polynomial time (unless PTIME = NP). We first consider the problem

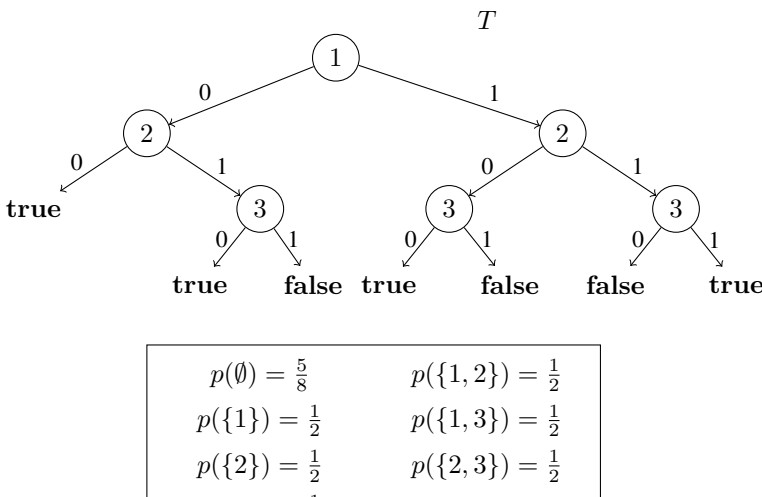

Figure 2: The decision tree $T$ and the values $p(X)$ from Example 2.

Compute-Minimum-SR, and in fact prove a stronger result by considering the family of problems $\delta$-Compute-Minimum-SR where $\delta \in (0, 1]$ is assumed to be fixed. More precisely, we obtain as a corollary of Theorem 1 that 1-Compute-Minimum-SR cannot be solved efficiently. Moreover, a non-trivial modification of the proof of this theorem shows that this negative result continues to hold for every fixed $\delta \in (0, 1]$.

**Theorem 2.** *Fix* $\delta \in (0, 1]$. *Then assuming that* PTIME $\neq$ NP*, there is no polynomial-time algorithm for* $\delta$-Compute-Minimum-SR.

Let us now look at the problem Compute-Minimal-SR. When $\delta = 1$, this problem can be solved in polynomial time as stated in Theorem 1. However, it is conjectured by Izza et al. (2021) that assuming PTIME $\neq$ NP, this positive behavior does not extend to the general problem Compute-Minimal-SR, in which $\delta$ is an input confidence parameter. Our main result confirms that this conjecture is correct.

**Theorem 3.** *Assuming that* PTIME $\neq$ NP*, there is no polynomial-time algorithm for* Compute-Minimal-SR.

*Proof sketch.* We first show that the following decision problem, called Check-Sub-SR, is NP-hard. This problem takes as input a tuple $(T, \boldsymbol{x})$, for $T$ a decision tree of dimension $n$, and $\boldsymbol{x} \in \{0, 1, \bot\}^n$ a partial instance, and the goal is to decide whether there is a partial instance $\boldsymbol{y} \subsetneq \boldsymbol{x}$ with $\mathrm{Pr}_{\boldsymbol{z}}[T(\boldsymbol{z}) = 1 \mid \boldsymbol{z} \in \mathrm{COMP}(\boldsymbol{y})] \geq \mathrm{Pr}_{\boldsymbol{z}}[T(\boldsymbol{z}) = 1 \mid \boldsymbol{z} \in \mathrm{COMP}(\boldsymbol{x})]$. We then show that if Compute-Minimal-SR admits a polynomial time algorithm, then Check-Sub-SR is in PTIME, which contradicts the assumption that PTIME $\neq$ NP. The latter reduction requires an involved construction exploiting certain properties of the hard instances for Check-Sub-SR.

To show that Check-Sub-SR is NP-hard, we use a polynomial time reduction from a decision problem over formulas in CNF, called Minimal-Expected-Clauses, and which we also show to be NP-hard. Both the NP-hardness of Minimal-Expected-Clauses and the reduction from Minimal-Expected-Clauses to Check-Sub-SR may be of independent interest.

We now define the problem Minimal-Expected-Clauses. Let $\varphi$ be a CNF formula over variables $X = \{x_1, \ldots, x_n\}$. Partial assignments of the variables in $X$, as well as the notions of subsumption and completions over them, are defined in exactly the same way as for partial instances over features. For a partial assignment $\mu$ over $X$, we denote by $E(\varphi, \mu)$ the expected number of clauses of $\varphi$ satisfied by a random completion of $\mu$. We then consider the following problem for fixed $k \geq 2$ (recall that a $k$-CNF formula is a CNF formula where each clause has at most $k$ literals):

| | |
|---|---|
| PROBLEM : | $k$-Minimal-Expected-Clauses |
| INPUT : | $(\varphi, \sigma)$, for $\varphi$ a $k$-CNF formula and $\sigma$ a partial assignment |
| OUTPUT : | Yes, if there is a partial assignment $\mu \subsetneq \sigma$ such that $E(\varphi, \mu) \geq E(\varphi, \sigma)$ and No otherwise |

We show that $k$-Minimal-Expected-Clauses is NP-hard even for $k = 2$, via a reduction from the well-known clique problem. Finally, the reduction from $k$-Minimal-Expected-Clauses to Check-Sub-SR builds an instance $(T, \boldsymbol{x})$ from $(\varphi, \sigma)$ in a way that there is a direct correspondence between partial assignments $\mu \subseteq \sigma$ and partial instances $\boldsymbol{y} \subseteq \boldsymbol{x}$, satisfying that

$$\Pr_{\boldsymbol{x}}[T(\boldsymbol{z}) = 1 \mid \boldsymbol{z} \in \text{COMP}(\boldsymbol{y})] = \frac{E(\varphi, \mu)}{m},$$

where $m$ is the number of clauses of $\varphi$. This implies that $(\varphi, \sigma)$ is a Yes-instance for $k$-Minimal-Expected-Clauses if and only if $(T, \boldsymbol{x})$ is a Yes-instance for Check-Sub-SR. $\square$

## 5 Tractable Cases

We now study restrictions on decision trees that lead to polynomial time algorithms for Compute-Minimum-SR or Compute-Minimal-SR, hence avoiding the general intractability results shown in the previous section. We identify two such restrictions: *bounded split number* and *monotonicity*.

### 5.1 Bounded split number

Let $T$ be a decision tree of dimension $n$. For a set $U$ of nodes of $T$, we denote by $\mathcal{F}(U)$ the set of features from $\{1, \ldots, n\}$ labeling the nodes in $U$. For each node $u$ of $T$, we denote by $N_u^{\downarrow}$ the set of nodes appearing in $T_u$, that is, the subtree of $T$ rooted at $u$. On the other hand, we denote by $N_u^{\uparrow}$ the set of nodes of $T$ minus the set of nodes $N_u^{\downarrow}$. We define the *split number* of the decision tree $T$ to be

$$\max_{\text{node } u \text{ in } T} \left| \mathcal{F}(N_u^{\downarrow}) \cap \mathcal{F}(N_u^{\uparrow}) \right|.$$

Intuitively, the split number of a decision tree $T$ is a measure of the interaction (number of common features) between the subtrees of the form $T_u$ and their exterior. A small split number allows us to essentially treat each subtree $T_u$ independently (in particular, the left and right subtrees below any node), which in turn leads to efficient algorithms for the problems Compute-Minimum-SR and Compute-Minimal-SR.

**Theorem 4.** *Let $c \geq 1$ be a fixed integer. Both Compute-Minimum-SR and Compute-Minimal-SR can be solved in polynomial time for decision trees with split number at most $c$.*

*Proof Sketch.* It suffices to provide a polynomial time algorithm for Compute-Minimum-SR. (The same algorithm works for Compute-Minimal-SR as a minimum $\delta$-SR is in particular minimal.) In turn, using standard arguments, it is enough to provide a polynomial time algorithm for the following decision problem Check-Minimum-SR: Given a tuple $(T, \boldsymbol{x}, \delta, k)$, where $T$ is a decision tree of dimension $n$, $\boldsymbol{x} \in \{0, 1, \bot\}^n$ is a partial instance, $\delta \in (0, 1]$, and $k \geq 0$ is an integer, decide whether there is a partial instance $\boldsymbol{y} \subseteq \boldsymbol{x}$ such that $n - |\boldsymbol{y}|_{\bot} \leq k$ and $\Pr_{\boldsymbol{z}}[T(\boldsymbol{z}) = 1 \mid \boldsymbol{z} \in \text{COMP}(\boldsymbol{y})] \geq \delta$.

In order to solve Check-Minimum-SR over an instance $(T, \boldsymbol{x}, \delta, k)$, where $T$ has split number at most $c$, we apply dynamic programming over $T$ in a bottom-up manner. Let $Z \subseteq \{1, \ldots, n\}$ be the set of features defined in $\boldsymbol{x}$, that is, features $i$ with $\boldsymbol{x}[i] \neq \bot$. Those are the features we could eventually remove when looking for $\boldsymbol{y}$. For each node $u$ in $T$, we solve a polynomial number of subproblems over the subtree $T_u$. We define

$$\text{Int}(u) := \mathcal{F}(N_u^{\downarrow}) \cap \mathcal{F}(N_u^{\uparrow}) \cap Z \qquad \text{New}(u) := \left( \mathcal{F}(N_u^{\downarrow}) \setminus \text{Int}(u) \right) \cap Z.$$

In other words, $\text{Int}(u)$ are the features appearing both inside and outside $T_u$, while $\text{New}(u)$ are the features only inside $T_u$, that is, the new features introduced below $u$. Both sets are restricted to $Z$ as features not in $Z$ play no role in the process.

Each particular subproblem is indexed by a possible size $s \in \{0, \dots, k\}$ and a possible set $J \subseteq \mathsf{Int}(u)$ and the goal is to compute the quantity:

$$p_{u,s,J} := \max_{\boldsymbol{y} \in \mathcal{C}_{u,s,J}} \Pr_{\boldsymbol{z}}[T_u(\boldsymbol{z}) = 1 \mid \boldsymbol{z} \in \mathrm{COMP}(\boldsymbol{y})],$$

where $\mathcal{C}_{u,s,J}$ is the space of partial instances $\boldsymbol{y} \subseteq \boldsymbol{x}$ with $n - |\boldsymbol{y}|_\perp \leq s$ and such that $\boldsymbol{y}[i] = \boldsymbol{x}[i]$ for $i \in J$ and $\boldsymbol{y}[i] = \perp$ for $i \in \mathsf{Int}(u) \setminus J$. In other words, the set $J$ fixes the behavior on $\mathsf{Int}(u)$ (keep features in $J$, remove features in $\mathsf{Int}(u) \setminus J$) and hence the maximization occurs over choices on the set $\mathsf{New}(u)$ (which features are kept and which features are removed). The key idea is that $p_{u,s,J}$ can be computed inductively using the information already computed for the children $u_1$ and $u_2$ of $u$. Intuitively, this holds since the common features between $T_{u_1}$ and $T_{u_2}$ are at most $c$, which is a fixed constant, and hence we can efficiently synchronize the information stored for $u_1$ and $u_2$. Finally, to solve the instance $(T, \boldsymbol{x}, \delta, k)$ we simply check whether $p_{r,k,\emptyset} \geq \delta$, for the root $r$ of $T$. $\qquad \square$

## 5.2 Monotonicity

*Monotonic* classifiers have been studied in the context of XAI as they often present tractable cases for different explanations, as shown by Marques-Silva et al. (2021). The computation of minimal sufficient reasons for monotone models was known to be in PTIME since the work of Goldsmith et al. (2005). We show that this is also the case for computing minimal $\delta$-SRs under a mild assumption on the class of models.

Let us define the ordering $\preceq$ for instances in $\{0, 1\}^n$ as follows:

$$\boldsymbol{x} \preceq \boldsymbol{z} \quad \text{iff} \quad \boldsymbol{x}[i] \leq \boldsymbol{z}[i], \text{ for all } i \in \{1, \dots, n\}.$$

We can now define monotonicity as follows. A model $\mathcal{M}$ of dimension $n$ is said to be *monotone* if for every pair of instances $\boldsymbol{x}, \boldsymbol{z} \in \{0, 1\}^n$, it holds that $\boldsymbol{x} \preceq \boldsymbol{z} \implies \mathcal{M}(\boldsymbol{x}) \leq \mathcal{M}(\boldsymbol{z})$.

We now prove that the problem of computing minimal probabilistic sufficient reasons can be solved in polynomial time for any class $\mathfrak{C}$ of monotone Boolean models for which the problem of counting positive completions can be solved efficiently. Formally, the latter problem is defined as follows: given a model $\mathcal{M} \in \mathfrak{C}$ of dimension $n$ and a partial instance $\boldsymbol{y} \in \{0, 1, \perp\}^n$, compute $|\{\boldsymbol{x} \in \mathrm{COMP}(\boldsymbol{y}) \mid \mathcal{M}(\boldsymbol{x}) = 1\}|$. We call this problem $\mathfrak{C}$-Count-Positive-Completions

**Theorem 5.** *Let $\mathfrak{C}$ be a class of monotone Boolean models such that the problem $\mathfrak{C}$-Count-Positive-Completions can be solved in polynomial time. Then Compute-Minimal-SR can be solved in polynomial time over $\mathfrak{C}$.*

*Proof sketch.* Consider a partial instance $\boldsymbol{y}$ of dimension $n$ and $i \in \{1, \dots, n\}$. Suppose $\boldsymbol{y}[i] \neq \perp$. We write $\boldsymbol{y} \setminus \{i\}$ for the partial instance $\boldsymbol{y}'$ that is exactly equal to $\boldsymbol{y}$ save for the fact that $\boldsymbol{y}'[i] = \perp$. We make use of the following lemma, which is a probabilistic counterpart to Observation 1.

**Lemma 1.** *Let $\mathfrak{C}$ be a class of monotone models, $\mathcal{M} \in \mathfrak{C}$ a model of dimension $n$, and $\boldsymbol{x} \in \{0, 1\}^n$ an instance. Consider any $\delta \in (0, 1]$. Then if $\boldsymbol{y} \subseteq \boldsymbol{x}$ is a $\delta$-SR for $\boldsymbol{x}$ under $\mathcal{M}$ which is not minimal, then there is a partial instance $\boldsymbol{y} \setminus \{i\}$, for some $i \in \{1, \dots, n\}$, that is also a $\delta$-SR for $\boldsymbol{x}$ under $\mathcal{M}$.*

With this lemma we can prove Theorem 5. In fact, it can be shown then that a slight variant of Algorithm 1 computes in polynomial time a minimal $\delta$-SR for $\boldsymbol{x}$ under $\mathcal{M}$. $\qquad \square$

As a corollary, the computation of minimal probabilistic sufficient reasons can be carried out in polynomial time not only over monotone decision trees, but also over monotone free BDDs.

**Corollary 1.** *The problem of computing minimal $\delta$-SRs can be solved in polynomial time over the class of monotone free BDDs.*

## 6  SAT to the Rescue!

Despite the theoretical intractability results presented earlier on, many NP-complete problems can be solved over practical instances with the aid of *SAT solvers*. By definition, any NP-complete problem can be *encoded* as a CNF satisfiability (SAT) problem, which is then solved by a highly optimized program, a *SAT solver*. In particular, if a satisfying assignment is found for the CNF instance, one

can translate such an assignment back to a solution for the original problem. In fact, this paradigm has been successfully used in the literature for other explainability problems (Ignatiev et al., 2021, 2022; Izza et al., 2020b). The effectiveness of this approach is highly dependent on the particular encoding being used (Biere et al., 2009), where the aspect that arguably impacts performance the most is the size of the encoding, measured as the number of clauses of the resulting CNF formula.

In this section we present an encoding that uses some standard automated reasoning tools (e.g., *sequential encondings* (Sinz, 2005)) combined with ad-hoc *bit-blasting* (Biere et al., 2009), where the arithmetic operations required for probabilistic reasoning are implemented as Boolean circuits and then encoded as CNF by Tseitin transformations. The appendix describes experimentation both over synthetic datasets and standard datasets (MNIST) and reports empirical results. A recent report by Izza et al. (2022) also uses automated reasoning for this problem, although through an SMT solver.

Let us consider the following decision version of the Compute-Minimum-SR problem.

| | |
|---|---|
| PROBLEM: : | Decide-Minimum-SR |
| INPUT : | A decision tree $T$ of dimension $n$, an instance $\boldsymbol{x}$ of dimension $n$, an integer $k \leq n$, and $\delta \in (0, 1]$ |
| OUTPUT : | Yes if there is a $\delta$-SR $\boldsymbol{y}$ for $\boldsymbol{x}$ under $T$ with $|\boldsymbol{y}|_\perp \leq k$, and No otherwise. |

This problem is NP-complete. Membership is already proven by Izza et al. (2021). Hardness follows directly from Theorem 2, as if one were able to solve Decide-Minimum-SR in polynomial time, then a simple binary search over $k$ would allow solving Compute-Minimum-SR in polynomial time.

## 6.1 Deterministic Encoding

Let us first propose an encoding for the particular case of $\delta = 1$. First, create Boolean variables $f_i$ for $i \in \{1, \ldots, n\}$, with $f_i$ representing that $\boldsymbol{y}[i] \neq \perp$, where $\boldsymbol{y}$ is the desired $\delta$-SR. Then, for every node $u$ of the tree $T$, create a variable $r_u$ representing that node $u$ is *reachable* by a completion of $\boldsymbol{y}$, meaning that there exists a completion of $\boldsymbol{y}$ that goes through node $u$ when evaluated over $T$. We then want to enforce that:

1. $r_{\text{ROOT}} = 1$, where ROOT is the root of $T$. This means that the root is always reachable.

2. The desired $\delta$-SR $\boldsymbol{y}$ satisfies $n - |\boldsymbol{y}|_\perp \leq k$, meaning that $\sum_{i=1}^{n} f_i \leq k$.

3. If $T(\boldsymbol{x}) = 1$, and $F$ is the set of **false** leaves of $T$, then $r_\ell = 0$ for every $\ell \in F$. This means that if we want to explain a positive instance, the completions of the explanation $\boldsymbol{y}$ must all be positive (recall we assume $\delta = 1$), and thus no **false** leaf should be reachable. Conversely, If $T(\boldsymbol{x}) = 0$, and $G$ is the set of **true** leaves of $T$, then $r_\ell = 0$ for every $\ell \in G$.

4. The semantics of reachability is *consistent*: if a node $u$ is reachable, and its labeled with feature $i$, then if $f_i = 0$, meaning that feature $i$ is undefined in $\boldsymbol{y}$, both children of $u$ should be reachable too. In case $f_i = 1$, only the child along the edge corresponding to $\boldsymbol{x}[i]$ should be reachable.

Let us analyze the size of this encoding. Condition 1 requires a single clause to be enforced. Condition 2 can be enforced with $O(nk)$ variables and clauses using the linear encoding of Sinz (2005). Condition 3 requires at most one clause per leaf of $T$ and thus at most $O(|T|)$ clauses. Condition 4 can be implemented with a constant number of clauses per node. We thus incur in a total of $O(nk + |T|)$ variables and clauses, which is pretty much optimal considering $\Omega(n + |T|)$ is a lower bound on the representation of the input.

Note as well that from a satisfying assignment we can trivially recover the desired explanation $\boldsymbol{y}$ as

$$\boldsymbol{y}[i] = \begin{cases} \boldsymbol{x}[i] & \text{if } f_i = 1, \\ \perp & \text{otherwise.} \end{cases}$$

An efficient encoding supporting values of $\delta$ different than 1 is significantly more challenging and involved, and thus we only provide an outline here. An exhaustive description, together with the implementation, is provided in the supplementary material.

## 6.2 Probabilistic Encoding

In order to encode that $\Pr_{\boldsymbol{z}}[T(\boldsymbol{z}) = T(\boldsymbol{x}) \mid \boldsymbol{z} \in \mathrm{COMP}(\boldsymbol{y})] \geq \delta$, we will directly encode that $|\{\boldsymbol{z} \in \mathrm{COMP}(\boldsymbol{y}) \mid T(\boldsymbol{z}) = T(\boldsymbol{x})\}| \geq \lceil \delta 2^{|\boldsymbol{y}|_\perp} \rceil$, where we assume for simplicity that the ceiling can be take safely, in order to have a value we can represent by an integer. This assumption is not crucial. As before, we will have $f_i$ variables representing whether $\boldsymbol{y}[i] \neq \perp$ or not, and enforce that $\sum_{i=1}^n f_i \leq k$. Now define variables $c_j$ for $j \in \{0, \ldots, n\}$, such that $c_j$ is true exactly when $\sum_{i=1}^n f_i = j$. This can again be done efficiently via a linear encoding. Let $t_i$ for $i \in \{0, \ldots, n\}$ a series of variables that represent the bits of an integer $t$. The integer $t$ will correspond to the value of $\lceil \delta 2^{|\boldsymbol{y}|_\perp} \rceil$. Note that as $|\boldsymbol{y}|_\perp$ can only take $n+1$ different values, there are only $n+1$ different values that $t$ can take. Moreover, the value of the $c_j$ variables completely determines the value of $t$, and thus we can manually encode how the $c_j$ variables determine the bits $t_i$. We can now assume that we have access to $t$ through its bits $t_i$. Our goal now is to build a binary representation of

$$\alpha = |\{\boldsymbol{z} \in \mathrm{COMP}(\boldsymbol{y}) \mid T(\boldsymbol{z}) = T(\boldsymbol{x})\}|,$$

so that we can then implement the condition $\alpha \geq t$ with a Boolean circuit on their bits.

In order to represent $\alpha$, we will decompose the number of instances $\boldsymbol{z}$ according to the leaves of $T$ as follows. If $L$ is the set of leaves of $T$ whose label matches $T(\boldsymbol{x})$, then

$$|\{\boldsymbol{z} \in \mathrm{COMP}(\boldsymbol{y}) \mid T(\boldsymbol{z}) = T(\boldsymbol{x})\}| = \sum_{\ell \in L} |\{\boldsymbol{z} \in \mathrm{COMP}(\boldsymbol{y}) \mid T(\boldsymbol{z}) \rightsquigarrow \ell\}|,$$

where notation $T(\boldsymbol{z}) \rightsquigarrow \ell$ means that the evaluation of $\boldsymbol{z}$ under $T$ ends in the leaf $\ell$. For a given leaf $\ell$, we can compute its *weight* $w_\ell := |\{\boldsymbol{z} \in \mathrm{COMP}(\boldsymbol{y}) \mid T(\boldsymbol{z}) \rightsquigarrow \ell\}|$ as described next. Let $F_\ell$ be the set of labels appearing in the unique path from the root of $T$ down to $\ell$. Now let $u_\ell$ be the number of undefined features of $\boldsymbol{y}$ along said unique path, and thus $u_\ell$ can be defined as follows: $u_\ell := |F_\ell \setminus \{i \mid f_i = 1\}|$. The sets $F_\ell$ depend only on $T$ and thus can be precomputed. Therefore we can use a linear encoding again to define the values $u_\ell$. It is simple to observe now that $w_\ell = 2^{u_\ell}$. which means that by representing the values $u_\ell$ we can trivially represent the values $w_\ell$ in binary (they will simply consist of a 1 in their $(u_\ell + 1)$-th position right-to-left), and then implement a Boolean addition circuit to compute $\alpha = \sum_\ell w_\ell$. This concludes the outline of the encoding. Its number of variables clauses can be shown to be at most $O(n^2|T| + nk)$.

## 7   Conclusions and Future Work

The results proven in this paper suggest that minimal (or minimum) explanations might be hard to obtain in practice even for decision trees, especially in problems where the feature space has large dimension. A way to circumvent the limitations proven in our work is to relax the guarantee of minimality, or introduce some probability of error, as done in the work of Blanc et al. (2021). A promising direction of future research is to better understand the kind of guarantees and settings in which is still possible to obtain tractability.

Our work leaves open some interesting technical questions: (1) What is the *parameterized* complexity of computing minimum $\delta$-SRs over decision trees, assuming that the parameter is the size of the explanation one is looking for? It is not hard to see that W[2] hardness follows from our proofs (i.e., they are parameterized reductions all the way to Set Cover), but membership in any class is fully open. (2) Does Theorem 2 continue to hold for monotone decision trees? That is, is it the case that computing minimum $\delta$-SRs over monotone decision trees is hard for every fixed $\delta \in (0, 1]$? (3) Is it the case that the hardness of computation for minimal sufficient reasons holds for a *fixed* $\delta \in (0, 1]$? If so, does it hold for every fixed such a $\delta$, or only for some? (4) Is it possible to extend the positive behavior of decision trees with bounded split number to free BDDs?

## Acknowledgments

Barceló is funded by Fondecyt grant 1200967. Arenas and Barceló have been funded by ANID - Millennium Science Initiative Program - Code ICN17002. Barceló has also been funded by National Center for Artificial Intelligence CENIA FB210017, Basal ANID. Romero is funded by Fondecyt grant 11200956, the Data Observatory Foundation, and the National Center for Artificial Intelligence CENIA FB210017, Basal ANID. Subercaseaux is supported by NSF awards CCF-2015445, CCF1955785, and CCF2006953.

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
