# Appendix

**Organization.** The supplementary material is organized as follows. In section A, we provide a proof that there is no polynomial-time algorithm for the problem of computing minimum $\delta$-sufficient reasons (unless PTIME = NP), even when $\delta$ is a fixed value. In section B, we provide a proof that there is no polynomial-time algorithm for the problem of computing minimal $\delta$-sufficient reasons (unless PTIME = NP). In particular, we define in this section a decision problem that we prove to be NP-hard in Section B.1, and then we show in Section B.2 that this decision problem can be solved in polynomial-time if there exists a polynomial-time algorithm for the problem of computing minimal $\delta$-sufficient reasons. In Section C, we provide a proof that minimum and minimal $\delta$-sufficient reasons can be computed in polynomial-time when the split number (defined in Section 5.1) is bounded. Finally, we provide in Section D a proof of Lemma 1, which completes the proof of tractability of the problem of computing minimal $\delta$-sufficient reasons for each class of monotone Boolean models for which the problem of counting positive completions can be solved in polynomial time. Finally, we describe in Section E our experimental evaluation of the encodings given in Section 6.

## A  Proof of Theorem 2

**Theorem 2.** *Fix $\delta \in (0, 1]$. Then assuming that* PTIME $\neq$ NP*, there is no polynomial-time algorithm for $\delta$-*Compute-Minimum-SR*.*

Before we can prove this theorem, we will require some auxiliary lemmas. Given rational numbers $a, b$ with $a \leq b$, recall that notation $[a, b]$ refers to the set of rational numbers $x$ such that $a \leq x \leq b$ (and likewise for $[a, b)$).

**Lemma 2.** *Fix $\delta \in (0, 1)$. Given as input an integer $n$ one can build in $n^{O(1)}$ time a decision tree $T_\delta$ of dimension $n$, such that*

$$\left| \Pr_{\boldsymbol{z}}[T_\delta(\boldsymbol{z}) = 1 \mid \boldsymbol{z} \in \text{COMP}(\perp^n)] - \delta \right| \leq \frac{1}{2^n},$$

*and moreover, there exists an instance $\boldsymbol{x}^\dagger$ for $T_\delta$ such that every partial instance $\boldsymbol{y} \subseteq \boldsymbol{x}^\dagger$ holds*

$$\Pr_{\boldsymbol{z}}[T_\delta(\boldsymbol{z}) = 1 \mid \boldsymbol{z} \in \text{COMP}(\boldsymbol{y})] \leq \Pr_{\boldsymbol{z}}[T_\delta(\boldsymbol{z}) = 1 \mid \boldsymbol{z} \in \text{COMP}(\perp^n)].$$

*Proof.* Let $c = \lfloor \delta 2^n \rfloor$, and note that $\delta - \frac{1}{2^n} \leq \frac{c}{2^n} \leq \delta$, and thus $|\delta - \frac{c}{2^n}| \leq \frac{1}{2^n}$. This implies that we can prove the first part of the lemma by building in polynomial time a tree $T_c$ over $n$ variables, that has exactly $c$ different positive instances, as then its probability of accepting a random completion of $\perp^n$ will be exactly $\frac{c}{2^n}$. Note as well that $c < 2^n$ as $\delta < 1$.

As a first step, let us write $c$ in binary, obtaining

$$c = \alpha_0 2^0 + \alpha_1 2^1 + \cdots + \alpha_{n-1} 2^{n-1},$$

with $\alpha_i \in \{0, 1\}$ for each $i$. Then to build $T_c$ start by creating $n$ vertices, labeled 0 through $n-1$. These $n$ labels are the variables of $T_c$. For each $i \in \{1, \ldots, n-1\}$, connect vertex labeled $i$ to vertex labeled $i-1$ with a 0-edge, making vertex labeled $n-1$ the root of $T_c$. Then, for each vertex with label $i \in \{0, \ldots, n-1\}$, set its 1-edge towards a leaf with label **true** if $\alpha_i = 1$, and towards a leaf with label **false** if $\alpha_i = 0$. The 0-edge of vertex labeled 0 goes towards a leaf with label **false**. Now let us count how many different positive instances does $T_c$ have. We can do this by summing over all true leaves of $T_c$. Each true leaf comes from a 1-edge from a vertex labeled $i \in \{0, \ldots, n-1\}$. For every $i \in \{0, \ldots, n-1\}$, if the vertex labeled $i$ has a true leaf when following its 1-edge, then the number of instances reaching that true leaf is exactly $2^i$, as the variables whose value is not determined by the path to that leaf are those with labels less than $i$, which are exactly $i$ variables. Therefore, the number of different positive instances of $T_c$ along a 1-edge is the sum of $2^i$ for every $i$ such that $\alpha_i = 1$, which is exactly $c$. An example is given in Figure 3. This concludes the proof of the first part of the lemma as the construction is clearly polynomial in $n$. For the second part, let us build $\boldsymbol{x}^\dagger$ by setting $\boldsymbol{x}^\dagger[i] = 1 - \alpha_i$ for every $i \in \{0, \ldots, n-1\}$. In the example presented in Figure 3, we would build

$$\boldsymbol{x}^\dagger = (1, \quad 1, \quad 0, \quad 0, \quad 1).$$

We will now prove that for any $\boldsymbol{y} \subseteq \boldsymbol{x}^\dagger$, it holds that

$$\Pr_{\boldsymbol{z}}[T_c(\boldsymbol{z}) = 1 \mid \boldsymbol{z} \in \text{COMP}(\boldsymbol{y})] \leq \Pr_{\boldsymbol{z}}[T_c(\boldsymbol{z}) = 1 \mid \boldsymbol{z} \in \text{COMP}(\bot^n)].$$

We do this via a finite induction argument by strengthening our induction hypothesis; for $i \in \{0, \ldots, n-1\}$, let $T_c^i$ be the sub-tree of $T_c$ rooted at the vertex labeled $i$, and let us claim that for every $i \in \{0, \ldots, n-1\}$ we have that

$$\Pr_{\boldsymbol{z}}\big[T_c^i(\boldsymbol{z}) = 1 \mid \boldsymbol{z} \in \text{COMP}(\boldsymbol{y})\big] \leq \Pr_{\boldsymbol{z}}\big[T_c^i(\boldsymbol{z}) = 1 \mid \boldsymbol{z} \in \text{COMP}(\bot^n)\big],$$

which implies what we want to show when taking $i = n - 1$. The base case of the induction is when $i = 0$, in which case the claim trivially holds as if $\boldsymbol{y}[0] = \bot$ we have equality, and if $\boldsymbol{y}[0] \neq \bot$ then by construction

$$\Pr_{\boldsymbol{z}}\big[T_c^i(\boldsymbol{z}) = 1 \mid \boldsymbol{z} \in \text{COMP}(\boldsymbol{y})\big] = 0.$$

For the inductive case, let $i > 0$, and proceed by cases; if $\boldsymbol{y}[i] = \bot$, then by letting $t_i \in \{0, 1\}$ be an indicator variable for whether the leaf across the 1-edge from vertex $i$ is labeled **true** we have that

$$\begin{aligned}
\Pr_{\boldsymbol{z}}\big[T_c^i(\boldsymbol{z}) = 1 \mid \boldsymbol{z} \in \text{COMP}(\boldsymbol{y})\big] &= \frac{1}{2}t_i + \frac{1}{2}\Pr_{\boldsymbol{z}}\big[T_c^{i-1}(\boldsymbol{z}) = 1 \mid \boldsymbol{z} \in \text{COMP}(\boldsymbol{y})\big] \\
&\leq \frac{1}{2}t_i + \frac{1}{2}\Pr_{\boldsymbol{z}}\big[T_c^{i-1}(\boldsymbol{z}) = 1 \mid \boldsymbol{z} \in \text{COMP}(\bot^n)\big] \\
&= \Pr_{\boldsymbol{z}}\big[T_c^i(\boldsymbol{z}) = 1 \mid \boldsymbol{z} \in \text{COMP}(\bot^n)\big],
\end{aligned}$$

where the inequality has used the inductive hypothesis. On the other hand, if $\boldsymbol{y}[i] = 1$, that implies $\boldsymbol{x}^\dagger[i] = 1$ and thus $\alpha_i = 0$, which means the leaf across the 1-edge from vertex $i$ is labeled with **false**, and thus

$$Pr_{\boldsymbol{z}}\big[T_c^i(\boldsymbol{z}) = 1 \mid \boldsymbol{z} \in \text{COMP}(\boldsymbol{y})\big] = 0,$$

which trivially satisfies the claim. For the last case, if $\boldsymbol{y}[i] = 0$, then $\boldsymbol{x}^\dagger[i] = 0$ and thus $\alpha_i = 1$, which means the leaf across the 1-edge from vertex $i$ is labeled with **true**. Therefore we have

$$\begin{aligned}
\Pr_{\boldsymbol{z}}\big[T_c^i(\boldsymbol{z}) = 1 \mid \boldsymbol{z} \in \text{COMP}(\boldsymbol{y})\big] &= \Pr_{\boldsymbol{z}}\big[T_c^{i-1}(\boldsymbol{z}) = 1 \mid \boldsymbol{z} \in \text{COMP}(\boldsymbol{y})\big] \\
&\leq \Pr_{\boldsymbol{z}}\big[T_c^{i-1}(\boldsymbol{z}') = 1 \mid \boldsymbol{z} \in \text{COMP}(\bot^n)\big] \\
&\leq \frac{1}{2} + \frac{1}{2}\Pr_{\boldsymbol{z}}\big[T_c^{i-1}(\boldsymbol{z}) = 1 \mid \boldsymbol{z} \in \text{COMP}(\bot^n)\big] \quad \text{(as } \Pr[\cdot] \leq 1) \\
&= \Pr_{\boldsymbol{z}}\big[T_c^i(\boldsymbol{z}) = 1 \mid \boldsymbol{z} \in \text{COMP}(\bot^n)\big].
\end{aligned}$$

This completes the induction argument, and thus we conclude the proof of the lemma. $\qquad\square$

We are now ready to prove Proposition 2. We will use notation $\log(x)$ to refer to the logarithm in base 2 of $x$.

*Proof of Theorem 2.* We will prove that deciding whether a $\delta$-SR of size $k$ exists is NP-hard. We will reduce from the case $\delta = 1$, proved NP-hard by Barceló et al. (2020). We assume of course that $\delta < 1$, as otherwise the result is already known.

Let $(T, \boldsymbol{x}, k)$ be an input of the Minimum Sufficient Reason problem (i.e., $\delta = 1$), and let $n$ be the dimension of $T$ and $\boldsymbol{x}$. Assume without loss of generality that $T(\boldsymbol{x}) = 1$. If the given input of Minimum Sufficient Reason is positive, then there is a partial instance $\boldsymbol{y} \subseteq \boldsymbol{x}$ with $|\boldsymbol{y}|_\bot \geq n - k$ such that $\Pr_{\boldsymbol{z}}[T(\boldsymbol{z}) = 1 \mid \boldsymbol{z} \in \text{COMP}(\boldsymbol{y})] = 1$, and otherwise for every partial instance $\boldsymbol{y} \subseteq \boldsymbol{x}$ with $|\boldsymbol{y}|_\bot \geq n - k$ it holds that

$$\Pr_{\boldsymbol{z}}[T(\boldsymbol{z}) = 0 \mid \boldsymbol{z} \in \text{COMP}(\boldsymbol{y})] \geq \frac{1}{2^n}.$$

Let us build a tree $F_\delta$ with $3n + 3 + \lceil \log(1/\delta) \rceil$ variables as follows. First build $T_\delta$ of dimension $2n + 3 + \lceil \log(1/\delta) \rceil$ by using Lemma 2, and then replace every true leaf of $T_\delta$ by a copy of $T$.

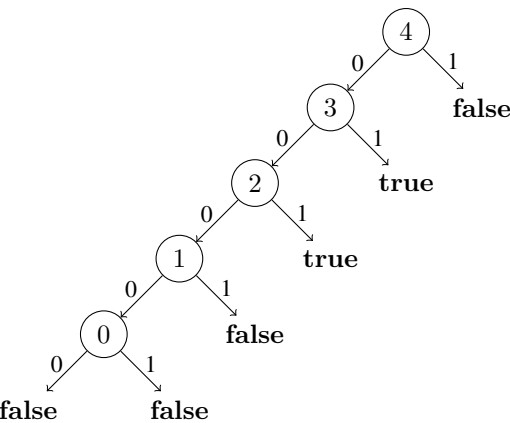

Figure 3: Example of $T_c$, the tree constructed in the proof of Lemma 2, for $n = 5$ and $\delta = \frac{2}{5}$. In this case $c = 12 = 2^2 + 2^3$. Note that $\Pr_{\boldsymbol{z}}[T_c(\boldsymbol{z}) = 1 \mid \boldsymbol{z} \in \mathrm{COMP}(\bot^n)] = \frac{c}{2^n} = \frac{12}{32}$, and $\left| \frac{2}{5} - \frac{12}{32} \right| = \frac{1}{40} < \frac{1}{32}$.

Assume the $2n + 3 + \lceil \log(1/\delta) \rceil$ variables of $T_\delta$ are disjoint from the $n$ variables that appear in $T$, and thus $F_\delta$ has the proposed number of variables. An example of the construction of $F_\delta$ is illustrated in Figure 4.

Define
$$\delta' := \Pr_{\boldsymbol{z}}\Big[ T_\delta(\boldsymbol{z}) = 1 \mid \boldsymbol{z} \in \mathrm{COMP}\Big( \bot^{2n+2+\lceil \log(1/\delta) \rceil} \Big) \Big],$$
and recall that $|\delta' - \delta| \leq \frac{1}{2^{2n+3+\lceil \log(1/\delta) \rceil}}$. Now, let us build a final decision tree $T_\delta^\star$ with $4n + k + 4 + \lceil \log(1/\delta') \rceil + \lceil \log(1/\delta) \rceil$ variables as follows. Create $\ell := n + k + 1 + \lceil \log(1/\delta') \rceil$ vertices, labeled $r_i$ for $i \in \{1, \dots, \ell\}$, and assume these labels are disjoint from the ones used in $F_\delta$. Let $r_1$ be the root of $T_\delta^\star$, and for each $i \in \{1, \dots, \ell - 1\}$, connect vertex labeled with $r_i$ to vertex labeled with $r_{i+1}$ using a 0-edge. The 0-edge from vertex labeled $r_\ell$ goes towards a leaf labeled with **true**. The 1-edge from every vertex $r_i$ goes towards the root of a different copy of $F_\delta$. Note that this construction, illustrated in Figure 5, takes polynomial time. Now, consider the instance $\boldsymbol{x}^\star$ that is defined (i) exactly as $\boldsymbol{x}$ for the variables of $T$, (ii) exactly as in the instance $\boldsymbol{x}^\dagger$ coming from Lemma 2 for the variables of $T_\delta$ in $F_\delta$, and (iii) with all variables $r_i$ set to 0. Note that $T_\delta^\star(\boldsymbol{x}^\star) = 1$. Now we prove both directions of the reduction separately. Assume fir that the instance $(T, \boldsymbol{x}, k)$ is a positive instance for Minimum Sufficient Reason. Then we claim that there is a $\delta$-SR for $T_\delta^\star$ of size at most $k$. Indeed, let $\boldsymbol{y} \subseteq \boldsymbol{x}$ be a sufficient reason for $\boldsymbol{x}$ under $T$ with at most $k$ defined components. Then consider the partial instance $\boldsymbol{y}^\star \subseteq \boldsymbol{x}^\star$, that is only defined in the components where $\boldsymbol{y}$ is defined. Now let us study $\Pr_{\boldsymbol{z}}[T_\delta^\star(\boldsymbol{z}') = 1 \mid \boldsymbol{z} \in \mathrm{COMP}(\boldsymbol{y}^\star)]$. The probability that $\boldsymbol{z}$ ends up in the true leaf on the 0-edge from vertex $r_\ell$ is $\frac{1}{2^\ell}$. In any other case, $\boldsymbol{z}$ takes a path that goes into a copy of $F_\delta$, where its probability of acceptance is $\delta' \geq \delta - \frac{1}{2^{2n+3+\lceil \log(1/\delta) \rceil}}$ because of Lemma 2 and using that $\boldsymbol{y}^\star$ is undefined for all the variables of $T_c$. These two facts imply that

$$\Pr_{\boldsymbol{z}}[T_\delta^\star(\boldsymbol{z}) = 1 \mid \boldsymbol{z} \in \mathrm{COMP}(\boldsymbol{y}^\star)] \geq \frac{1}{2^\ell} + \delta - \frac{1}{2^{2n+3+\lceil \log(1/\delta) \rceil}}.$$

Now consider that

$$\delta \leq \delta' + \frac{1}{2^{2n+3+\lceil \log(1/\delta) \rceil}}$$
$$\leq \delta' + \frac{1}{2^{2n+3}} \cdot \frac{1}{2^{\lceil \log(1/\delta) \rceil}}$$
$$\leq \delta' + \frac{1}{2^{2n+3}} \cdot \frac{1}{2^{\log(1/\delta)}}$$
$$= \delta' + \frac{\delta}{2^{2n+3}},$$

from where

$$\delta \leq \delta'\left( 1 - \frac{1}{2^{2n+3}} \right),$$

and thus

$$\log(1/\delta) \geq \log(1/\delta') + \log\left(1 - \frac{1}{2^{2n+3}}\right)$$

$$= \log(1/\delta') - \log\left(\frac{2^{2n+3}}{2^{2n+3}-1}\right)$$

$$= \log(1/\delta') - 2n - 3 + \log(2^{2n+3}-1)$$

$$\geq \log(1/\delta') - 2n - 3 + 2n + 2 \qquad \text{(using } 2^{2n+3} - 1 \geq 2^{2n+2})$$

$$= \log(1/\delta') - 1.$$

From this we obtain that

$$\ell = n + k + 1 + \lceil \log(1/\delta') \rceil$$

$$\leq 2n + 1 + \lceil \log(1/\delta') \rceil$$

$$\leq 2n + 1 + \log(1/\delta') + 1$$

$$\leq 2n + 3 + \log(1/\delta)$$

$$\leq 2n + 3 + \lceil \log(1/\delta) \rceil,$$

which allows us to conclude that

$$\Pr_{\boldsymbol{z}}[T_\delta^\star(\boldsymbol{z}) = 1 \mid \boldsymbol{z} \in \text{COMP}(\boldsymbol{y}^\star)] \geq \frac{1}{2^\ell} + \delta - \frac{1}{2^{2n+3+\lceil \log(1/\delta) \rceil}} \geq \delta.$$

On the other hand, if $(T, \boldsymbol{x}, k)$ is a negative instance for Minimum Sufficient Reason, consider any partial $\boldsymbol{y}^\star$ with at most $k$ defined components, and note that by hypothesis we have that $\Pr_{\boldsymbol{z}}[T(\boldsymbol{z}) = 1 \mid \boldsymbol{z} \in \text{COMP}(\boldsymbol{y}^\star)] \leq 1 - \frac{1}{2^n}$. This implies, together with the second part of Lemma 2, that

$$\Pr_{\boldsymbol{z}}[F_\delta(\boldsymbol{z}) = 1 \mid \boldsymbol{z} \in \text{COMP}(\boldsymbol{y}^\star)] \leq \delta\left(1 - \frac{1}{2^n}\right),$$

and thus subsequently

$$\Pr_{\boldsymbol{z}}[T_\delta^\star(\boldsymbol{z}) = 1 \mid \boldsymbol{z} \in \text{COMP}(\boldsymbol{y}^\star)] \leq \delta\left(1 - \frac{1}{2^n}\right) + \frac{1}{2^{\ell-k}},$$

by using that with at most $k$ defined components in $T_\delta^\star$, the probability of reaching the true leaf across the 0-edge from $r_\ell$ is at most $\frac{1}{2^{\ell-k}}$. To conclude, note that

$$\Pr_{\boldsymbol{vz}}[T_\delta^\star(\boldsymbol{z}) = 1 \mid \boldsymbol{z} \in \text{COMP}(\boldsymbol{y}^\star)] \leq \delta\left(1 - \frac{1}{2^n}\right) + \frac{1}{2^{\ell-k}}$$

$$= \delta - \frac{\delta}{2^n} + \frac{1}{2^{n+1+\lceil \log(1/\delta') \rceil}}$$

$$\leq \delta - \frac{\delta}{2^n} + \frac{1}{2^{n+1+\log(1/\delta')}}$$

$$= \delta + \frac{(\delta' - \delta) - \delta}{2^{n+1}}$$

$$\leq \delta + \frac{\frac{1}{2^{2n+3+\lceil \log(1/\delta) \rceil}} - \delta}{2^{n+1}}$$

$$\leq \delta + \frac{\frac{\delta}{2^{2n+3}} - \delta}{2^{n+1}}$$

$$= \delta + \delta\left(\frac{-1 + \frac{1}{2^{2n+3}}}{2^{n+1}}\right)$$

$$< \delta.$$

We have thus concluded that $\boldsymbol{y}^\star$ is a $\delta$-SR for $\boldsymbol{x}^\star$ over $T_\delta^\star$ if and only if $(T, \boldsymbol{x}, k)$ is a positive instance of Minimum Sufficient Reason, which completes our reduction. □

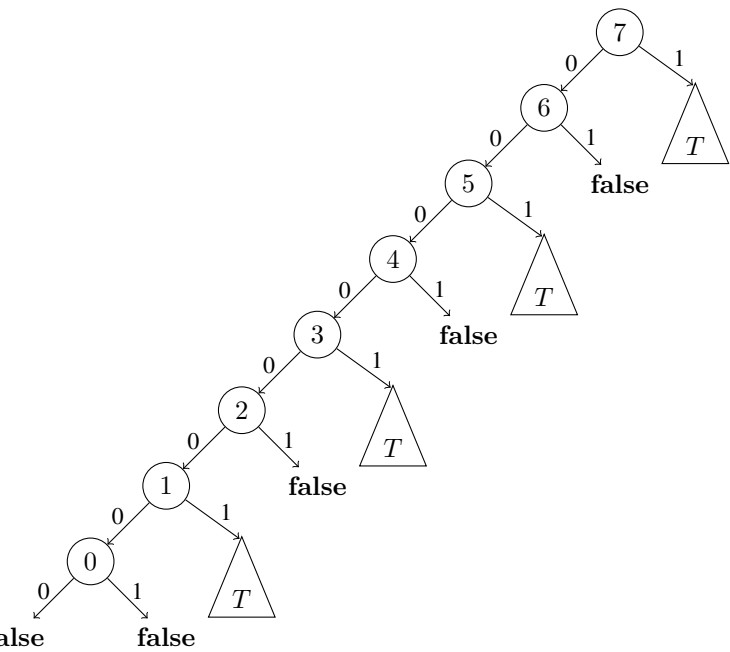

Figure 4: Illustration of the construction of $F_\delta$ for $\delta = \frac{2}{3}$ and $n = 2$. Thus $2n + 3 + \lceil \log(1/\delta) \rceil = 8$ and $c = \lfloor \frac{2}{3} \cdot 2^8 \rfloor = 170 = 2^7 + 2^5 + 2^3 + 2^1$.

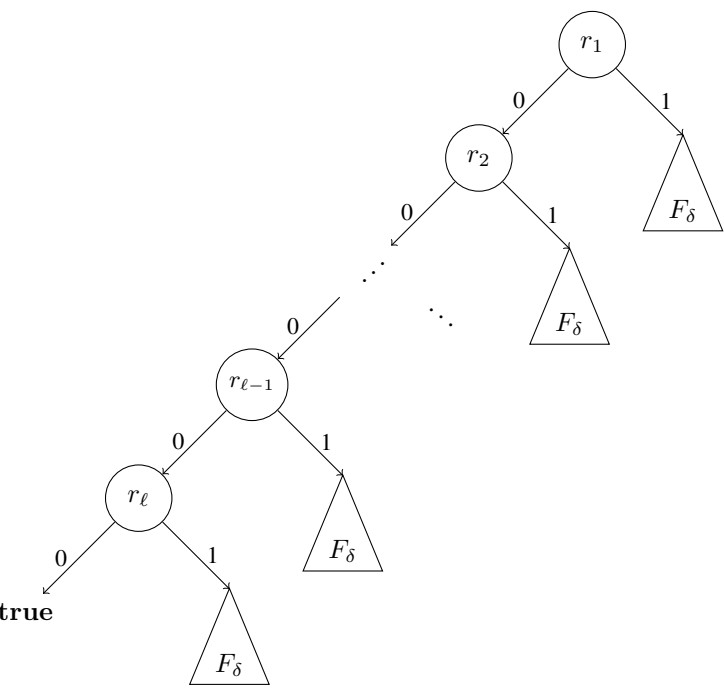

Figure 5: Illustration of the construction of $T_\delta^\star$. Recall that $\ell = n + k + 1 + \lceil \log(1/\delta') \rceil$.

# B Proof of Theorem 3

**Theorem 3.** *Assuming that* PTIME $\neq$ NP*, there is no polynomial-time algorithm for* Compute-Minimal-SR.

We start by explaining the high-level idea of the proof. First, we will show that the following decision problem, called Check-Sub-SR, is NP-hard.

| | |
|---|---|
| PROBLEM : | Check-Sub-SR |
| INPUT : | $(T, \boldsymbol{y})$, for a decision tree $T$ of dimension $n$, and partial instance $\boldsymbol{y} \in \{0, 1, \bot\}^n$ |
| OUTPUT : | Yes, if there is a partial instance $\boldsymbol{y}' \subsetneq \boldsymbol{y}$ such that $\Pr_{\boldsymbol{z}}[T(\boldsymbol{z}) = 1 \mid \boldsymbol{z} \in \mathrm{COMP}(\boldsymbol{y}')] \geq \Pr_{\boldsymbol{z}}[T(\boldsymbol{z}) = 1 \mid \boldsymbol{z} \in \mathrm{COMP}(\boldsymbol{y})]$, and No otherwise |

We then show that if Compute-Minimal-SR admits a polynomial time algorithm, then Check-Sub-SR is in PTIME, which contradicts the assumption that PTIME $\neq$ NP. The latter reduction requires an involved construction exploiting certain properties of the hard instances for Check-Sub-SR.

To show that Check-Sub-SR is NP-hard, we use a polynomial time reduction from a decision problem over formulas in CNF, called Minimal-Expected-Clauses, and which we also show to be NP-hard. Both the NP-hardness of Minimal-Expected-Clauses and the reduction from Minimal-Expected-Clauses to Check-Sub-SR may be of independent interest.

We now define the problem Minimal-Expected-Clauses. Let $\varphi$ be a CNF formula over variables $X = \{x_1, \ldots, x_n\}$. Assignments and partial assignments of the variables in $X$, as well as the notions of subsumption and completions over them, are defined in exactly the same way as for partial instances over features. For a partial assignment $\mu$ over $X$, we denote by $E(\varphi, \mu)$ the expected number of clauses of $\varphi$ satisfied by a random completion of $\mu$. We then consider the following problem for fixed $k \geq 2$ (recall that a $k$-CNF formula is a CNF formula where each clause has at most $k$ literals):

| | |
|---|---|
| PROBLEM : | $k$-Minimal-Expected-Clauses |
| INPUT : | $(\varphi, \sigma)$, for $\varphi$ a $k$-CNF formula and $\sigma$ a partial assignment |
| OUTPUT : | Yes, if there is a partial assignment $\mu \subsetneq \sigma$ such that $E(\varphi, \mu) \geq E(\varphi, \sigma)$ and No otherwise |

We show that $k$-Minimal-Expected-Clauses is NP-hard even for $k = 2$, via a reduction from the well-known clique problem. Finally, the reduction from $k$-Minimal-Expected-Clauses to Check-Sub-SR builds an instance $(T, \boldsymbol{y})$ from $(\varphi, \sigma)$ in a way that there is a direct correspondence between partial assignments $\mu \subseteq \sigma$ and partial instances $\boldsymbol{y}' \subseteq \boldsymbol{y}$, satisfying that

$$\Pr_{\boldsymbol{z}}[T(\boldsymbol{z}) = 1 \mid \boldsymbol{z} \in \mathrm{COMP}(\boldsymbol{y}')] = \frac{E(\varphi, \mu)}{m},$$

where $m$ is the number of clauses of $\varphi$. This implies that $(\varphi, \sigma)$ is a Yes-instance for $k$-Minimal-Expected-Clauses if and only if $(T, \boldsymbol{y})$ is a Yes-instance for Check-Sub-SR.

Below in Section B.1, we show the NP-hardness of $k$-Minimal-Expected-Clauses and the reduction from $k$-Minimal-Expected-Clauses to Check-Sub-SR, obtaining the NP-hardness of Check-Sub-SR. We conclude in Section B.2 with the reduction from Check-Sub-SR to Compute-Minimal-SR, obtaining Theorem 3.

## B.1 Hardness of the decision problem

We start with some simple observations regarding the number $E(\varphi, \mu)$ for a CNF formula $\varphi$ and a partial assignment $\mu$. By linearity of expectation, we can write $E(\varphi, \mu)$ as the sum

$$E(\varphi, \mu) = \sum_{C \text{ clause of } \varphi} \mathsf{Prob}_{C,\mu}, \tag{1}$$

where $\mathsf{Prob}_{C,\mu}$ is the probability that a random completion of $\mu$ satisfies the clause $C$.

In turn, the probabilities $\mathsf{Prob}_{C,\mu}$ can be easily computed as:

- $\mathsf{Prob}_{C,\mu} = 1$, if there is a positive literal $x$ in $C$ with $\mu(x) = 1$; or there is a negative literal $\neg x$ in $C$ with $\mu(x) = 0$.
- $\mathsf{Prob}_{C,\mu} = 1 - \frac{1}{2^\eta}$, where $\eta$ is the number of literals in $C$ of the form $x$ or $\neg x$ with $\mu(x) = \bot$.

Finally, note that for an assignment $\sigma$, $E(\varphi, \sigma)$ is simply the number of clauses of $\varphi$ satisfied by $\sigma$.

Now we are ready to show our first hardness result:

**Proposition 1.** $k$-Minimal-Expected-Clauses *is NP-hard even for* $k = 2$.

*Proof.* We reduce from the clique problem. Recall this problem asks, given a graph $G$ and an integer $k \geq 3$, whether there is a *clique* of size $k$, that is, a set $K$ of $k$ vertices such that there is an edge between any pair of distinct vertices from $K$. Let $G$ be a graph and $k \geq 3$. We can assume without loss of generality that $k$ is odd and the degree of every vertex $x$ of $G$, denoted by $\deg(x)$, is at least $k - 1$; if $k$ is even we can consider the equivalent instance given by the graph $G'$ that extends $G$ with a fresh node connected via an edge with all the other nodes and $k' = k + 1$. On the other hand, we can iteratively remove vertices of degree less than $k - 1$ as those cannot be part of any clique of size $k$. We define an instance $(\varphi, \sigma)$ for 2-Minimal-Expected-Clauses as follows. The variables of $\varphi$ are the nodes of $G$. For each variable $x$ we have the following clauses in $\varphi$:

- A clause $A_x = (\neg x)$. This clause $A_x$ is repeated $\frac{k-1}{2} + \deg(x) - (k-1)$ times in $\varphi$. Note this quantity is always a positive integer.

- A set of clauses $B_x = \{(x \vee \neg y_1), \ldots, (x \vee \neg y_{\deg(x)})\}$, where $y_1, \ldots, y_{\deg(x)}$ are the neighbors of $x$ in $G$. Each clause in $B_x$ appears only once in $\varphi$.

Additionally, for each set $\{x, y\}$ where $x \neq y$ and $\{x, y\}$ is *not* an edge in $G$, we have a clause $Z_{x,y} = (x \vee y)$ repeated $4e$ times in $\varphi$, where $e$ is the number of edges in $G$.

We define the assignment $\sigma$ such that $\sigma(x) = 1$, for all variables $x$ of $\varphi$.

For an arbitrary partial assignment $\mu$ to the variables of $\varphi$, with $\mu \subseteq \sigma$, we define

$$\mathsf{utility}_{\varphi,\sigma}(\mu) := E(\varphi, \mu) - E(\varphi, \sigma).$$

In particular, the instance $(\varphi, \sigma)$ is a Yes-instance for 2-Minimal-Expected-Clauses if and only if there is $\mu \subsetneq \sigma$ with $\mathsf{utility}_{\varphi,\sigma}(\mu) \geq 0$. By equation (1), we can write

$$\mathsf{utility}_{\varphi,\sigma}(\mu) = \sum_{C \text{ clause of } \varphi} \mathsf{utility}_{\varphi,\sigma}(\mu, C),$$

where $\mathsf{utility}_{\varphi,\sigma}(\mu, C)$ is defined as:

$$\mathsf{utility}_{\varphi,\sigma}(\mu, C) := \mathsf{Prob}_{C,\mu} - \mathsf{Prob}_{C,\sigma}.$$

We have that:

$$\mathsf{Prob}_{C,\sigma} = \begin{cases} 0 & \text{if } C = A_x \text{ for some variable } x \\ 1 & \text{if } C \in B_x \text{ for some variable } x \text{ or } C = Z_{x,y} \text{ for some set } \{x, y\} \end{cases}$$

On the other hand, for the probability $\mathsf{Prob}_{C,\mu}$ we have the following:

1. Assume $C = A_x = (\neg x)$ for some variable $x$. Then $\mathsf{Prob}_{C,\mu}$ is

   (a) $\frac{1}{2}$, if $\mu(x) = \bot$ (and hence $\mathsf{utility}_{\varphi,\sigma}(\mu, C) = \frac{1}{2}$), and
   (b) $0$ otherwise (then $\mathsf{utility}_{\varphi,\sigma}(\mu, C) = 0$).

2. Suppose $C = (x \vee \neg y) \in B_x$ for some variable $x$. Then $\mathsf{Prob}_{C,\mu}$ is

   (a) $\frac{3}{4}$ if $\mu(x) = \bot$ and $\mu(y) = \bot$ (and hence $\mathsf{utility}_{\varphi,\sigma}(\mu, C) = -\frac{1}{4}$),

(b) $\frac{1}{2}$ if $\mu(x) = \bot$ and $\mu(y) = 1$ (and then $\mathsf{utility}_{\varphi,\sigma}(\mu, C) = -\frac{1}{2}$), and

(c) 1 otherwise (then $\mathsf{utility}_{\varphi,\sigma}(\mu, C) = 0$).

3. Suppose $C = Z_{x,y} = (x \vee y)$ for some set $\{x, y\}$. Then $\mathsf{Prob}_{C,\mu}$ is

(a) $\frac{3}{4}$ if $\mu(x) = \bot$ and $\mu(y) = \bot$ (and hence $\mathsf{utility}_{\varphi,\sigma}(\mu, C) = -\frac{1}{4}$), and

(b) 1 otherwise (then $\mathsf{utility}_{\varphi,\sigma}(\mu, C) = 0$).

We now show the correctness of our construction. Suppose $G$ has a clique $K$ of size $k \geq 3$. Let $\mu$ be the partial assignment that sets $\mu(x) = \bot$ if $x \in K$ and $\mu(x) = 1$ if $x \notin K$. Note that $\mu \subsetneq \sigma$. We claim that $\mathsf{utility}_{\varphi,\sigma}(\mu) = 0$ and hence $(\varphi, \sigma)$ is a Yes-instance. Let $C$ be a clause in $\varphi$. If $C$ is of the form $Z_{x,y}$, then $\mathsf{utility}_{\varphi,\sigma}(\mu, C) = 0$. Indeed, by construction, $\{x, y\}$ is not an edge, and since $K$ is a clique, then $\mu(x) = 1$ or $\mu(y) = 1$. This means we are always in case 3(b) above. If $x \notin K$ and $C$ is of the form $A_x$ or belongs to $\mathcal{B}_x$, then $\mathsf{utility}_{\varphi,\sigma}(\mu, C) = 0$, since $\mu(x) = 1$ and hence we fall either in case 1(b) or 2(c) above. It follows that $\mathsf{utility}_{\varphi,\sigma}(\mu)$ is the sum of the utilities of all the clauses involved with variables $x \in K$. That is:

$$\mathsf{utility}_{\varphi,\sigma}(\mu) = \sum_{x \in K} \left[ \left( \frac{k-1}{2} + \deg(x) - (k-1) \right) \mathsf{utility}_{\varphi,\sigma}(\mu, A_x) + \sum_{C \in \mathcal{B}_x} \mathsf{utility}_{\varphi,\sigma}(\mu, C) \right].$$
(2)

Take $x \in K$. Then $\mathsf{utility}_{\varphi,\sigma}(\mu, A_x) = \frac{1}{2}$ as $\mu(x) = \bot$, and then case 1(a) applies. On the other hand, for a clause $C \in \mathcal{B}_x$ we have two cases:

- $C = (x \vee \neg y)$ for $y \in K$. In this case, $\mathsf{utility}_{\varphi,\sigma}(\mu, C) = -\frac{1}{4}$ as we are in case 2(a) above.

- $C = (x \vee \neg y)$ for $y \notin K$. In this case, $\mathsf{utility}_{\varphi,\sigma}(\mu, C) = -\frac{1}{2}$ as we are in case 2(b) above.

Moreover, note that the first case occurs exactly for $k - 1$ clauses in $\mathcal{B}_x$, as $x$ has precisely $k - 1$ neighbors in the clique $K$. The second case occurs exactly for $\deg(x) - (k - 1) \geq 0$ clauses in $\mathcal{B}_x$. Replacing in equation (2), we obtain:

$$\mathsf{utility}_{\varphi,\sigma}(\mu) = \sum_{x \in K} \left( \frac{k-1}{4} + \frac{\deg(x)}{2} - \frac{k-1}{2} \right) + \left( -\frac{k-1}{4} - \frac{\deg(x)}{2} + \frac{k-1}{2} \right)$$
$$= 0.$$

We conclude that $(\varphi, \sigma)$ is a Yes-instance.

Suppose now that there is a partial assignment $\mu$, with $\mu \subsetneq \sigma$ and $\mathsf{utility}_{\varphi,\sigma}(\mu) \geq 0$. Let $K$ be the set of variables $x$ such that $\mu(x) = \bot$. For $x \notin K$ and $C = A_x$ or $C \in \mathcal{B}_x$, we have $\mathsf{utility}_{\varphi,\sigma}(\mu, C) = 0$, as we are in cases 1(b) or 2(c) above. Then we can write:

$$\mathsf{utility}_{\varphi,\sigma}(\mu) = \sum_{x \in K} \left[ \left( \frac{k-1}{2} + \deg(x) - (k-1) \right) \mathsf{utility}_{\varphi,\sigma}(\mu, A_x) + \sum_{C \in \mathcal{B}_x} \mathsf{utility}_{\varphi,\sigma}(\mu, C) \right]$$
$$+ \sum_{\{x, y\} \text{ non-edge}} 4e \left( \mathsf{utility}_{\varphi,\sigma}(\mu, Z_{x,y}) \right).$$
(3)

We claim that $|K| \geq k$. Towards a contradiction, suppose $|K| = \ell < k$. As $\mathsf{utility}_{\varphi,\sigma}(\mu, Z_{x,y}) \leq 0$ for every pair $\{x, y\}$, the last term in equation (3) is $\leq 0$, and then:

$$\mathsf{utility}_{\varphi,\sigma}(\mu) \leq \sum_{x \in K} \left[ \left( \frac{k-1}{2} + \deg(x) - (k-1) \right) \mathsf{utility}_{\varphi,\sigma}(\mu, A_x) + \sum_{C \in \mathcal{B}_x} \mathsf{utility}_{\varphi,\sigma}(\mu, C) \right].$$
(4)

Take $x \in K$. Following the same argument as before, we have that $\mathsf{utility}_{\varphi,\sigma}(\mu, A_x) = \frac{1}{2}$ and for a clause $C \in \mathcal{B}_x$ we have the two cases:

- $C = (x \vee \neg y)$ for $y \in K$, and $\mathsf{utility}_{\varphi,\sigma}(\mu, C) = -\frac{1}{4}$.

- $C = (x \vee \neg y)$ for $y \notin K$, and $\text{utility}_{\varphi,\sigma}(\mu, C) = -\frac{1}{2}$.

Let say the first case occurs precisely for $r$ clauses from $\mathcal{B}_x$. Then:

$$\sum_{C \in \mathcal{B}_x} \text{utility}_{\varphi,\sigma}(\mu, C) = -\frac{r}{4} - \frac{\deg(x) - r}{2} = \frac{r}{4} - \frac{\deg(x)}{2}. \tag{5}$$

Note that $r \leq \ell - 1$ and from equation (5) we obtain (recall $\ell < k$):

$$\sum_{C \in \mathcal{B}_x} \text{utility}_{\varphi,\sigma}(\mu, C) \leq \frac{\ell - 1}{4} - \frac{\deg(x)}{2} < \frac{k - 1}{4} - \frac{\deg(x)}{2}.$$

Replacing in equation (4), we obtain:

$$\begin{aligned}
\text{utility}_{\varphi,\sigma}(\mu) &< \sum_{x \in K} \left[ \left( \frac{k-1}{4} + \frac{\deg(x)}{2} - \frac{k-1}{2} \right) + \frac{k-1}{4} - \frac{\deg(x)}{2} \right] \\
&= \sum_{x \in K} \left[ \frac{\deg(x)}{2} - \frac{k-1}{4} + \frac{k-1}{4} - \frac{\deg(x)}{2} \right] \\
&= 0.
\end{aligned}$$

We conclude that $\text{utility}_{\varphi,\sigma}(\mu) < 0$ which is a contradiction. Hence $|K| \geq k$.

Finally, we show that $K$ is a clique. By contradiction, assume there is a pair $\{\tilde{x}, \tilde{y}\}$ such that $\tilde{x} \neq \tilde{y}$, $\tilde{x}, \tilde{y} \in K$ and $\{\tilde{x}, \tilde{y}\}$ is not an edge in $G$. Then there is a clause $Z_{\tilde{x}, \tilde{y}}$ which is repeated $M$ times in $\varphi$. Since $\mu(\tilde{x}) = \bot$ and $\mu(\tilde{y}) = \bot$, we have $\text{utility}_{\varphi,\sigma}(\mu, Z_{\tilde{x}, \tilde{y}}) = -\frac{1}{4}$, as we are in case 3(a) above. As $\text{utility}_{\varphi,\sigma}(\mu, Z_{x,y}) \leq 0$ for all pairs $\{x, y\}$, we obtain:

$$\sum_{\{x, y\} \text{ non-edge}} 4e\left(\text{utility}_{\varphi,\sigma}(\mu, Z_{x,y})\right) \leq 4e\left(\text{utility}_{\varphi,\sigma}(\mu, Z_{\tilde{x}, \tilde{y}})\right) \leq -e$$

For $x \in K$, since $\text{utility}_{\varphi,\sigma}(\mu, C) \leq 0$, for all $C \in \mathcal{B}_x$, we have $\sum_{C \in \mathcal{B}_x} \text{utility}_{\varphi,\sigma}(\mu, C) \leq 0$ and hence:

$$\sum_{x \in K} \sum_{C \in \mathcal{B}_x} \text{utility}_{\varphi,\sigma}(\mu, C) \leq 0$$

On the other hand, for $x \in K$, we have $\text{utility}_{\varphi,\sigma}(\mu, A_x) = \frac{1}{2}$. Combining all this with equation (3) we obtain:

$$\begin{aligned}
\text{utility}_{\varphi,\sigma}(\mu) &\leq \sum_{x \in K} \left( \frac{\deg(x)}{2} - \frac{k-1}{4} \right) - e \\
&< \sum_{x \in K} \frac{\deg(x)}{2} - e \qquad \text{(since } k \geq 3\text{)} \\
&\leq \sum_{x \text{ in } G} \frac{\deg(x)}{2} - e \\
&= 0.
\end{aligned}$$

We conclude that $\text{utility}_{\varphi,\sigma}(\mu) < 0$, and thus obtain a contradiction. Hence $G$ contains a clique of size $k$. $\qquad\square$

We now provide the reduction from 2-Minimal-Expected-Clauses to Check-Sub-SR, showing the hardness of the latter problem.

**Proposition 2.** *Check-Sub-SR is* NP-*hard.*

*Proof.* We reduce from 2-Minimal-Expected-Clauses. Let $(\varphi, \sigma)$ be an instance of 2-Minimal-Expected-Clauses. Let $m$ be the number of clauses of $\varphi$ and assume that $\varphi$ has $n$ variables $x_1, \ldots, x_n$. Without loss of generality we assume that $m$ is a power of 2. Define a decision tree $T$ of dimension $n + m - 1$ as follows. Start with a *perfect* binary tree $S$ of depth $\log_2 m$, that is,

each internal node has two children, and each leaf is at depth $\log_2 m$. In particular, $S$ has $m$ leaves and $m - 1$ internal nodes. All the internal nodes of $S$ are labeled with a different fresh feature from $\{n + 1, \ldots, n + m - 1\}$. For each clause $C$ in $\varphi$, pick a different leaf $\ell_C$ of $S$. It is easy to see that for each clause $C$ we can define a decision tree $S_C$ over the features $\{1, \ldots, n\}$ such that for every assignment $\mu : \{x_1, \ldots, x_n\} \to \{0, 1\}$ to the variables of $\varphi$, the corresponding instance $\boldsymbol{x} \in \{0, 1\}^n$ where $\boldsymbol{x}[i] = \mu(x_i)$ satisfies that $S_C(\boldsymbol{x}) = 1$ if and only if $\mu$ satisfies $C$. The decision tree $T$ is obtained from $S$ by identifying for each clause $C$, the leaf $\ell_C$ with the root of the decision tree $S_C$.

For any partial assignment $\mu : \{x_1, \ldots, x_n\} \to \{0, 1, \bot\}$ for $\varphi$, we denote by $\boldsymbol{y}_\mu$ the partial instance of dimension $n + m - 1$ such that $\boldsymbol{y}_\mu[i] = \mu(x_i)$ for every $i \in \{1, \ldots, n\}$ and $\boldsymbol{y}_\mu[i] = \bot$ for every $i \in \{n + 1, \ldots, n + m - 1\}$. The output of the reduction is $(T, \boldsymbol{y}_\sigma)$. Observe that the transformation from $\mu$ to $\boldsymbol{y}_\mu$ is a bijection between the sets $\{\mu \mid \mu \subseteq \sigma\}$ and $\{\boldsymbol{y}_\mu \mid \boldsymbol{y}_\mu \subseteq \boldsymbol{y}_\sigma\}$. By construction, for any partial assignment $\mu \subseteq \sigma$, we have:

$$\Pr_{\boldsymbol{z}}[T(\boldsymbol{z}) = 1 \mid \boldsymbol{z} \in \text{COMP}(\boldsymbol{y}_\mu)] = \frac{E(\varphi, \mu)}{m}.$$

Hence $(\varphi, \sigma)$ is a Yes-instance of 2-Minimal-Expected-Clauses if and only if $(T, \boldsymbol{y}_\sigma)$ is a Yes-instance of Check-Sub-SR. $\qquad\square$

**Remark 1.** *We can assume that the instance $(T, \boldsymbol{y}_\sigma)$ constructed in the proof of Proposition 2, satisfies that*

$$\Pr_{\boldsymbol{z}}[T(\boldsymbol{z}) = 1 \mid \boldsymbol{z} \in \text{COMP}(\boldsymbol{y}_\sigma)] > \frac{1}{2}.$$

Indeed, the above probability is simply $\frac{E(\varphi, \sigma)}{m}$, where $m$ is the number of clauses of $\varphi$. On the other hand, from the proof of Proposition 1, we can choose $\sigma$ such that $\sigma(x_i) = 1$ for every variable $x_i \in \{x_1, \ldots, x_n\}$ of $\varphi$. It follows that $E(\varphi, \sigma)$ is simply the number of clauses satisfied by $\sigma$, which are all the clauses in $\mathcal{B}_x$ for some variable $x$, and all the clauses of the form $Z_{x,y}$. Note that the total number of clauses from the sets $\mathcal{B}_x$ is greater that the total number clauses of the form $A_x$, and hence $\frac{E(\varphi, \sigma)}{m} > \frac{1}{2}$. Indeed, there are $\deg(x)$ clauses in $\mathcal{B}_x$, and summing over all the variables $x$, we obtain $2e$, where $e$ are the number of edges in the graph $G$. On the other hand, each clause $A_x$ is repeated $\frac{k-1}{2} + \deg(x) - (k - 1) = \deg(x) - \frac{k-1}{2}$ times. Taking the sum over all the variables $x$, we obtain $2e - n\left(\frac{k-1}{2}\right) < 2e$. This property will be useful in the Section B.2.

## B.2 From hardness of decision to hardness of computation

We will show a Turing-reduction from a variant of Check-Sub-SR to Compute-Minimal-SR, thus establishing that the latter cannot be solved in polynomial time unless $P = NP$.

For the sake of readability, given a partial instance $\boldsymbol{y}$, in this proof we use notation $\boldsymbol{z} \sim \mathbb{U}(\boldsymbol{y})$ to indicate that $\boldsymbol{z}$ is generated uniformly at random from the set $\text{COMP}(\boldsymbol{y})$. For instance, we obtain the following simplification by using this terminology:

$$\Pr_{\boldsymbol{z}}[T(\boldsymbol{z}) = 1 \mid \boldsymbol{z} \in \text{COMP}(\boldsymbol{y})] = \Pr_{\boldsymbol{z} \sim \mathbb{U}(\boldsymbol{y})}[T(\boldsymbol{z}) = 1]$$

We will require a particular kind of hard instances for the Check-Sub-SR in order to make our reduction work. In particular, we now define the notion of *strongly-balanced* inputs, which intuitively captures the idea that defined features in a partial instance $\boldsymbol{y}$ appear at the same depth in different branches of a the decision tree $T$. In order to make this definition precise, consider that every path $\pi$ from the root to a leaf in a decision tree can be identified with a sequence of labels $s_\pi$ corresponding to the labels of the nodes of $\pi$, where the last label of $\pi$ is either **true** or **false**. We use notation $s_\pi[i]$ for the $i$-th label in the sequence $s_\pi$. With this notation, we can introduce the following definition.

**Definition 2.** *Given a decision tree $T$ of dimension $d$ and $\boldsymbol{y} \in \{0, 1, \bot\}^n$ a partial instance, we say that the pair $(T, \boldsymbol{y})$ is strongly-balanced if*

$$\Pr_{\boldsymbol{z} \sim \mathbb{U}(\boldsymbol{y})}[T(\boldsymbol{z}) = 1] > \frac{1}{2},$$

*and there exists $k \in \mathbb{N}$ such that for every root-to-leaf path $\pi$ in $T$, the sequence $s_\pi$ satisfies*

$$\boldsymbol{y}[s_\pi[i]] = \bot \iff i \leq k.$$

If $(T, \boldsymbol{y})$ is strongly-balanced, then there exists a unique value $k \in \mathbb{N}$ that satisfies the second condition of the definition. We denote this value by $u(T, \boldsymbol{y})$. In particular, if $\boldsymbol{y} \in \{0,1\}^n$, then $(T, \boldsymbol{y})$ is strongly-balanced and $u(T, \boldsymbol{y}) = 0$.

Now let us define the following problem.

| | |
|---|---|
| PROBLEM : | SB-Check-SUB-SR |
| INPUT : | $(T, \boldsymbol{y})$, for $T$ a decision tree of dimension $n$ and $\boldsymbol{y} \in \{0, 1, \bot\}^n$ a partial instance, where $(T, \boldsymbol{y})$ is strongly-balanced. |
| OUTPUT : | Yes, if there is a partial instance $\boldsymbol{y}' \subsetneq \boldsymbol{y}$ such that $\Pr_{\boldsymbol{z} \sim \mathbb{U}(\boldsymbol{y}')}[T(\boldsymbol{z}) = 1] \geq \Pr_{\boldsymbol{z} \sim \mathbb{U}(\boldsymbol{y})}[T(\boldsymbol{z}) = 1]$, and No otherwise. |

One can now check that the proof of Proposition 2 directly proves NP-hardness for this problem, and thus we can reduce from it to prove hardness for the computation variant. Indeed, the first part of the definition of strongly-balanced follows from Remark 1. The second part follows from the fact that the construction in the proof of Proposition 2 starts with a perfect binary tree $S$.

**Lemma 3.** *If there is a polynomial time algorithm for* Compute-Minimal-SR, *then there is a polynomial time algorithm for* SB-Check-SUB-SR.

*Proof.* Let us enumerate the features in $T$ as $1, \ldots, n$. Also, let $S$ be the set of features defined in $\boldsymbol{y}$, that is, $\boldsymbol{y}[i] \neq \bot \iff i \in S$. We will first build a decision tree $T'$ of dimension $2n - |S|$, with the following features:

1. Create a feature $i$ for $i \in S$.

2. For every $i \in \{1, \ldots, n\} \setminus S$ create features $i$ and $i'$.

Note that this amounts to the promised number of features. We will build $T'$ through a recursive process $\mathcal{R}$ defined next. First, note that any decision tree can be described inductively as either a **true/false** leaf, or a tuple $(r, L, R)$, where $r$ is the root node, $L$ is a decision tree whose root is the left child of $r$, and $R$ is a decision tree whose root is the right child of $r$. We can now define $\mathcal{R}$ as a recursive procedure that when called with argument $\tau$ proceed as follows:

1. If $\tau$ is a leaf then simply return $\tau$.

2. If $\tau = (r, L, R)$, and node $r$ is labeled with feature $i \in S$, then simply return

$$(r, \mathcal{R}(L), \mathcal{R}(R)).$$

3. If $\tau = (r, L, R)$, and node $r$ is labeled with feature $i \in \{1, \ldots, n\} \setminus S$, then return the following decision tree:

$$(r, (u, \mathcal{R}(L), \textbf{false}), (v, \textbf{true}, \mathcal{R}(R))),$$

where nodes $u$ and $v$ are new nodes, both labeled with feature $i'$.

As anticipated, $T' = \mathcal{R}(T)$. An example illustrating the process is depicted in Figure 6. Now we will create a tree $T''$ of dimension $2n - |S| + m$ that on top of the previous features incorporates features $b_j$ for each $j \in \{1, \ldots, m\}$, where $m$ is an integer we will specify later on. In order to construct $T''$, we start by defining $\boldsymbol{y}_0$ as the partial instance of dimension $2n - |S| + m$ such that $\boldsymbol{y}_0[i] = \boldsymbol{y}[i]$ for every $i \in S$ and

$$\boldsymbol{y}_0[b_j] = 0, \quad \forall j \in \{1, \ldots, m\},$$

with the remaining components of $\boldsymbol{y}_0$ being left undefined. Let $T_{\boldsymbol{y}_0}$ be a tree of dimension $2n - |S| + m$ that accepts exactly the completions of $\boldsymbol{y}_0$; this can be trivially done by creating a tree that accepts exactly the features that are defined in $\boldsymbol{y}_0$, and then observing that when running an instance whose feature space is a superset of this, then the instance will be accepted if and only if it is a completion of $\boldsymbol{y}_0$. Now let $T_1$ be a tree of dimension $2n + |S| + m$ that implements the following Boolean formula:

$$\phi = \sum_{j=1}^{m} b_j \geq 2.$$

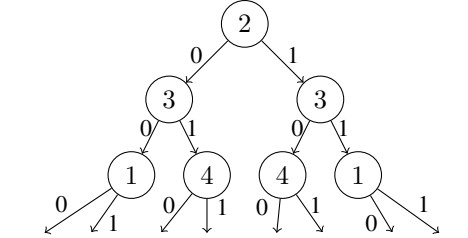

(a) Original decision tree $T$.

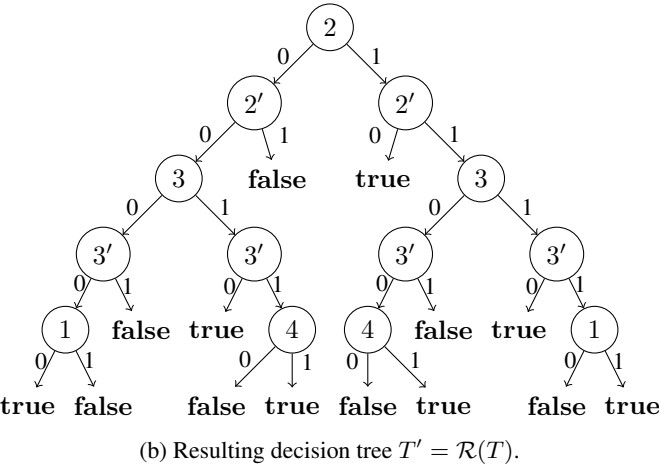

(b) Resulting decision tree $T' = \mathcal{R}(T)$.

Figure 6: Illustration of the recursive process $\mathcal{R}$ over an example where $\boldsymbol{y} = (0, \perp, \perp, 0)$. Note that the pair $(T, \boldsymbol{y})$ is strongly-balanced.

**Claim 1.** *Decision tree $T_1$, implementing the function $\phi$, can be constructed in polynomial time.*

*Proof of Claim 1.* This proof can be easily done by a direct construction. Indeed, consider the following Boolean formulas:

$$f(x_1, \ldots, x_n) := \sum_{i=1}^{n} x_i \geq 2,$$

$$g(x_1, \ldots, x_n) := \sum_{i=1}^{n} x_i \geq 1 = \bigvee_{i=1}^{n} x_i.$$

We then note that

$$f(x_1, \ldots, x_n) = [\neg x_n \wedge f(x_1, \ldots, x_{n-1})] \vee [x_n \wedge g(x_1, \ldots, x_{n-1})],$$

and thus we can build a decision tree for $f$ recursively as illustrated in Figure 7. Note that $g(x_1, \ldots, x_k)$ can be trivially implemented by a decision tree of size $O(k)$. Thus the recursive equation characterizing the size $\alpha(n)$ of a decision tree for $f(x_1, \ldots, x_n)$, is simply

$$\alpha(n) = 1 + \alpha(n-1) + O(n),$$

from where we get $\alpha(n) \in O(n^2)$, thus concluding the proof of the claim.

$\square$

Now, let us build an instance $\boldsymbol{x}$ of dimension $2n - |S| + m$ as follows. For each $i \in S$ let $\boldsymbol{x}[i] = \boldsymbol{y}[i]$, thus ensuring $\boldsymbol{x}$ will be a completion of $\boldsymbol{y}$. Then for each $j \in \{1, \ldots, m\}$ let $\boldsymbol{x}[b_j] = 0$, and finally for each $i \in \{1, \ldots, n\} \setminus S$, let $\boldsymbol{x}[i] = 0$ and $\boldsymbol{x}[i'] = 1$.

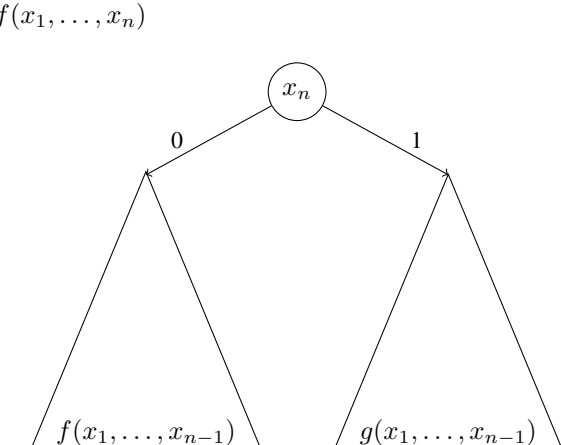

$$f(x_1, \ldots, x_n)$$

Figure 7: Illustation of the construction for Claim 1.

For example, if $\boldsymbol{y} = (0, \quad \bot, \quad \bot, \quad 1)$, and $m = 3$ then

$$\boldsymbol{x} = (0, \quad 0, \quad 1, \quad 0, \quad 1, \quad 1, \quad 0, \quad 0, \quad 0),$$

where the features $b_j$ have been placed at the end of the vector.

Let $\boldsymbol{y}^\star$ be the partial instance of dimension $2n - |S| + m$ such that $\boldsymbol{y}^\star[i] = \boldsymbol{y}[i]$ for every $i \in S$, and undefined otherwise. Let us abuse notation and assume now that $T'$ has dimension $2n - |S| + m$, even though it only explicitly uses the first $2n - |S|$ features, as this would make it compatible with other decision trees and instance we have constructed. Finally, let $T^\star$ be the decision tree defined as

$$T^\star = T_{\boldsymbol{y}_0} \cup (T' \cap T_1),$$

and note that the union and intersection of decision trees can be computed in polynomial time through a standard algorithm (see e.g., Wegener (2000)).

Let us now define

$$\delta := \Pr_{\boldsymbol{z} \sim \mathbb{U}(\boldsymbol{y}^\star)} [(T' \cap T_1)(\boldsymbol{z}) = 1].$$

We now claim that the result of Compute-Minimal-SR$(T^\star, \boldsymbol{x}, \delta)$ is different from $\boldsymbol{y}^\star$ if and only if $(T, \boldsymbol{y})$ is a positive instance of SB-Check-SUB-SR. But before we can prove this, we will need some intermediary tools and claims that we develop next.

Let us start by distinguishing two kinds of leaves of $T'$. Let us say that the leaves of $T'$ introduced in step 1 of the recursive procedure $\mathcal{R}$ are *natural*, while those introduced in step 3 are *artificial*. We denote by $\mathcal{N}$ the set of natural leaves of $T'$ and by $\mathcal{A}$ the set of artificial leaves of $T'$. Moreover, let $\mathcal{N}_t, \mathcal{N}_f$ represent the **true** and **false** natural leaves, and define $\mathcal{A}_t, \mathcal{A}_f$ analogously. We will also use $T'_{\downarrow \boldsymbol{z}}$ to denote the leaf where instance $\boldsymbol{z}$ ends when evaluated over tree $T'$. With this notation, $T'(\boldsymbol{z}) = 1$ is equivalent to $T'_{\downarrow \boldsymbol{z}} \in \mathcal{A}_t \cup \mathcal{N}_t$. We now make the following claims.

**Claim 2.** *For every partial instance $\boldsymbol{y}'^\star \subseteq \boldsymbol{y}^\star$, it holds that*

$$\Pr_{\boldsymbol{z} \sim \mathbb{U}(\boldsymbol{y}'^\star)} \left[ T'_{\downarrow \boldsymbol{z}} \in \mathcal{A}_t \mid T'_{\downarrow \boldsymbol{z}} \in \mathcal{A} \right] = \Pr_{\boldsymbol{z} \sim \mathbb{U}(\boldsymbol{y}'^\star)} \left[ T'_{\downarrow \boldsymbol{z}} \in \mathcal{A}_f \mid T'_{\downarrow \boldsymbol{z}} \in \mathcal{A} \right] = \frac{1}{2}.$$

*Proof of Claim 2.* Observe that every leaf $\ell \in \mathcal{A}$ has a parent node $v$ in $T'$ labeled with some feature $i'$ whose parent node $u$ in $T'$ is labeled with feature $i$. Let $G(\ell) = u$ be said the grandparent of $\ell$, and assume that $G^{-1}(u) = \{\ell' \mid G(\ell') = u\}$. With this notation, we can split the set $\mathcal{A}$ as follows:

$$\mathcal{A} = \bigcup_{\text{node } u \text{ with label } i \notin S} \{T'_{\downarrow \boldsymbol{z}} \mid G(T'_{\downarrow \boldsymbol{z}}) = u\},$$

where the union is actually disjoint. Thus, we have that for every partial instance $\boldsymbol{y}'^\star \subseteq \boldsymbol{y}^\star$:

$$\Pr_{\boldsymbol{z}\sim\mathbb{U}(\boldsymbol{y}'^\star)}\left[T'_{\downarrow\boldsymbol{z}} \in \mathcal{A}_t \mid T'_{\downarrow\boldsymbol{z}} \in \mathcal{A}\right] =$$

$$\sum_{\text{node } u \text{ with label } i\notin S} \Pr_{\boldsymbol{z}\sim\mathbb{U}(\boldsymbol{y}'^\star)}\left[T'_{\downarrow\boldsymbol{z}} \in \mathcal{A}_t \mid T'_{\downarrow\boldsymbol{z}} \in \mathcal{A}\cap G^{-1}(u)\right]\cdot \Pr_{\boldsymbol{z}\sim\mathbb{U}(\boldsymbol{y}'^\star)}\left[T'_{\downarrow\boldsymbol{z}} \in \mathcal{A}\cap G^{-1}(u) \mid T'_{\downarrow\boldsymbol{z}} \in \mathcal{A}\right],$$

but we have the following equation for the last term

$$\Pr_{\boldsymbol{z}\sim\mathbb{U}(\boldsymbol{y}'^\star)}\left[T'_{\downarrow\boldsymbol{z}} \in \mathcal{A}_t \mid T'_{\downarrow w} \in \mathcal{A}\cap G^{-1}(u)\right] = \frac{1}{2},$$

as the event is equivalent to $\boldsymbol{z}[i] = 1, \boldsymbol{z}[i'] = 0$, and this is equally likely to the complement event $\boldsymbol{z}[i] = 0, \boldsymbol{z}[i'] = 1$, given that $\boldsymbol{y}'^\star[i] = \boldsymbol{y}'^\star[i'] = \bot$. Therefore

$$\Pr_{\boldsymbol{z}\sim\mathbb{U}(\boldsymbol{y}'^\star)}\left[T'_{\downarrow\boldsymbol{z}} \in \mathcal{A}_t \mid T'_{\downarrow\boldsymbol{z}} \in \mathcal{A}\right] =$$

$$\sum_{\text{node } u \text{ with label } i\notin S} \frac{1}{2}\cdot \Pr_{\boldsymbol{z}\in\mathbb{U}(\boldsymbol{y}'^\star)}\left[T'_{\downarrow\boldsymbol{z}} \in \mathcal{A}\cap G^{-1}(u) \mid T'_{\downarrow\boldsymbol{z}} \in \mathcal{A}\right] =$$

$$\frac{1}{2}\cdot \sum_{\text{node } u \text{ with label } i\notin S} \Pr_{\boldsymbol{z}\in\mathbb{U}(\boldsymbol{y}'^\star)}\left[T'_{\downarrow\boldsymbol{z}} \in \mathcal{A}\cap G^{-1}(u) \mid T'_{\downarrow\boldsymbol{z}} \in \mathcal{A}\right] = \frac{1}{2}.$$

$\square$

**Claim 3.** *Given a partial instance $\boldsymbol{y}' \subseteq \boldsymbol{y}$, we can naturally define $\boldsymbol{y}'^\star$ as the partial instance of dimension $2n - |S| + m$ that matches $\boldsymbol{y}'$ on its defined features, and holds $\boldsymbol{y}'^\star[i] = \boldsymbol{y}'^\star[i'] = \bot$ for every feature $i$ such that $\boldsymbol{y}'[i] = \bot$. Then it holds that*

$$\Pr_{\boldsymbol{z}\sim\mathbb{U}(\boldsymbol{y}')}[T(\boldsymbol{z}) = 1] = \Pr_{\boldsymbol{z}\sim\mathbb{U}(\boldsymbol{y}'^\star)}\left[T'_{\downarrow\boldsymbol{z}} \in \mathcal{N}_t \mid T'_{\downarrow\boldsymbol{z}} \in \mathcal{N}\right].$$

*Proof of Claim 3.* Given that the resulting leaf is natural, for every node $u$ of $T'$ such that $u$ is labeled with feature $i \notin S$, the tuple $(u, L, R)$ was considered when constructing $T'$ and the path of $w$ in $T'$ goes through $u$, it must be the case that $\boldsymbol{z}[i] = \boldsymbol{z}[i'] = 0$ or $\boldsymbol{z}[i] = \boldsymbol{z}[i'] = 1$, as otherwise $T'_{\downarrow\boldsymbol{z}} \in \mathcal{A}$. But these two alternatives are equally likely by definition of $T'$. Thus, by using a simple induction argument, for every leaf $\ell$ of $T$ with a corresponding natural leaf $\ell'$ of $T'$, it holds that

$$\Pr_{\boldsymbol{z}\sim\mathbb{U}(\boldsymbol{y}')}[T_{\downarrow\boldsymbol{z}} = \ell] = \Pr_{\boldsymbol{z}\sim\mathbb{U}(\boldsymbol{y}'^\star)}\left[T'_{\downarrow\boldsymbol{z}} = \ell' \mid T'_{\downarrow\boldsymbol{z}} \in \mathcal{N}\right],$$

from where the claim immediately follows. $\square$

**Claim 4.** *By choosing $m \geq \max\{2u(T,\boldsymbol{y}) + 2n, 9\}$, we have that*

$$\Pr_{\boldsymbol{z}\sim\mathbb{U}(\boldsymbol{y}^\star)}[(T' \cap T_1)(\boldsymbol{z}) = 1] > \frac{1}{2}.$$

*Proof of Claim 4.* First, consider that for $T_1$, we have that

$$\Pr_{\boldsymbol{z}\sim\mathbb{U}(\boldsymbol{y}^\star)}[T_1(\boldsymbol{z}) = 1] = 1 - \left(\frac{1}{2}\right)^m - m\left(\frac{1}{2}\right)^m = 1 - (m+1)\left(\frac{1}{2}\right)^m,$$

while on the other hand

$$
\begin{aligned}
\Pr_{\boldsymbol{z}\sim\mathbb{U}(\boldsymbol{y}^\star)}[T'(\boldsymbol{z}) = 1] &= \Pr_{\boldsymbol{z}\sim\mathbb{U}(\boldsymbol{y}^\star)}\left[T'_{\downarrow\boldsymbol{z}} \in \mathcal{A}_t\right] + \Pr_{\boldsymbol{z}\sim\mathbb{U}(\boldsymbol{y}^\star)}\left[T'_{\downarrow\boldsymbol{z}} \in \mathcal{N}_t\right] \\
&= \Pr_{\boldsymbol{z}\sim\mathbb{U}(\boldsymbol{y}^\star)}\left[T'_{\downarrow\boldsymbol{z}} \in \mathcal{A}_t \mid T'_{\downarrow\boldsymbol{z}} \in \mathcal{A}\right]\cdot \Pr_{\boldsymbol{z}\sim\mathbb{U}(\boldsymbol{y}^\star)}\left[T'_{\downarrow\boldsymbol{z}} \in \mathcal{A}\right] \\
&\quad + \Pr_{\boldsymbol{z}\sim\mathbb{U}(\boldsymbol{y}^\star)}\left[T'_{\downarrow\boldsymbol{z}} \in \mathcal{N}_t \mid T'_{\downarrow\boldsymbol{z}} \in \mathcal{N}\right]\cdot \Pr_{\boldsymbol{z}\sim\mathbb{U}(\boldsymbol{y}^\star)}\left[T'_{\downarrow\boldsymbol{z}} \in \mathcal{N}\right] \\
&= \frac{1}{2}\cdot \Pr_{\boldsymbol{z}\sim\mathbb{U}(\boldsymbol{y}^\star)}\left[T'_{\downarrow\boldsymbol{z}} \in \mathcal{A}\right] \\
&\quad + \Pr_{\boldsymbol{z}\sim\mathbb{U}(\boldsymbol{y}^\star)}\left[T'_{\downarrow\boldsymbol{z}} \in \mathcal{N}_t \mid T'_{\downarrow\boldsymbol{z}} \in \mathcal{N}\right]\cdot \Pr_{\boldsymbol{z}\sim\mathbb{U}(\boldsymbol{y}^\star)}\left[T'_{\downarrow\boldsymbol{z}} \in \mathcal{N}\right],
\end{aligned}
$$

where the last equality uses Claim 2. Let us now show that

$$\Pr_{\boldsymbol{z}\sim\mathbb{U}(\boldsymbol{y}^\star)}[T'_{\downarrow\boldsymbol{z}}\in\mathcal{N}_t\mid T'_{\downarrow\boldsymbol{z}}\in\mathcal{N}]\ \geq\ \frac{1}{2}+\left(\frac{1}{2}\right)^n.$$

By Claim 3 this is the same as showing that

$$\Pr_{\boldsymbol{z}\sim\mathbb{U}(\boldsymbol{y})}[T(\boldsymbol{z})=1]\ \geq\ \frac{1}{2}+\left(\frac{1}{2}\right)^n,$$

which is guaranteed by the definition of the SB-Check-SUB-SR problem, as we know that

$$\Pr_{\boldsymbol{z}\sim\mathbb{U}(\boldsymbol{y})}[T(\boldsymbol{z})=1]\ >\ \frac{1}{2},$$

and also that $\Pr_{\boldsymbol{z}\sim\mathbb{U}(\boldsymbol{y})}[T(\boldsymbol{z})=1]$ must be of the form $\left(\frac{k}{2^n}\right)$ with $k\in\mathbb{N}$, given that $n$ is the dimension of $T$.

Now, consider that

$$\Pr_{\boldsymbol{z}\sim\mathbb{U}(\boldsymbol{y}^\star)}[T'_{\downarrow\boldsymbol{z}}\in\mathcal{A}]\ =\ 1-\left(\frac{1}{2}\right)^{u(T,\boldsymbol{y})}.$$

Notice that this holds because $T$ is strongly-balanced, so falling into a natural leaf in $T'$ requires going through $u(T,\boldsymbol{y})$ layers without choosing an artificial leaf, which happens with probability $\frac{1}{2}$ at each layer. Thus, we have that

$$\Pr_{\boldsymbol{z}\sim\mathbb{U}(\boldsymbol{y}^\star)}[T'(\boldsymbol{z})=1]\ \geq\ \frac{1}{2}\left(1-\left(\frac{1}{2}\right)^{u(T,\boldsymbol{y})}\right)+\left(\frac{1}{2}+\left(\frac{1}{2}\right)^n\right)\left(\frac{1}{2}\right)^{u(T,\boldsymbol{y})}$$

$$=\ \frac{1}{2}+\left(\frac{1}{2}\right)^{n+u(T,\boldsymbol{y})}.$$

Moreover, given that $T'$ and $T_1$ impose restrictions over disjoint sets of features, we have that

$$\Pr_{\boldsymbol{z}\sim\mathbb{U}(\boldsymbol{y}^\star)}[T'(\boldsymbol{z})=1\mid T_1(\boldsymbol{z})=1]\ =\ \Pr_{\boldsymbol{z}\sim\mathbb{U}(\boldsymbol{y}^\star)}[T'(\boldsymbol{z})=1].$$

Putting together all the previous results, we obtain that

$$\Pr_{\boldsymbol{z}\sim\mathbb{U}(\boldsymbol{y}^\star)}[(T'\cap T_1)(\boldsymbol{z})=1]=\Pr_{\boldsymbol{z}\sim\mathbb{U}(\boldsymbol{y}^\star)}[T'(\boldsymbol{z})=1\mid T_1(\boldsymbol{z})=1]\cdot\Pr_{\boldsymbol{z}\sim\mathbb{U}(\boldsymbol{y}^\star)}[T_1(\boldsymbol{z})=1]$$

$$=\Pr_{\boldsymbol{z}\sim\mathbb{U}(\boldsymbol{y}^\star)}[T'(\boldsymbol{z})=1]\cdot\Pr_{\boldsymbol{z}\sim\mathbb{U}(\boldsymbol{y}^\star)}[T_1(\boldsymbol{z})=1]$$

$$\geq\left(\frac{1}{2}+\left(\frac{1}{2}\right)^{n+u(T,\boldsymbol{y})}\right)\left(1-(m+1)\left(\frac{1}{2}\right)^m\right)$$

$$=\frac{1}{2}-(m+1)\left(\frac{1}{2}\right)^{m+1}+\left(\frac{1}{2}\right)^{n+u(T,\boldsymbol{y})}-(m+1)\left(\frac{1}{2}\right)^{m+n+u(T,\boldsymbol{y})}$$

$$\geq\frac{1}{2}-(m+1)\left(\frac{1}{2}\right)^m+\left(\frac{1}{2}\right)^{n+u(T,\boldsymbol{y})}$$

$$\geq\frac{1}{2}-\left(\frac{1}{2}\right)^{m-\lceil\log(m+1)\rceil}+\left(\frac{1}{2}\right)^{n+u(T,\boldsymbol{y})}.$$

But we are assuming $m\geq\max\{2n+2u(T,\boldsymbol{y}),9\}$, which implies that $m\geq2n+2u(T,\boldsymbol{y})$. Hence, we have that

$$m-\lceil\log(m+1)\rceil>n+u(T,\boldsymbol{y}),$$

as $m-\lceil\log(m+1)\rceil>\frac{m}{2}$ since $m\geq9$. We conclude that

$$\frac{1}{2}-\left(\frac{1}{2}\right)^{m-\lceil\log(m+1)\rceil}+\left(\frac{1}{2}\right)^{n+u(T,\boldsymbol{y})}\ >\ \frac{1}{2},$$

from which the claim follows. $\qquad\square$

**Claim 5.** *For every partial instance $\boldsymbol{y'^\star} \subseteq \boldsymbol{y^\star}$, it holds that*

$$\Pr_{\boldsymbol{z} \sim \mathbb{U}(\boldsymbol{y'^\star})}\left[T'_{\downarrow\boldsymbol{z}} \in \mathcal{A}\right] = \Pr_{\boldsymbol{z} \sim \mathbb{U}(\boldsymbol{y^\star})}\left[T'_{\downarrow\boldsymbol{z}} \in \mathcal{A}\right],$$

$$\Pr_{\boldsymbol{z} \sim \mathbb{U}(\boldsymbol{y'^\star})}\left[T'_{\downarrow\boldsymbol{z}} \in \mathcal{N}\right] = \Pr_{\boldsymbol{z} \sim \mathbb{U}(\boldsymbol{y^\star})}\left[T'_{\downarrow\boldsymbol{z}} \in \mathcal{N}\right].$$

*Proof of Claim 5.* We only need to prove that $\Pr_{\boldsymbol{z} \sim \mathbb{U}(\boldsymbol{y'^\star})}[T'_{\downarrow\boldsymbol{z}} \in \mathcal{A}] = \Pr_{\boldsymbol{z} \sim \mathbb{U}(\boldsymbol{y^\star})}[T'_{\downarrow\boldsymbol{z}} \in \mathcal{A}]$. As shown in the proof of Claim 4, this follows from the strongly-balanced property of $T$. $\qquad\square$

With these claims we can finally prove the reduction is correct. That is, we will show that Compute-Minimal-SR$(T^\star, \boldsymbol{x}, \delta)$ is different from $\boldsymbol{y^\star}$ if and only if $(T, \boldsymbol{y})$ is a positive instance of SB-Check-SUB-SR.

**Forward direction.** Assume the result of Compute-Minimal-SR$(T^\star, \boldsymbol{x}, \delta)$ is some partial instance $\boldsymbol{y'^\star}$ different from $\boldsymbol{y^\star}$. Note immediately that it is not possible that $\boldsymbol{y^\star} \subsetneq \boldsymbol{y'^\star}$ as by definition of $\delta$, we have that

$$\Pr_{\boldsymbol{z} \sim \mathbb{U}(\boldsymbol{y^\star})}[T^\star(\boldsymbol{z}) = 1] \geq \delta,$$

which would contradict the minimality of $\boldsymbol{y'^\star}$. Let us first prove that $\boldsymbol{y'^\star} \subseteq \boldsymbol{y^\star}$. As a first step, we show that $\boldsymbol{y'^\star}[i] = \boldsymbol{y'^\star}[i'] = \perp$ for every $i \notin S$. We do this by way of contradiction, assuming first that $\boldsymbol{y'^\star}[i] = 0$ or $\boldsymbol{y'^\star}[i'] = 1$ for some $i \notin S$, and exposing how either case generates a contradiction.[1]

1. If there is some feature $i \notin S$ such that $\boldsymbol{y'^\star}[i] = 0$, let us define $\boldsymbol{y^\dagger}$ to be equal to $\boldsymbol{y'^\star}$ except that $\boldsymbol{y^\dagger}[i] = \perp$. This means that $\boldsymbol{y^\dagger} \subsetneq \boldsymbol{y'^\star}$. Moreover, let $\boldsymbol{y_1^\dagger}$ be equal to $\boldsymbol{y'^\star}$ except that $\boldsymbol{y_1^\dagger}[i] = 1$. We will now show that $\boldsymbol{y^\dagger}$ is also a valid output of the computation problem, which will contradict the minimality of $\boldsymbol{y'^\star}$. Indeed, given that $(T_{\boldsymbol{y_0}} \cap T_1)(\boldsymbol{z}) = 0$ for every instance $\boldsymbol{z}$, it holds that

$$\Pr_{\boldsymbol{z} \sim \mathbb{U}(\boldsymbol{y^\dagger})}[T^\star(\boldsymbol{z}) = 1] = \Pr_{\boldsymbol{z} \sim \mathbb{U}(\boldsymbol{y^\dagger})}[T_{\boldsymbol{y_0}}(\boldsymbol{z}) = 1] + \Pr_{\boldsymbol{z} \sim \mathbb{U}(\boldsymbol{y^\dagger})}[(T' \cap T_1)(\boldsymbol{z}) = 1]$$

$$= \Pr_{\boldsymbol{z} \sim \mathbb{U}(\boldsymbol{y'^\star})}[T_{\boldsymbol{y_0}}(\boldsymbol{z}) = 1] + \Pr_{\boldsymbol{z} \sim \mathbb{U}(\boldsymbol{y^\dagger})}[(T' \cap T_1)(\boldsymbol{z}) = 1],$$

and we can then observe that the events $T'(\boldsymbol{z}) = 1$ and $T_1(\boldsymbol{z}) = 1$ are independent, thus implying that

$$\Pr_{\boldsymbol{z} \sim \mathbb{U}(\boldsymbol{y^\dagger})}[(T' \cap T_1)(\boldsymbol{z}) = 1] = \Pr_{\boldsymbol{z} \sim \mathbb{U}(\boldsymbol{y^\dagger})}[T'(\boldsymbol{z}) = 1] \cdot \Pr_{\boldsymbol{z} \sim \mathbb{U}(\boldsymbol{y^\dagger})}[T_1(\boldsymbol{z}) = 1].$$

By construction of $\boldsymbol{y^\dagger}$, we also have that

$$\Pr_{\boldsymbol{z} \sim \mathbb{U}(\boldsymbol{y^\dagger})}[T_1(\boldsymbol{z}) = 1] = \Pr_{\boldsymbol{z} \sim \mathbb{U}(\boldsymbol{y'^\star})}[T_1(\boldsymbol{z}) = 1].$$

Thus, it now suffices to show that

$$\Pr_{\boldsymbol{z} \sim \mathbb{U}(\boldsymbol{y_1^\dagger})}[T'(\boldsymbol{z}) = 1] \geq \Pr_{\boldsymbol{z} \sim \mathbb{U}(\boldsymbol{y'^\star})}[T'(\boldsymbol{z}) = 1],$$

as this implies by definition of $\boldsymbol{y'^\star}$, $\boldsymbol{y^\dagger}$ and $\boldsymbol{y_1^\dagger}$ that

$$\Pr_{\boldsymbol{z} \sim \mathbb{U}(\boldsymbol{y^\dagger})}[T'(\boldsymbol{z}) = 1] \geq \Pr_{\boldsymbol{z} \sim \mathbb{U}(\boldsymbol{y'^\star})}[T'(\boldsymbol{z}) = 1],$$

which in turn implies by the previous discussion that

$$\Pr_{\boldsymbol{z} \sim \mathbb{U}(\boldsymbol{y^\dagger})}[T^\star(\boldsymbol{z}) = 1] \geq \Pr_{\boldsymbol{z} \sim \mathbb{U}(\boldsymbol{y'^\star})}[T^\star(\boldsymbol{z}) = 1] \geq \delta,$$

and leads to a contradiction.

In order to prove that $\Pr_{\boldsymbol{z} \sim \mathbb{U}(\boldsymbol{y_1^\dagger})}[T'(\boldsymbol{z}) = 1] \geq \Pr_{\boldsymbol{z} \sim \mathbb{U}(\boldsymbol{y'^\star})}[T'(\boldsymbol{z}) = 1]$, we will consider two cases, either $\boldsymbol{y'^\star}[i'] = 1$ or $\boldsymbol{y'^\star}[i'] = \perp$.

---

[1]Recall that $\boldsymbol{x}[i] = 0$ and $\boldsymbol{x}[i'] = 1$, so if $\boldsymbol{y'^\star}[i] \neq \perp$ or $\boldsymbol{y'^\star}[i'] \neq \perp$, then $\boldsymbol{y'^\star}[i] = 0$ or $\boldsymbol{y'^\star}[i'] = 1$ as $\boldsymbol{y'^\star} \subseteq \boldsymbol{x}$.

(a) If $\boldsymbol{y'}^{\star}[i'] = 1$, then we have that

$$\Pr_{\boldsymbol{z} \sim \mathbb{U}(\boldsymbol{y}_1^{\dagger})}[T'(\boldsymbol{z}) = 1] \geq \Pr_{\boldsymbol{z} \sim \mathbb{U}(\boldsymbol{y'}^{\star})}[T'(\boldsymbol{z}) = 1],$$

as for every node $u$ in $T'$ labeled with $i$, any completion $\boldsymbol{z}$ of $\boldsymbol{y'}^{\star}$ that goes through $u$ will be rejected by construction (landing in an artificial **false** leaf), and for paths of $T'$ that do not go through a node labeled $u$ there is no difference between completions of $\boldsymbol{y'}^{\star}$ and those for $\boldsymbol{y}_1^{\dagger}$.

(b) If $\boldsymbol{y'}^{\star}[i'] = \bot$, then we have that for every node $u$ in $T'$ which corresponds to $(u, L, R)$ according to the recursive definition of a decision tree, and is labeled with $i$, the probability of acceptance of a random completion $\boldsymbol{z}$ conditioned on its path going through $u$, which is denoted by $\boldsymbol{z} \rightsquigarrow u$, is as follows:

$$\Pr_{\boldsymbol{z} \sim \mathbb{U}(\boldsymbol{y'}^{\star})}[T'(\boldsymbol{z}) = 1 \mid \boldsymbol{z} \rightsquigarrow u] = \frac{1}{2} \cdot \Pr_{\boldsymbol{z} \sim \mathbb{U}(\boldsymbol{y'}^{\star})}[L(\boldsymbol{z}) = 1 \mid \boldsymbol{z} \rightsquigarrow u],$$

while on the other hand we have

$$\Pr_{\boldsymbol{z} \sim \mathbb{U}(\boldsymbol{y}_1^{\dagger})}[T'(\boldsymbol{z}) = 1 \mid \boldsymbol{z} \rightsquigarrow u] \quad = \quad \frac{1}{2} + \frac{1}{2} \cdot \Pr_{\boldsymbol{z} \sim \mathbb{U}(\boldsymbol{y}_1^{\dagger})}[R(\boldsymbol{z}) = 1 \mid \boldsymbol{z} \rightsquigarrow u].$$

By considering that

$$\frac{1}{2} \geq \frac{1}{2} \cdot \Pr_{\boldsymbol{z} \sim \mathbb{U}(\boldsymbol{y'}^{\star})}[L(\boldsymbol{z}) = 1 \mid \boldsymbol{z} \rightsquigarrow u],$$

we conclude that

$$\Pr_{\boldsymbol{z} \sim \mathbb{U}(\boldsymbol{y}_1^{\dagger})}[T'(\boldsymbol{z}) = 1 \mid \boldsymbol{z} \rightsquigarrow u] \geq \frac{1}{2} \cdot \Pr_{\boldsymbol{z} \sim \mathbb{U}(\boldsymbol{y'}^{\star})}[L(\boldsymbol{z}) = 1 \mid \boldsymbol{z} \rightsquigarrow u].$$

from which we conclude that

$$\Pr_{\boldsymbol{z} \sim \mathbb{U}(\boldsymbol{y}_1^{\dagger})}[T'(\boldsymbol{z}) = 1 \mid \boldsymbol{z} \rightsquigarrow u] \geq \Pr_{\boldsymbol{z} \sim \mathbb{U}(\boldsymbol{y'}^{\star})}[T'(\boldsymbol{z}) = 1 \mid \boldsymbol{z} \rightsquigarrow u].$$

Therefore, by considering that $\boldsymbol{z} \rightsquigarrow u_1$ and $\boldsymbol{z} \rightsquigarrow u_2$ are disjoint events of $u_1$, $u_2$ are distinct nodes of $T'$ with labeled $i$, we have that

$$\Pr_{\boldsymbol{z} \sim \mathbb{U}(\boldsymbol{y}_1^{\dagger})}[T'(\boldsymbol{z}) = 1] = \sum_{\substack{u \text{ is a node of } T' \\ \text{with label } i}} \Pr_{\boldsymbol{z} \sim \mathbb{U}(\boldsymbol{y}_1^{\dagger})}[T'(\boldsymbol{z}) = 1 \mid \boldsymbol{z} \rightsquigarrow u] \cdot \Pr_{\boldsymbol{z} \sim \mathbb{U}(\boldsymbol{y}_1^{\dagger})}[\boldsymbol{z} \rightsquigarrow u]$$

$$\geq \sum_{\substack{u \text{ is a node of } T' \\ \text{with label } i}} \Pr_{\boldsymbol{z} \sim \mathbb{U}(\boldsymbol{y'}^{\star})}[T'(\boldsymbol{z}) = 1 \mid \boldsymbol{z} \rightsquigarrow u] \cdot \Pr_{\boldsymbol{z} \sim \mathbb{U}(\boldsymbol{y}_1^{\dagger})}[\boldsymbol{z} \rightsquigarrow u]$$

$$= \sum_{\substack{u \text{ is a node of } T' \\ \text{with label } i}} \Pr_{\boldsymbol{z} \sim \mathbb{U}(\boldsymbol{y'}^{\star})}[T'(\boldsymbol{z}) = 1 \mid \boldsymbol{z} \rightsquigarrow u] \cdot \Pr_{\boldsymbol{z} \sim \mathbb{U}(\boldsymbol{y'}^{\star})}[\boldsymbol{z} \rightsquigarrow u]$$

$$= \Pr_{\boldsymbol{z} \sim \mathbb{U}(\boldsymbol{y'}^{\star})}[T'(\boldsymbol{z}) = 1]$$

This concludes the proof of this case.

2. It remains to analyze the case when $\boldsymbol{y'}^{\star}[i] = \bot$ and $\boldsymbol{y'}^{\star}[i'] = 1$, which can be proved in the same way as the previous case $\boldsymbol{y'}^{\star}[i] = 0$ and $\boldsymbol{y'}^{\star}[i'] = \bot$.

After this case analysis, we can safely assume that $\boldsymbol{y'}^{\star}[i] = \boldsymbol{y'}^{\star}[i'] = \bot$ for every $i \notin S$. We will now show that $\boldsymbol{y'}^{\star}[b_j] = \bot$ for every $j \in \{1, \ldots, m\}$. To see this, consider that in general it could be that $\boldsymbol{y'}^{\star}$ *forces* a certain number $k$ of features $b_j$ to get value 0, meaning that there is a set $K \subseteq \{1, \ldots, m\}$ with $|K| = k$ such that $\boldsymbol{y'}^{\star}[b_j] = 0$ for $j \in K$. We will argue that $k = 0$. Let us start by arguing that $k \leq m - 2$. Indeed, assume expecting a contradiction that $k > m - 2$, then by definition

$$\Pr_{\boldsymbol{z} \sim \mathbb{U}(\boldsymbol{y'}^{\star})}[T_1(\boldsymbol{z}) = 1] = 0,$$

and thus
$$\Pr_{\boldsymbol{z}\sim\mathbb{U}(\boldsymbol{y'}^\star)}[T^\star(\boldsymbol{z})=1]=\Pr_{\boldsymbol{z}\sim\mathbb{U}(\boldsymbol{y'}^\star)}[T_{\boldsymbol{y}_0}(\boldsymbol{z})=1],$$

but
$$\Pr_{\boldsymbol{z}\sim\mathbb{U}(\boldsymbol{y'}^\star)}[T_{\boldsymbol{y}_0}(\boldsymbol{z})=1]\le\frac{1}{2},$$

as $\boldsymbol{y'}^\star$ cannot be a superset of $\boldsymbol{y}^\star$, and thus at least one feature $i$ of $\boldsymbol{y}^\star$ is undefined in $\boldsymbol{y'}^\star$, and the event $\boldsymbol{z}[i]=\boldsymbol{y}^\star[i]$, which happens with probability $\frac{1}{2}$, is required for $T_{\boldsymbol{y}_0}(\boldsymbol{z})=1$. But by definition of $\delta$, if $\boldsymbol{y'}^\star$ is the output of the computational problem, then its probability of acceptance is at least
$$\Pr_{\boldsymbol{z}\sim\mathbb{U}(\boldsymbol{y}^\star)}[(T'\cap T_1)(\boldsymbol{z})=1],$$

and this probability is greater than $\frac{1}{2}$ (Claim 4), and thus we have a contradiction. We now safely assume $k\le m-2$ and thus $m-k\ge 2$. Observe that as at least one component of $\boldsymbol{y}^\star$ is undefined in $\boldsymbol{y'}^\star$, we have
$$\Pr_{\boldsymbol{z}\sim\mathbb{U}(\boldsymbol{y'}^\star)}[T_{\boldsymbol{y}_0}(\boldsymbol{z})=1]\le\frac{1}{2}\cdot\left(\frac{1}{2}\right)^{m-k},$$

and thus
$$\Pr_{\boldsymbol{z}\sim\mathbb{U}(\boldsymbol{y'}^\star)}[(T'\cap T_1)(\boldsymbol{z})=1]\ge\delta-\frac{1}{2}\cdot\left(\frac{1}{2}\right)^{m-k},$$

which, considering that
$$\Pr_{\boldsymbol{z}\sim\mathbb{U}(\boldsymbol{y'}^\star)}[T_1(\boldsymbol{z})=1]=1-\left(\frac{1}{2}\right)^{m-k}-(m-k)\left(\frac{1}{2}\right)^{m-k},$$

implies that
$$\Pr_{\boldsymbol{y}\sim\mathbb{U}(\boldsymbol{z'}^\star)}[T'(\boldsymbol{z})=1]\ge\frac{\delta-\frac{1}{2}\cdot\left(\frac{1}{2}\right)^{m-k}}{1-\left(\frac{1}{2}\right)^{m-k}-(m-k)\left(\frac{1}{2}\right)^{m-k}},$$

as $T'(\boldsymbol{z})=1$ and $T_1(\boldsymbol{z})=1$ are independent events. We now show that the RHS of the previous equation is greater than $\delta$. Indeed, for ease of notation set $r:=(m-k+1)$ and note how the RHS can be rewritten as
$$\frac{\delta-\left(\frac{1}{2}\right)^r}{1-r\left(\frac{1}{2}\right)^{r-1}}.$$

Now consider that as $m-k\ge 2$ we have that $r\ge 3$ and thus $2^{r-1}>r$, which implies $r\left(\frac{1}{2}\right)^{r-1}<1$, and thus the denominator of the previous equation is positive, implying that what we want to show is equivalent to
$$\delta-\left(\frac{1}{2}\right)^r>\delta\left(1-r\left(\frac{1}{2}\right)^{r-1}\right),$$

which is in turn equivalent to
$$\delta r\left(\frac{1}{2}\right)^{r-1}>\left(\frac{1}{2}\right)^r,$$

but as by Claim 4 we have $\delta>\frac{1}{2}$, and $r\ge 3$, the previous equation is trivially true. We have therefore showed that
$$\Pr_{\boldsymbol{z}\sim\mathbb{U}(\boldsymbol{y'}^\star)}[T'(\boldsymbol{z})=1]>\delta.$$

Now let $\boldsymbol{y}^\ominus$ be the partial instance such that $\boldsymbol{y}^\ominus[i]=\boldsymbol{y'}^\star[i]$ for every $i\in S$, and is undefined in all other features. Note that $\boldsymbol{y}^\ominus\subseteq\boldsymbol{y'}^\star$ and also $\boldsymbol{y}^\ominus\subseteq\boldsymbol{y}^\star$. If $\boldsymbol{y}^\ominus=\boldsymbol{y'}^\star$, then $\boldsymbol{y'}^\star\subseteq\boldsymbol{y}^\star$ which is what we are hoping to prove. So we now assume $\boldsymbol{y}^\ominus\subsetneq\boldsymbol{y'}^\star$ expecting a contradiction. Note that $T'$ does not use the $b_j$ features at all, and therefore we have that
$$\Pr_{\boldsymbol{z}\sim\mathbb{U}(\boldsymbol{y}^\ominus)}[T'(\boldsymbol{z})=1]=\Pr_{\boldsymbol{z}\sim\mathbb{U}(\boldsymbol{y'}^\star)}[T'(\boldsymbol{z})=1]>\delta.$$

We will use this to prove that $y^\ominus$ would have been a valid outcome of the computing problem, thus contradicting the minimality of $y'^\star$. Indeed,

$$\Pr_{z \sim \mathbb{U}(y^\ominus)}[T^\star(z) = 1] > \Pr_{z \sim \mathbb{U}(y^\ominus)}[T'(z) = 1] \cdot \Pr_{z \sim \mathbb{U}(y^\ominus)}[T_1(z) = 1],$$

and note that as

$$\Pr_{z \sim \mathbb{U}(y^\ominus)}[T'(z) = 1] > \delta,$$

it must be the case that

$$\Pr_{z \sim \mathbb{U}(y^\ominus)}[T'(z) = 1] \geq \delta + \left(\frac{1}{2}\right)^{2n - |S|},$$

as only $2n - |S|$ features appear as labels in $T'$ and, thus, the completion probability of any partial instance over $T'$ must be an integer multiple of $\left(\frac{1}{2}\right)^{2n - |S|}$. Now let us abbreviate $2n - |S|$ as $\ell$ and choose $m \geq 2\ell$. We thus have that

$$\Pr_{z \sim \mathbb{U}(y^\ominus)}[T^\star(z) = 1] \geq \Pr_{z \sim \mathbb{U}(y^\ominus)}[T'(z) = 1] \cdot \Pr_{z \sim \mathbb{U}(y^\ominus)}[T_1(z) = 1]$$

$$\geq \left(\delta + \left(\frac{1}{2}\right)^{2n - |S|}\right)\left(1 - (m + 1)\left(\frac{1}{2}\right)^m\right)$$

$$\geq \left(\delta + \left(\frac{1}{2}\right)^{2n - |S|}\right)\left(1 - (2\ell + 1)\left(\frac{1}{2}\right)^{2\ell}\right)$$

$$= \delta - \delta(2\ell + 1)\left(\frac{1}{2}\right)^{2\ell} + \left(\frac{1}{2}\right)^\ell - (2\ell + 1)\left(\frac{1}{2}\right)^{3\ell}$$

$$\geq \delta - (2\ell + 1)\left(\frac{1}{2}\right)^{2\ell} + \left(\frac{1}{2}\right)^\ell - (2\ell + 1)\left(\frac{1}{2}\right)^{2\ell}$$

$$= \delta - (4\ell + 2)\left(\frac{1}{2}\right)^{2\ell} + \left(\frac{1}{2}\right)^\ell$$

$$= \delta + \left(\frac{1}{2}\right)^\ell\left(1 - (4\ell + 2)\left(\frac{1}{2}\right)^\ell\right),$$

where the last parenthesis is positive for $\ell \geq 5$, which can be assumed without loss of generality as otherwise the original instance of the decision problem would have constant size. We have thus concluded that

$$\Pr_{z \sim \mathbb{U}(y^\ominus)}[T^\star(z) = 1] \geq \Pr_{z \sim \mathbb{U}(y^\ominus)}[(T' \cap T_1)(z) = 1] \geq \delta, \tag{6}$$

thus showing that $y^\ominus$ is a valid outcome for the computing problem, which contradicts the minimality of $y'^\star$. This in turn implies that $y'^\star = y^\ominus$, and thus subsequently that $y'^\star \subseteq y^\star$. Let us now show how by combining Claims 2, 3 and 5, we can conclude the forward direction entirely. Indeed, note that the trivial equality

$$\Pr_{z \sim \mathbb{U}(y^\star)}[T_1(z) = 1] = \Pr_{z \sim \mathbb{U}(y'^\star)}[T_1(z) = 1]$$

implies that

$$\Pr_{z \sim \mathbb{U}(y^\star)}[T'(z) = 1] \leq \Pr_{w \sim \mathbb{U}(y'^\star)}[T'(z) = 1],$$

as we already have proved that $\Pr_{z \sim \mathbb{U}(y'^\star)}[(T' \cap T_1)(z) = 1] \geq \delta$ by equation 6 and the fact that $y^\ominus = y'^\star$, and we have that $\delta = \Pr_{z \sim \mathbb{U}(y^\star)}[(T' \cap T_1)(z) = 1]$. We can use Claims 2, 3 and 5 to

conclude that

$$
\begin{aligned}
\Pr_{\boldsymbol{z}\sim\mathbb{U}(\boldsymbol{y})}[T(\boldsymbol{z})=1] &= \Pr_{\boldsymbol{z}\sim\mathbb{U}(\boldsymbol{y}^\star)}[T'_{\downarrow\boldsymbol{z}}\in\mathcal{N}_t \mid T'_{\downarrow\boldsymbol{z}}\in\mathcal{N}] \\[4pt]
&= \frac{\Pr_{\boldsymbol{z}\sim\mathbb{U}(\boldsymbol{y}^\star)}[T'_{\downarrow\boldsymbol{z}}\in\mathcal{N}_t]}{\Pr_{\boldsymbol{z}\sim\mathbb{U}(\boldsymbol{y}^\star)}[T'_{\downarrow\boldsymbol{z}}\in\mathcal{N}]} \\[4pt]
&= \frac{\Pr_{\boldsymbol{z}\sim\mathbb{U}(\boldsymbol{y}^\star)}[T'_{\downarrow\boldsymbol{z}}\in\mathcal{N}_t]}{\Pr_{\boldsymbol{z}\sim\mathbb{U}(\boldsymbol{y}'^\star)}[T'_{\downarrow\boldsymbol{z}}\in\mathcal{N}]} \\[4pt]
&= \frac{\Pr_{\boldsymbol{z}\sim\mathbb{U}(\boldsymbol{y}^\star)}[T'(\boldsymbol{z})=1]-\Pr_{\boldsymbol{z}\sim\mathbb{U}(\boldsymbol{y}^\star)}[T'_{\downarrow\boldsymbol{z}}\in\mathcal{A}_t]}{\Pr_{\boldsymbol{z}\sim\mathbb{U}(\boldsymbol{y}'^\star)}[T'_{\downarrow\boldsymbol{z}}\in\mathcal{N}]} \\[4pt]
&= \frac{\Pr_{\boldsymbol{z}\sim\mathbb{U}(\boldsymbol{y}^\star)}[T'(\boldsymbol{z})=1]-\Pr_{\boldsymbol{z}\sim\mathbb{U}(\boldsymbol{y}^\star)}[T'_{\downarrow\boldsymbol{z}}\in\mathcal{A}_t \mid T'_{\downarrow\boldsymbol{z}}\in\mathcal{A}]\cdot\Pr_{\boldsymbol{z}\sim\mathbb{U}(\boldsymbol{y}^\star)}[T'_{\downarrow\boldsymbol{z}}\in\mathcal{A}]}{\Pr_{\boldsymbol{z}\sim\mathbb{U}(\boldsymbol{y}'^\star)}[T'_{\downarrow\boldsymbol{z}}\in\mathcal{N}]} \\[4pt]
&= \frac{\Pr_{\boldsymbol{z}\sim\mathbb{U}(\boldsymbol{y}^\star)}[T'(\boldsymbol{z})=1]-\Pr_{\boldsymbol{z}\sim\mathbb{U}(\boldsymbol{y}'^\star)}[T'_{\downarrow\boldsymbol{z}}\in\mathcal{A}_t \mid T'_{\downarrow\boldsymbol{z}}\in\mathcal{A}]\cdot\Pr_{\boldsymbol{z}\sim\mathbb{U}(\boldsymbol{y}'^\star)}[T'_{\downarrow\boldsymbol{z}}\in\mathcal{A}]}{\Pr_{\boldsymbol{z}\sim\mathbb{U}(\boldsymbol{y}'^\star)}[T'_{\downarrow\boldsymbol{z}}\in\mathcal{N}]} \\[4pt]
&\leq \frac{\Pr_{\boldsymbol{z}\sim\mathbb{U}(\boldsymbol{y}'^\star)}[T'(\boldsymbol{z})=1]-\Pr_{\boldsymbol{z}\sim\mathbb{U}(\boldsymbol{y}'^\star)}[T'_{\downarrow\boldsymbol{z}}\in\mathcal{A}_t \mid T'_{\downarrow\boldsymbol{z}}\in\mathcal{A}]\cdot\Pr_{\boldsymbol{z}\sim\mathbb{U}(\boldsymbol{y}'^\star)}[T'_{\downarrow\boldsymbol{z}}\in\mathcal{A}]}{\Pr_{\boldsymbol{z}\sim\mathbb{U}(\boldsymbol{y}'^\star)}[T'_{\downarrow\boldsymbol{z}}\in\mathcal{N}]} \\[4pt]
&= \frac{\Pr_{\boldsymbol{z}\sim\mathbb{U}(\boldsymbol{y}'^\star)}[T'(\boldsymbol{z})=1]-\Pr_{\boldsymbol{z}\sim\mathbb{U}(\boldsymbol{y}'^\star)}[T'_{\downarrow\boldsymbol{z}}\in\mathcal{A}_t]}{\Pr_{\boldsymbol{z}\sim\mathbb{U}(\boldsymbol{y}'^\star)}[T'_{\downarrow\boldsymbol{z}}\in\mathcal{N}]} \\[4pt]
&= \frac{\Pr_{\boldsymbol{z}\sim\mathbb{U}(\boldsymbol{y}'^\star)}[T'_{\downarrow\boldsymbol{z}}\in\mathcal{N}_t]}{\Pr_{\boldsymbol{z}\sim\mathbb{U}(\boldsymbol{y}'^\star)}[T'_{\downarrow\boldsymbol{z}}\in\mathcal{N}]} \\[4pt]
&= \Pr_{\boldsymbol{z}\sim\mathbb{U}(\boldsymbol{y}'^\star)}[T'_{\downarrow\boldsymbol{z}}\in\mathcal{N}_t \mid T'_{\downarrow\boldsymbol{z}}\in\mathcal{N}] \\[4pt]
&= \Pr_{\boldsymbol{z}\sim\mathbb{U}(\boldsymbol{y}')}[T(\boldsymbol{z})=1],
\end{aligned}
$$

where $\boldsymbol{y}'$ is the partial instance of dimension $n$ such that $\boldsymbol{y}'[i]=\boldsymbol{y}'^\star[i]$ for every $i$ such that $\boldsymbol{y}'^\star[i]\neq\bot$, and $\boldsymbol{y}'$ is undefined in all other features. By this definition, $\boldsymbol{y}'\subsetneq\boldsymbol{y}$ as we had $\boldsymbol{y}'^\star\subsetneq\boldsymbol{y}^\star$ (because by assumption $\boldsymbol{y}'^\star\neq\boldsymbol{y}^\star$), and thus we have effectively proved that the instance $(T,z)$ is a positive instance of SB-Check-SUB-SR. This concludes the proof of the forward direction.

**Backward direction.** Assume the instance $(T,\boldsymbol{y})$ is a positive instance of SB-Check-SUB-SR and, thus, there exists some $\boldsymbol{y}'\subsetneq\boldsymbol{y}$ such that

$$
\Pr_{\boldsymbol{z}\sim\mathbb{U}(\boldsymbol{y}')}[T(\boldsymbol{z})=1] \geq \Pr_{\boldsymbol{z}\sim\mathbb{U}(\boldsymbol{y})}[T(\boldsymbol{z})=1].
$$

Define $\boldsymbol{z}'^\star$ of the dimension of $T^\star$ based on $\boldsymbol{y}'$ by setting $\boldsymbol{y}'^\star[i]=\boldsymbol{y}'[i]$ for every $i$ such that $\boldsymbol{y}'[i]\neq\bot$, and leave the rest of $\boldsymbol{y}'^\star$ undefined. Note that this definition immediately implies $\boldsymbol{y}'^\star\subsetneq\boldsymbol{y}^\star$. By Claim 3 the previous equation implies that

$$
\Pr_{\boldsymbol{z}\sim\mathbb{U}(\boldsymbol{y}'^\star)}\big[T'_{\downarrow\boldsymbol{z}}\in\mathcal{N}_t \mid T'_{\downarrow\boldsymbol{z}}\in\mathcal{N}\big] \geq \Pr_{\boldsymbol{z}\sim\mathbb{U}(\boldsymbol{y}^\star)}\big[T'_{\downarrow\boldsymbol{z}}\in\mathcal{N}_t \mid T'_{\downarrow\boldsymbol{z}}\in\mathcal{N}\big],
$$

which implies in turn that

$$
\frac{\Pr_{\boldsymbol{z}\sim\mathbb{U}(\boldsymbol{y}'^\star)}[T'_{\downarrow\boldsymbol{z}}\in\mathcal{N}_t]}{\Pr_{\boldsymbol{z}\sim\mathbb{U}(\boldsymbol{y}'^\star)}[T'_{\downarrow\boldsymbol{z}}\in\mathcal{N}]} \geq \frac{\Pr_{\boldsymbol{z}\sim\mathbb{U}(\boldsymbol{y}^\star)}[T'_{\downarrow\boldsymbol{z}}\in\mathcal{N}_t]}{\Pr_{\boldsymbol{z}\sim\mathbb{U}(\boldsymbol{y}^\star)}[T'_{\downarrow\boldsymbol{z}}\in\mathcal{N}]}.
$$

By Claim 5 the denominators of the previous inequality are equal and, thus,

$$\Pr_{\boldsymbol{z}\sim\mathbb{U}(\boldsymbol{y}'^\star)}\big[T'_{\downarrow\boldsymbol{z}}\in\mathcal{N}_t\big]\geq\Pr_{\boldsymbol{z}\sim\mathbb{U}(\boldsymbol{y}^\star)}\big[T'_{\downarrow\boldsymbol{z}}\in\mathcal{N}_t\big],$$

from where

$$\Pr_{\boldsymbol{z}\sim\mathbb{U}(\boldsymbol{y}'^\star)}\big[T'_{\downarrow\boldsymbol{z}}\in\mathcal{N}_t\big]+\Pr_{\boldsymbol{z}\sim\mathbb{U}(\boldsymbol{y}^\star)}\big[T'_{\downarrow\boldsymbol{z}}\in\mathcal{A}_t\big]\geq$$
$$\Pr_{\boldsymbol{z}\sim\mathbb{U}(\boldsymbol{y}^\star)}\big[T'_{\downarrow\boldsymbol{z}}\in\mathcal{N}_t\big]+\Pr_{\boldsymbol{z}\sim\mathbb{U}(\boldsymbol{y}^\star)}\big[T'_{\downarrow\boldsymbol{z}}\in\mathcal{A}_t\big]=\Pr_{\boldsymbol{z}\sim\mathbb{U}(\boldsymbol{y}^\star)}[T'(\boldsymbol{z})=1].$$

But combining Claims 2 and 5 we have that

$$\Pr_{\boldsymbol{z}\sim\mathbb{U}(\boldsymbol{y}^\star)}\big[T'_{\downarrow\boldsymbol{z}}\in\mathcal{A}_t\big]=\Pr_{\boldsymbol{z}\sim\mathbb{U}(\boldsymbol{y}'^\star)}\big[T'_{\downarrow\boldsymbol{z}}\in\mathcal{A}_t\big],$$

which when combined with the previous equation gives us

$$\Pr_{\boldsymbol{z}\sim\mathbb{U}(\boldsymbol{y}'^\star)}[T'(\boldsymbol{z})=1]\geq\Pr_{\boldsymbol{z}\sim\mathbb{U}(\boldsymbol{y}^\star)}[T'(\boldsymbol{z})=1],$$

and using again that

$$\Pr_{\boldsymbol{z}\sim\mathbb{U}(\boldsymbol{y}'^\star)}[T_1(\boldsymbol{z})=1]=\Pr_{\boldsymbol{z}\sim\mathbb{U}(\boldsymbol{y}^\star)}[T_1(\boldsymbol{z})=1],$$

we obtain that

$$\Pr_{\boldsymbol{z}\sim\mathbb{U}(\boldsymbol{y}'^\star)}[(T'\cap T_1)(\boldsymbol{z})=1]\geq\Pr_{\boldsymbol{z}\sim\mathbb{U}(\boldsymbol{y}^\star)}[(T'\cap T_1)(\boldsymbol{z})=1]=\delta.$$

Finally, by observing that

$$\Pr_{\boldsymbol{z}\sim\mathbb{U}(\boldsymbol{y}'^\star)}[T^\star(\boldsymbol{z})=1]\geq\Pr_{\boldsymbol{z}\sim\mathbb{U}(\boldsymbol{y}'^\star)}[(T'\cap T_1)(\boldsymbol{z})=1]\geq\delta,$$

we have that $\boldsymbol{y}'^\star$ is a valid output for the computational problem, and give it is a strict subset of $\boldsymbol{y}^\star$, the result of Compute-Minimal-SR$(T^\star,\boldsymbol{x},\delta)$ cannot be equal to $\boldsymbol{y}^\star$. This concludes the backward direction, and with it the entire proof is complete. $\qquad\square$

## C   Proof of Theorem 4

**Theorem 4.** *Let $c\geq 1$ be a fixed integer. Both* Compute-Minimum-SR *and* Compute-Minimal-SR *can be solved in polynomial time for decision trees with split number at most $c$.*

*Proof.* It suffices to provide a polynomial time algorithm for Compute-Minimum-SR. (The same algorithm works for Compute-Minimal-SR as a minimum $\delta$-SR is in particular minimal.) In turn, using standard arguments, it is enough to provide a polynomial time algorithm for the following decision problem Check-Minimum-SR: Given a tuple $(T,\boldsymbol{y},\delta,k)$, where $T$ is a decision tree of dimension $n$, $\boldsymbol{y}\in\{0,1,\perp\}^n$ is a partial instance, $\delta\in(0,1]$, and $k\geq 0$ is an integer, decide whether there is a partial instance $\boldsymbol{y}'\subseteq\boldsymbol{y}$ such that $n-|\boldsymbol{y}'|_\perp\leq k$ (i.e., $\boldsymbol{y}$ has at most $k$ defined components) and $\Pr_{\boldsymbol{z}}[T(\boldsymbol{z})=1\mid\boldsymbol{z}\in\textsc{Comp}(\boldsymbol{y}')]\geq\delta$.

In order to solve Check-Minimum-SR over an instance $(T,\boldsymbol{y},\delta,k)$, where $T$ has split number at most $c$, we apply dynamic programming over $T$ in a bottom-up manner. Let $Z\subseteq\{1,\ldots,n\}$ be the set of features defined in $\boldsymbol{y}$, that is, features $i$ with $\boldsymbol{y}[i]\neq\perp$. Those are the features we could eventually remove when looking for $\boldsymbol{y}'$. For each node $u$ in $T$, we solve a polynomial number of subproblems over the subtree $T_u$. We define

$$\mathsf{Int}(u):=\mathcal{F}\big(N_u^\downarrow\big)\cap\mathcal{F}\big(N_u^\uparrow\big)\cap Z\qquad\qquad\mathsf{New}(u):=\big(\mathcal{F}\big(N_u^\downarrow\big)\setminus\mathsf{Int}(u)\big)\cap Z.$$

In other words, $\mathsf{Int}(u)$ are the features appearing both inside and outside $T_u$, while $\mathsf{New}(u)$ are the features only inside $T_u$, that is, the new features introduced below $u$. Both sets are restricted to $Z$ as features not in $Z$ play no role in the process.

Each particular subproblem is indexed by a possible size $s\in\{0,\ldots,k\}$ and a possible set $J\subseteq\mathsf{Int}(u)$ with $|J|\leq s$ and the goal is to compute the quantity:

$$p_{u,s,J}:=\max_{\boldsymbol{y}'\in\mathcal{C}_{u,s,J}}\Pr_{\boldsymbol{z}}[T_u(\boldsymbol{z})=1\mid\boldsymbol{z}\in\textsc{Comp}(\boldsymbol{y}')],$$

where $\mathcal{C}_{u,s,J}$ is the space of partial instances $\boldsymbol{y}' \subseteq \boldsymbol{y}$ with $n - |\boldsymbol{y}'|_\perp \leq s$ and such that $\boldsymbol{y}'[i] = \boldsymbol{y}[i]$ for $i \in J$ and $\boldsymbol{y}'[i] = \perp$ for $i \in \mathsf{Int}(u) \setminus J$. In other words, the set $J$ fixes the behavior on $\mathsf{Int}(u)$ (keep features in $J$, remove features in $\mathsf{Int}(u) \setminus J$) and hence the maximization occurs over choices on the set $\mathsf{New}(u)$ (which features are kept and which features are removed). The key idea is that $p_{u,s,J}$ can be computed inductively using the information already computed for the children $u_1$ and $u_2$ of $u$. Intuitively, this holds since the common features between $T_{u_1}$ and $T_{u_2}$ are at most $c$, which is a fixed constant, and hence we can efficiently synchronize the information stored for $u_1$ and $u_2$. Finally, to solve the instance $(T, \boldsymbol{y}, \delta, k)$ we simply check whether $p_{r,k,\emptyset} \geq \delta$, for the root $r$ of $T$.

Formally, let us define for a set $H \subseteq Z$, the partial instance $\boldsymbol{y}_H \in \{0, 1, \perp\}^n$ such that $\boldsymbol{y}_H[i] = \boldsymbol{y}[i]$ for every $i \in H$, and $\boldsymbol{y}_H[i] = \perp$ for every $i \notin H$. In particular, $\boldsymbol{y}_H \subseteq \boldsymbol{y}$. Then we can write $p_{u,s,J}$ as

$$p_{u,s,J} = \max_{\substack{K \subseteq \mathsf{New}(u) \\ |K| \leq s - |J|}} \Pr_{\boldsymbol{z}}[T_u(\boldsymbol{z}) = 1 \mid \boldsymbol{z} \in \textsc{Comp}(\boldsymbol{y}_{J \cup K})].$$

Let $u_1$ and $u_2$ be the children of $u$. We have that $\mathsf{New}(u)$ is the disjoint union of:

$$\mathsf{New}(u) = \mathsf{New}(u_1) \cup \mathsf{New}(u_2) \cup \mathsf{Sync}(u),$$

where $\mathsf{Sync}(u) := \mathsf{New}(u) \cap \left( \mathcal{F}\left( N_{u_1}^\downarrow \right) \cap \mathcal{F}\left( N_{u_2}^\downarrow \right) \right)$. In other words, the features in $\mathsf{Sync}(u)$ are the features that are in both $T_{u_1}$ and $T_{u_2}$ but not outside $T_u$. We conclude by explaining the computation of $p_{u,s,J}$. We consider the following cases:

1. The feature $i$ labeling $u$ is in $J$. This means we have to keep feature $i$. If $\boldsymbol{y}[i] = 0$, then to compute $p_{u,s,J}$ we can simply look at $u_1$ (the left child). Note that $\mathsf{Int}(u_1)$ is the disjoint union of $\mathsf{Int}(u_1) \cap \mathsf{Int}(u)$ and $\mathsf{Sync}(u)$. Then

$$p_{u,s,J} = \max_{\substack{J' \subseteq \mathsf{Sync}(u) \\ |J'| \leq s - |\mathsf{Int}(u_1) \cap J|}} p_{u_1, s, (\mathsf{Int}(u_1) \cap J) \cup J'}.$$

This computation can be done in polynomial time as $\mathsf{Sync}(u) \leq c$ and then there are a constant number of possible $J' \subseteq \mathsf{Sync}(u)$. The case when $\boldsymbol{y}[i] = 1$ is analogous, taking $u_2$ instead of $u_1$.

2. The feature $i$ labeling $u$ is either outside $Z$ or belongs to $\mathsf{Int}(u) \setminus J$. This means feature $i$ is undefined. Again, we have that $\mathsf{Int}(u_1)$ is the disjoint union of $\mathsf{Int}(u_1) \cap \mathsf{Int}(u)$ and $\mathsf{Sync}(u)$. Similarly, $\mathsf{Int}(u_2)$ is the disjoint union of $\mathsf{Int}(u_2) \cap \mathsf{Int}(u)$ and $\mathsf{Sync}(u)$. Then

$$p_{u,s,J} = \max_{\substack{J' \subseteq \mathsf{Sync}(u) \\ |J'| \leq s - |J|}} \max_{\substack{0 \leq s_1, s_2 \leq s \\ s_1 + s_2 \leq s - |J| - |J'|}} \frac{1}{2} \cdot p_{u_1, s_1, (\mathsf{Int}(u_1) \cap J) \cup J'} + \frac{1}{2} \cdot p_{u_2, s_2, (\mathsf{Int}(u_2) \cap J) \cup J'}.$$

Again, this can be done in polynomial time as $\mathsf{Sync}(u) \leq c$.

3. Finally, the remaining case is that the feature $i$ labeling $u$ is in $\mathsf{New}(u)$. In that case we have the two possibilities: either we keep feature $i$ or we remove it. If $s - |J| = 0$, then the only possible choice is to remove the feature $i$, and hence $p_{u,s,J}$ is computed exactly as in case (2). If $s - |J| > 0$. Then we take the maximum between the cases when we keep feature $i$ and the case when we remove feature $i$. For the latter, $p_{u,s,J}$ is computed exactly as in case (2). For the former, we compute $p_{u,s,J}$ in a similar way as in case (1). More precisely, if $\boldsymbol{y}[i] = 0$, then:

$$p_{u,s,J} = \max_{\substack{J' \subseteq \mathsf{Sync}(u) \\ |J'| \leq s - 1 - |\mathsf{Int}(u_1) \cap J|}} p_{u_1, s-1, (\mathsf{Int}(u_1) \cap J) \cup J'}.$$

The case $\boldsymbol{y}[i] = 1$ is analogous.

$\square$

# D   Proof of Lemma 1

**Lemma 1.** *Let $\mathfrak{C}$ be a class of monotone models, $\mathcal{M} \in \mathfrak{C}$ a model of dimension $n$, and $\boldsymbol{x} \in \{0,1\}^n$ an instance. Consider any $\delta \in (0,1]$. Then if $\boldsymbol{y} \subseteq \boldsymbol{x}$ is a $\delta$-SR for $\boldsymbol{x}$ under $\mathcal{M}$ which is not minimal, then there is a partial instance $\boldsymbol{y}' := \boldsymbol{y} \setminus \{i\}$, for some $i \in \{1, \ldots, n\}$, such that $\boldsymbol{y}'$ is a $\delta$-SR for $\boldsymbol{x}$ under $\mathcal{M}$.*

*Proof.* Note that, if $\mathcal{M}(\boldsymbol{x}) = 1$ then we can safely assume that for every $i$ where $\boldsymbol{y}[i] \neq \bot$ it holds that $\boldsymbol{y}[i] = 1$, as otherwise if $\boldsymbol{y}[i^\star] = 0$ for some $i^\star$, then the lemma trivially holds by setting $\boldsymbol{y}' = \boldsymbol{y} \setminus \{i^\star\}$ because of monotonicity. Similarly, if $\mathcal{M}(\boldsymbol{x}) = 0$ then we can safely assume that for every $i$ where $\boldsymbol{y}[i] \neq \bot$ it holds that $\boldsymbol{y}[i] = 0$.

As by hypothesis $\boldsymbol{y}$ is not minimal, there exists a $\delta$-SR $\boldsymbol{y}^\star \subsetneq \boldsymbol{y}$ that minimizes $|\boldsymbol{y}^\star|_\bot$. We will prove that $|\boldsymbol{y}^\star|_\bot = |\boldsymbol{y}|_\bot + 1$, from where the lemma immediately follows.

Assume for the sake of a contradiction that $|\boldsymbol{y}^\star|_\bot > |\boldsymbol{y}|_\bot + 1$. Then, there must exist a feature $i^\star$ that $\boldsymbol{y}^\star[i^\star] = \bot \neq \boldsymbol{y}[i^\star]$, and such that $\boldsymbol{y}^\star \cup \{i^\star\} \neq \boldsymbol{y}$, where $\boldsymbol{y}^\star \cup \{i^\star\}$ is defined as

$$(\boldsymbol{y}^\star \cup \{i^\star\})[i] = \begin{cases} \boldsymbol{y}[i^\star] & \text{if } i = i^\star \\ \boldsymbol{y}^\star[i^\star] & \text{otherwise.} \end{cases}$$

Similarly we denote $\boldsymbol{y}^\star \cup (i^\star \to \alpha)$, with $\alpha \in \{0,1\}$, the partial instance defined as

$$(\boldsymbol{y}^\star \cup (i^\star \to \alpha))[i] = \begin{cases} \alpha & \text{if } i = i^\star \\ \boldsymbol{y}^\star[i^\star] & \text{otherwise.} \end{cases}$$

We now claim that $\boldsymbol{y}^\star \cup \{i^\star\}$ is also a $\delta$-SR for $\boldsymbol{x}$ under $\mathcal{M}$, which will contradict the minimality of $\boldsymbol{y}^\star$, as $|\boldsymbol{y}^\star \cup \{i^\star\}|_\bot < |\boldsymbol{y}^\star|$. Let us denote by $C(\mathcal{M}, \boldsymbol{y})$ the number of completions $\boldsymbol{z} \in \text{COMP}(\boldsymbol{y})$ such that $\mathcal{M}(\boldsymbol{z}) = 1$. Now there are two cases, if $\mathcal{M}(\boldsymbol{x}) = 1$ then

$$\begin{aligned} C(\mathcal{M}, \boldsymbol{y}^\star) &= C(\mathcal{M}, \boldsymbol{y}^\star \cup (i^\star \to 0)) + C(\mathcal{M}, \boldsymbol{y}^\star \cup (i^\star \to 1)) \\ &\leq 2C(\mathcal{M}, \boldsymbol{y}^\star \cup (i^\star \to 1)), \qquad \text{(Because of monotonicty)} \end{aligned}$$

from where

$$\frac{C(\mathcal{M}, \boldsymbol{y}^\star \cup (i^\star \to 1))}{2^{|\boldsymbol{y}^\star \cup (i^\star \to 1)|_\bot}} = \frac{C(\mathcal{M}, \boldsymbol{y}^\star \cup (i^\star \to 1))}{2^{|\boldsymbol{y}^\star|_\bot - 1}} \geq \frac{C(\mathcal{M}, \boldsymbol{y}^\star)}{2 \cdot 2^{|\boldsymbol{y}^\star|_\bot - 1}} \geq \delta,$$

which implies that $\boldsymbol{y}^\star \cup (i^\star \to 1)$ is also a $\delta$-SR (note that $\boldsymbol{y}^\star \cup (i^\star \to 1) = \boldsymbol{y}^\star \cup \{i^\star\}$ because of the initial observation), contradicting the minimality of $\boldsymbol{y}^\star$. Similarly, if $\mathcal{M}(\boldsymbol{x}) = 0$, then

$$\begin{aligned} C(\mathcal{M}, \boldsymbol{y}^\star) &= C(\mathcal{M}, \boldsymbol{y}^\star \cup (i^\star \to 0)) + C(\mathcal{M}, \boldsymbol{y}^\star \cup (i^\star \to 1)) \\ &\geq 2C(\mathcal{M}, \boldsymbol{y}^\star \cup (i^\star \to 0)), \qquad \text{(Because of monotonicty)} \end{aligned}$$

from where

$$\begin{aligned} \frac{2^{|\boldsymbol{y}^\star \cup (i^\star \to 0)|_\bot} - C(\mathcal{M}, \boldsymbol{y}^\star \cup (i^\star \to 0))}{2^{|\boldsymbol{y}^\star \cup (i^\star \to 0)|_\bot}} &= \frac{2^{|\boldsymbol{y}^\star|_\bot - 1} - C(\mathcal{M}, \boldsymbol{y}^\star \cup (i^\star \to 0))}{2^{|\boldsymbol{y}^\star|_\bot - 1}} \\ &\geq 1 - \frac{C(\mathcal{M}, \boldsymbol{y}^\star)}{2 \cdot 2^{|\boldsymbol{y}^\star|_\bot - 1}} \geq \delta, \end{aligned}$$

thus implying that $\boldsymbol{y}^\star \cup (i^\star \to 0)$ is also a $\delta$-SR for $\boldsymbol{x}$ under $\mathcal{M}$, which again contradicts the minimality of $\boldsymbol{y}^\star$.

$\square$

# E   Experiments

This section presents some experimental results both for the deterministic encoding ($\delta = 1$) and for the general probabilistic encoding ($\delta < 1$).

**Datasets** For testing the deterministic encoding we use the classical MNIST dataset (Deng, 2012), binarizing features by simply setting to black all pixels of value less than 128. For testing the general probabilistic encoding we build a dataset of 5x5 images that are either *tall* or *wide*, and the task is to predict the kind of a given rectangle. This idea is based on the dataset built by Choi et al. (2017) for illustrating sufficient reasons.[2]

**Training decision trees** We use `scikit-learn` (Pedregosa et al., 2011) to train decision trees. In order to accelerate the training, the `splitter` parameter is set to `random`. Also, due to the natural class unbalance on the MNIST dataset, we set the parameter `class_weight` to `balanced`.

**Hardware** All our experiments have been run on a personal computer with the following specifications: MacBook Pro (13-inch, M1, 2020), Apple M1 processor, 16 GB of RAM.

**Solver** We use *CaDiCaL* (Biere et al., 2020), a standard CDCL based solver. In order to find $k^\star$, the minimum $k$ for which an explanation of size $k$ exists one can either proceed by using a MaxSAT solver, directly to minimize the number of features used in the explanation, or use a standard SAT solver and do a search over $k$ to find the minimum size for which an explanation exists. After testing both approaches we use the latter as it showed to be more efficient in most cases. Instead of using binary search to find the $k^\star$, we use doubling search. This is because a single instance with $k = \frac{n}{2}$ at the start of a binary search can dominate the complexity, and often $k^\star \ll n$.

**Deterministic results** Given the compactness of the deterministic encoding, with $O(nk + |T|)$ clauses, it is feasible to use it for MNIST instances, for which $n = 28 \times 28 = 784$. Tables 2 and 3 exhibit results obtained for this dataset when recognizing digit 1. Figures 11 and 12 exhibit minimum sufficient reasons for positive and negative instance (respectively) on a decision tree for recognizing the digit 1. Figures 13 and 14 show examples when recognizing the digit 3, and finally Figures 15 and 16 exhibit examples when recognizing the digit 9. Figure 10 shows empirically how time scales linearly with $k^\star$, the size of the minimum sufficient reason found.

**Probabilistic results** Because of the complexity of the encoding, we test over the synthetic dataset described above in which the dimension is only $5 \times 5 = 25$. The positive class corresponds to *wide* rectangles. Table 1 summarizes the results obtained for this dataset, while Figure 17 shows examples. We emphasize the following observations:

1. Computing probabilistic sufficient reasons through the general probabilistic encoding (i.e., $\delta < 1$) is less efficient than computing deterministic ones, even by several orders of magnitude.

2. As the value of $\delta$ approaches one, the size of the minimum $\delta$-SR approaches $k^\star$, the size of the minimum sufficient reason for the given instance. On the other hand, as $\delta$ decreases the size of the minimum $\delta$-SR goes to 0. This trade-off implies that $\delta$ can be used to control the size of the obtained explanation.

3. The time per explanation increases significantly as $\delta$ approaches 1, even though the encoding itself does not get any larger, implying that the resulting CNF is more challenging. Interestingly enough, for $\delta = 1$ the deterministic encoding is very efficient, thus suggesting a discontinuity. It remains a challenging problem to compute $\delta$-SRs for $\delta < 1$ in a way that at least matches the efficiency of the case $\delta = 1$.

---

[2]Under the name of PI-explanations. To the best of our knowledge their dataset is not published and thus we recreated it.

Table 1: Experimental results for the probabilistic encoding over positive instances of tall rectangles. Each datapoint is the average of 3 instances.

| $\delta$ | Size of smallest explanation | Time | Number of leaves | Accuracy of the tree |
|---|---|---|---|---|
| 0.6 | 0.0 | 0.105s | 20 | 0.854 |
| 0.7 | 1.0 | 0.362s | 20 | 0.854 |
| 0.8 | 2.0 | 0.669s | 20 | 0.854 |
| 0.9 | 3.0 | 1.551s | 20 | 0.854 |
| 0.95 | 3.0 | 1.409s | 20 | 0.854 |
| 1.0 | 3.0 | 0.032s | 20 | 0.854 |
| 0.6 | 0.0 | 0.183s | 30 | 0.965 |
| 0.7 | 1.0 | 0.567s | 30 | 0.965 |
| 0.8 | 1.0 | 0.578s | 30 | 0.965 |
| 0.9 | 3.67 | 5.377s | 30 | 0.965 |
| 0.95 | 5.67 | 12.27s | 30 | 0.965 |
| 1.0 | 6.67 | 0.037s | 30 | 0.965 |
| 0.6 | 2.0 | 1.003s | 40 | 0.986 |
| 0.7 | 3.0 | 2.259s | 40 | 0.986 |
| 0.8 | 4.0 | 3.507s | 40 | 0.986 |
| 0.9 | 6.0 | 10.318s | 40 | 0.986 |
| 0.95 | 7.0 | 16.387s | 40 | 0.986 |
| 1.0 | 10.0 | 0.046s | 40 | 0.986 |
| 0.6 | 2.67 | 1.992s | 50 | 1.0 |
| 0.7 | 4.33 | 6.111s | 50 | 1.0 |
| 0.8 | 5.0 | 7.216s | 50 | 1.0 |
| 0.9 | 7.0 | 26.58s | 50 | 1.0 |
| 0.95 | 7.67 | 33.129s | 50 | 1.0 |
| 1.0 | 8.33 | 0.044s | 50 | 1.0 |

Table 2: Experimental results for the deterministic encoding over negative instances of digit 1 in MNIST. Each datapoint corresponds to the average of 10 instances.

| Number of leaves | Size of smallest explanation | Time | Accuracy of the tree |
|---|---|---|---|
| 100 | 3.6 | 0.17s | 0.988 |
| 125 | 3.1 | 0.141s | 0.989 |
| 150 | 3.7 | 0.196s | 0.989 |
| 175 | 4.1 | 0.216s | 0.99 |
| 200 | 4.3 | 0.255s | 0.991 |
| 225 | 3.9 | 0.269s | 0.991 |
| 250 | 4.1 | 0.304s | 0.992 |
| 275 | 4.1 | 0.334s | 0.992 |
| 300 | 4.4 | 0.329s | 0.993 |
| 325 | 4.0 | 0.292s | 0.993 |
| 350 | 4.3 | 0.327s | 0.993 |
| 375 | 4.0 | 0.283s | 0.993 |
| 400 | 5.2 | 0.337s | 0.993 |
| 425 | 6.0 | 0.346s | 0.993 |
| 450 | 6.8 | 0.495s | 0.993 |
| 475 | 6.4 | 0.401s | 0.993 |
| 500 | 5.8 | 0.497s | 0.993 |

Table 3: Experimental results for the deterministic encoding over positive instances of digit 1 in MNIST. Each datapoint corresponds to the average of 10 instances.

| Number of leaves | Size of smallest explanation | Time | Accuracy of the tree |
|---|---|---|---|
| 100 | 17.0 | 1.238 | 0.988 |
| 125 | 17.5 | 1.077 | 0.989 |
| 150 | 18.4 | 1.058 | 0.989 |
| 175 | 17.9 | 1.082 | 0.99 |
| 200 | 19.0 | 1.001 | 0.991 |
| 225 | 21.4 | 1.188 | 0.991 |
| 250 | 25.0 | 1.405 | 0.992 |
| 275 | 23.7 | 1.183 | 0.992 |
| 300 | 31.2 | 1.603 | 0.993 |
| 325 | 31.3 | 1.504 | 0.993 |
| 350 | 28.5 | 1.365 | 0.993 |
| 375 | 30.9 | 1.547 | 0.993 |
| 400 | 32.1 | 2.429 | 0.993 |
| 425 | 34.3 | 2.188 | 0.993 |
| 450 | 34.3 | 2.115 | 0.993 |
| 475 | 42.5 | 2.533 | 0.993 |
| 500 | 43.6 | 2.614 | 0.993 |

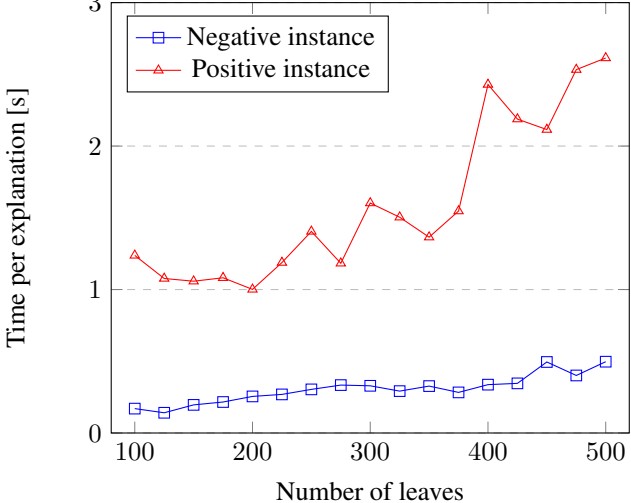

Figure 8: Time for computing a minimum sufficient reason ($\delta = 1$) as a function of decision tree size. All datapoints correspond to an average of 10 different instances for decision trees trained to recognize the digit 1 in the MNIST dataset.

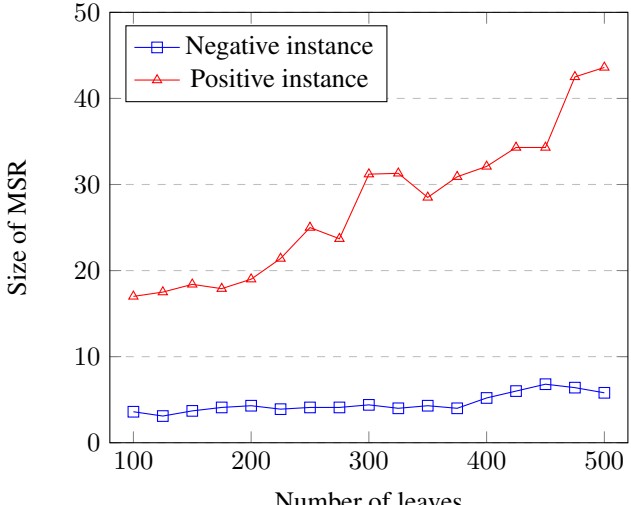

Figure 9: Size of minimum sufficient reasons ($\delta = 1$) as a function of decision tree size. All datapoints correspond to an average of 10 different instances for decision trees trained to recognize the digit 1 in the MNIST dataset.

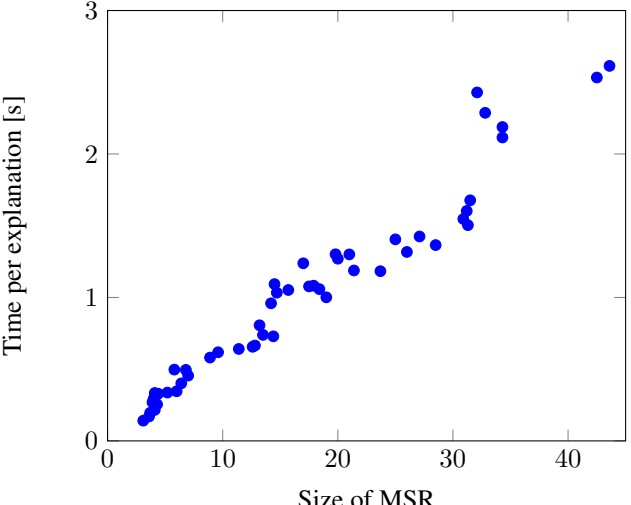

Figure 10: Relationship between the size of the Minimum Sufficient Reason and the time it takes to obtain it.

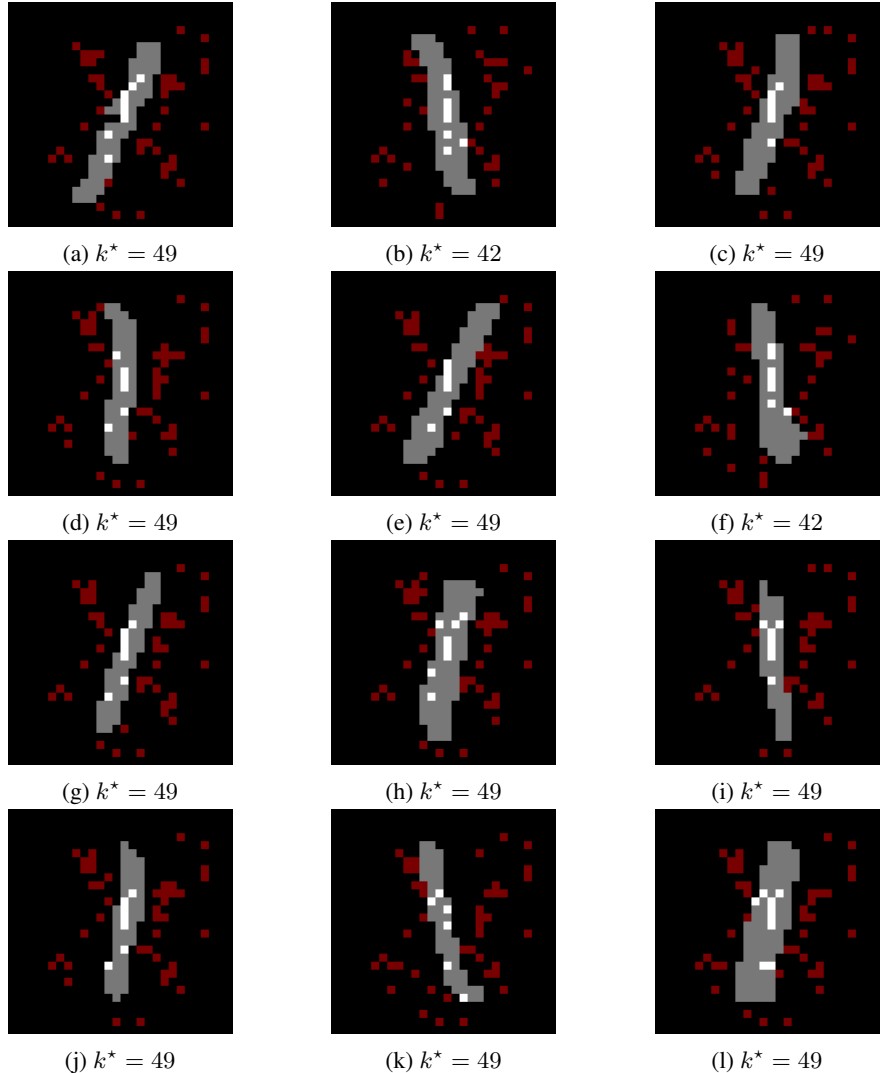

(a) $k^\star = 49$    (b) $k^\star = 42$    (c) $k^\star = 49$

(d) $k^\star = 49$    (e) $k^\star = 49$    (f) $k^\star = 42$

(g) $k^\star = 49$    (h) $k^\star = 49$    (i) $k^\star = 49$

(j) $k^\star = 49$    (k) $k^\star = 49$    (l) $k^\star = 49$

Figure 11: Examples of *Minimum Sufficient Reasons* over the MNIST dataset. All instances are (correctly predicted) positive instances for a decision tree of 591 leaves that detects the digit 1. Light pixels of the original image are depicted in grey, and the light pixels of the original image that are part of the minimum sufficient reason are colored white. Dark pixels that are part of the minimum sufficient reason are colored with red. Individual captions denote the size of the minimum sufficient reasons with $k^\star$.

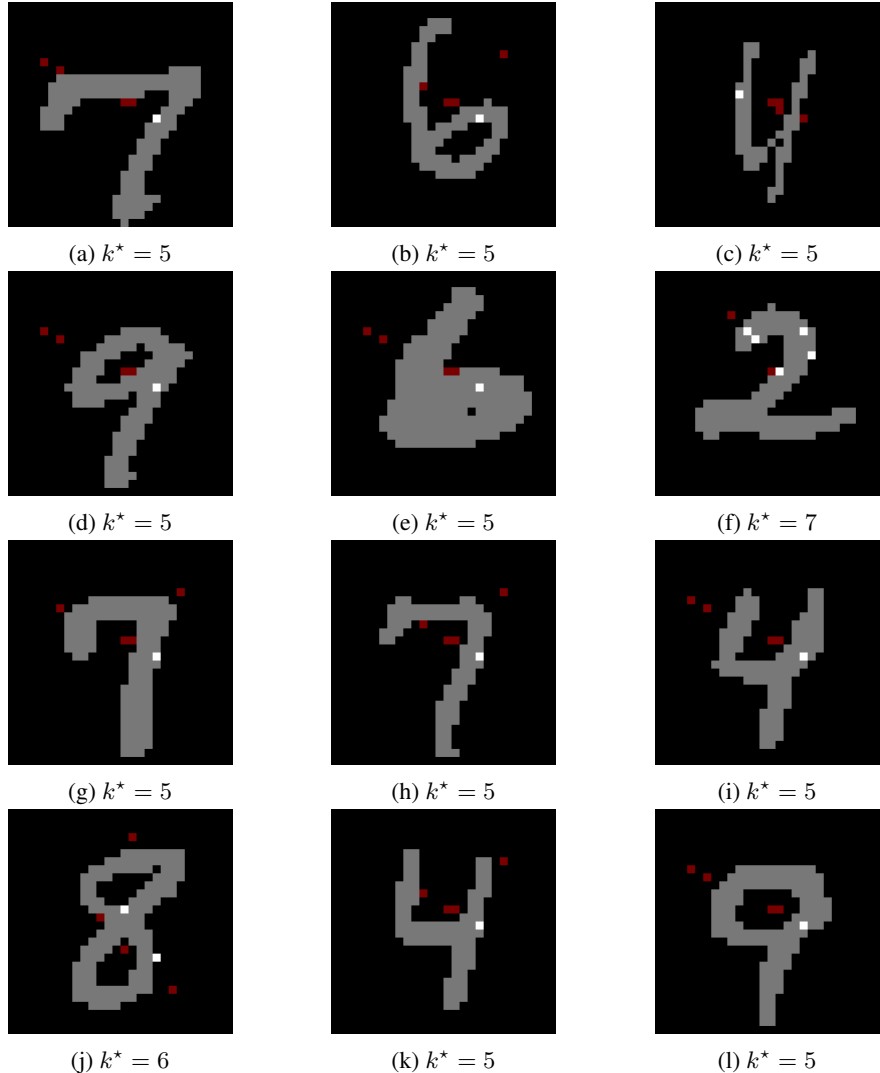

Figure 12: Examples of *Minimum Sufficient Reasons* over the MNIST dataset. All instances are correctly predicted negative instances for a decision tree of 591 leaves that detects the digit 1. Light pixels of the original image are depicted in grey, and the light pixels of the original image that are part of the minimum sufficient reason are colored white. Dark pixels that are part of the minimum sufficient reason are colored with red. Individual captions denote the size of the minimum sufficient reasons with $k^\star$.

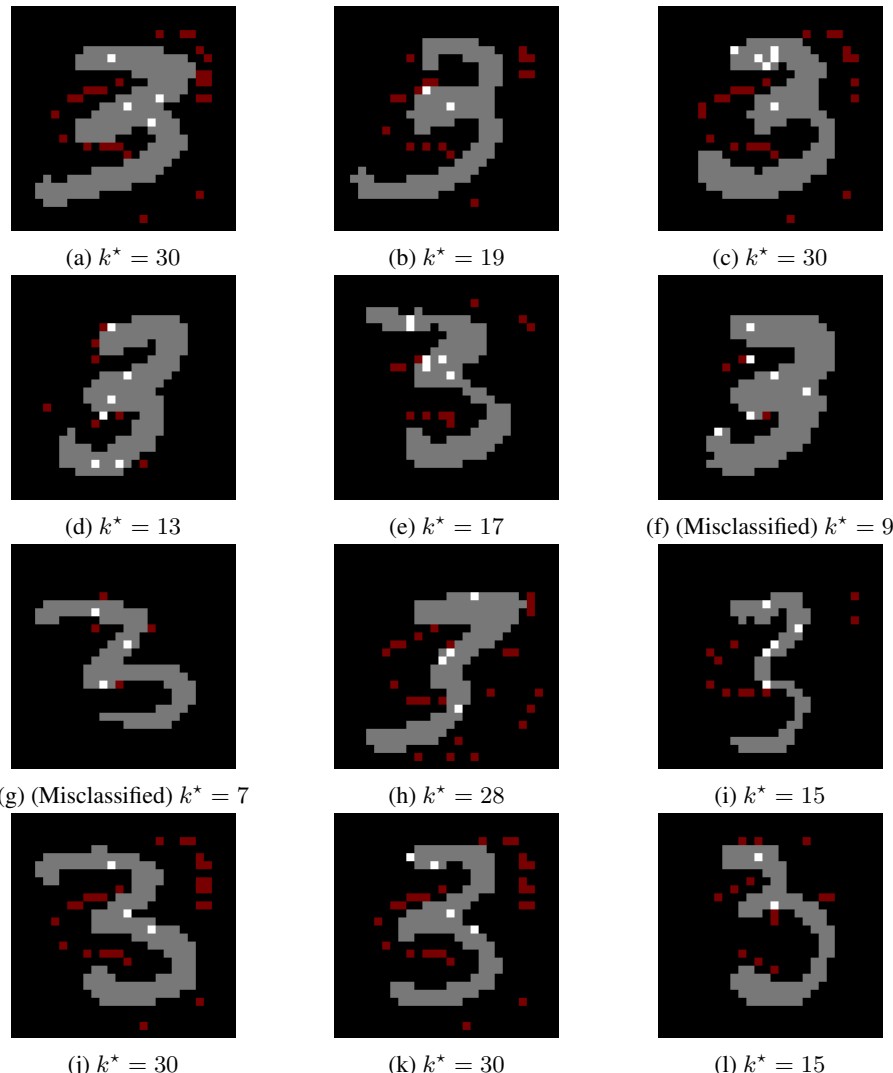

(a) $k^\star = 30$     (b) $k^\star = 19$     (c) $k^\star = 30$

(d) $k^\star = 13$     (e) $k^\star = 17$     (f) (Misclassified) $k^\star = 9$

(g) (Misclassified) $k^\star = 7$     (h) $k^\star = 28$     (i) $k^\star = 15$

(j) $k^\star = 30$     (k) $k^\star = 30$     (l) $k^\star = 15$

Figure 13: Examples of *Minimum Sufficient Reasons* over the MNIST dataset. All images correspond to positive instances for a decision tree of $1486$ leaves that detects the digit $3$. Two instances are misclassified. Light pixels of the original image are depicted in grey, and the light pixels of the original image that are part of the minimum sufficient reason are colored white. Dark pixels that are part of the minimum sufficient reason are colored with red. Individual captions denote the size of the minimum sufficient reasons with $k^\star$.

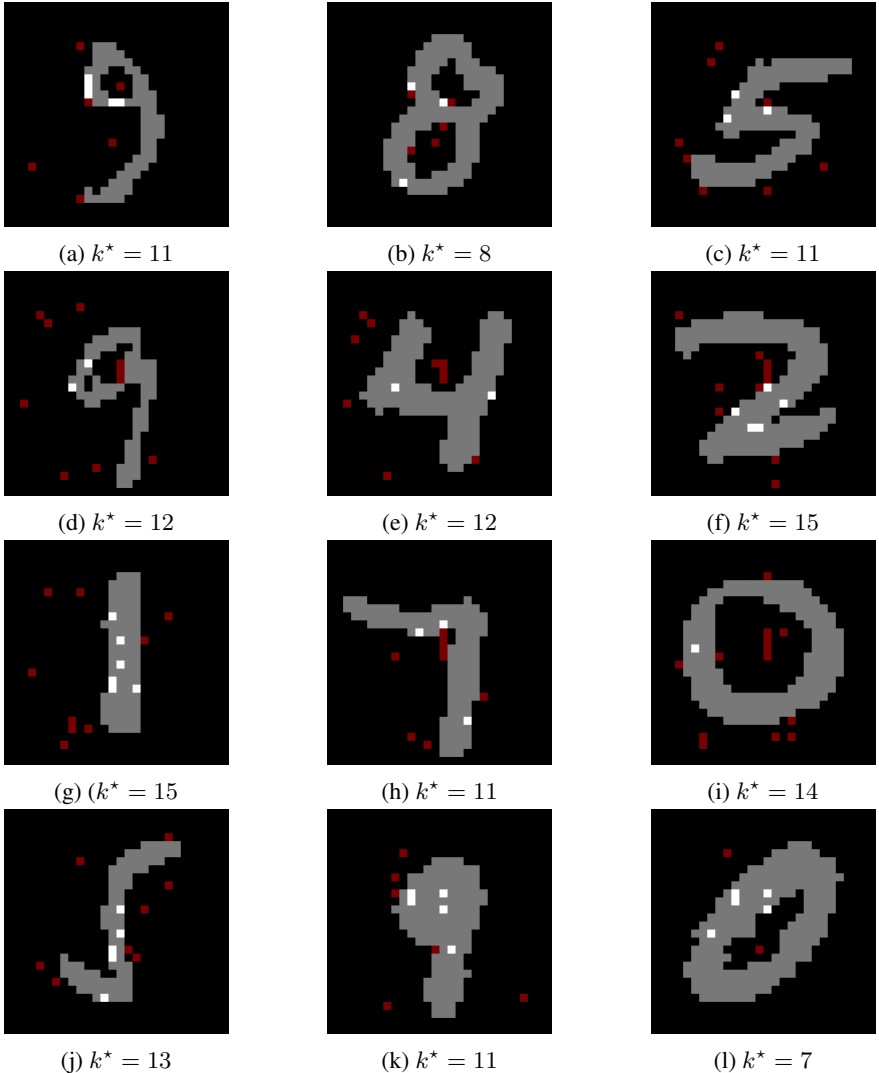

(a) $k^\star = 11$     (b) $k^\star = 8$     (c) $k^\star = 11$

(d) $k^\star = 12$     (e) $k^\star = 12$     (f) $k^\star = 15$

(g) ($k^\star = 15$     (h) $k^\star = 11$     (i) $k^\star = 14$

(j) $k^\star = 13$     (k) $k^\star = 11$     (l) $k^\star = 7$

Figure 14: Examples of *Minimum Sufficient Reasons* over the MNIST dataset. All images correspond to negative instances for a decision tree of $1486$ leaves that detects the digit $3$ and are classified correctly. Light pixels of the original image are depicted in grey, and the light pixels of the original image that are part of the minimum sufficient reason are colored white. Dark pixels that are part of the minimum sufficient reason are colored with red. Individual captions denote the size of the minimum sufficient reasons with $k^\star$.

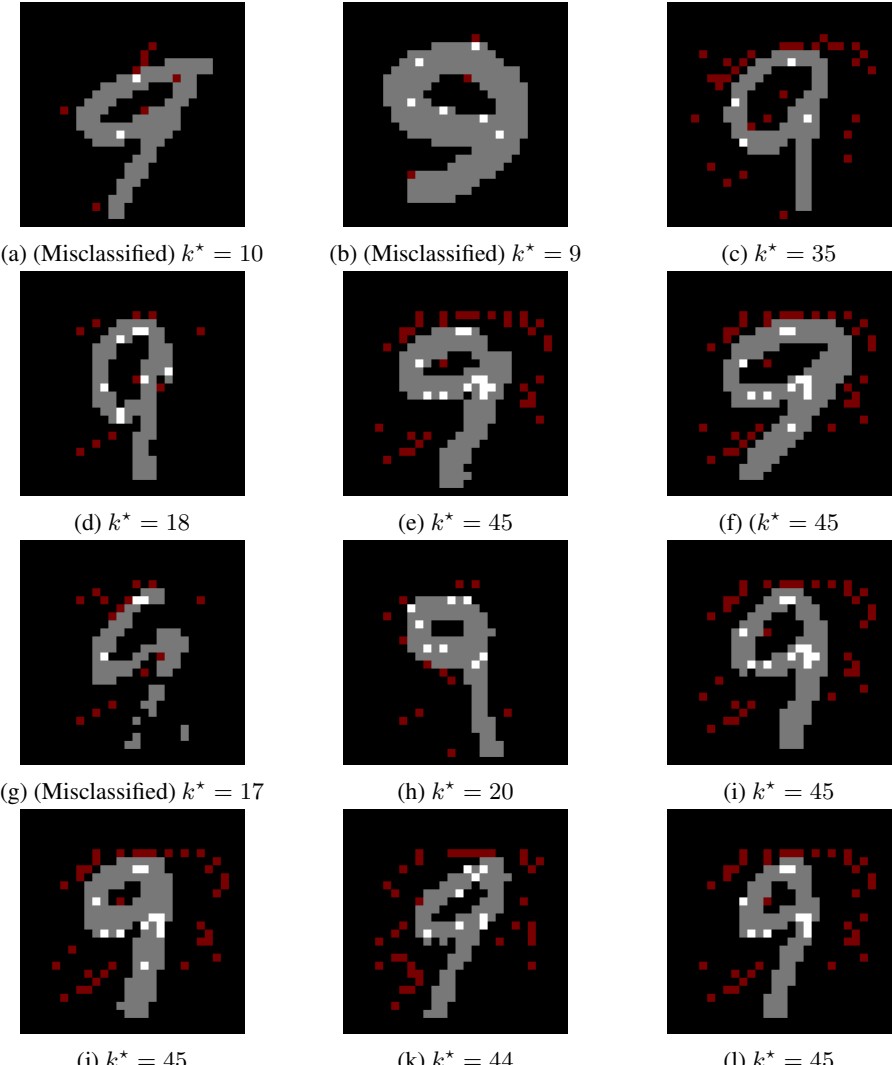

(a) (Misclassified) $k^\star = 10$     (b) (Misclassified) $k^\star = 9$     (c) $k^\star = 35$

(d) $k^\star = 18$     (e) $k^\star = 45$     (f) ($k^\star = 45$

(g) (Misclassified) $k^\star = 17$     (h) $k^\star = 20$     (i) $k^\star = 45$

(j) $k^\star = 45$     (k) $k^\star = 44$     (l) $k^\star = 45$

Figure 15: Examples of *Minimum Sufficient Reasons* over the MNIST dataset. All images correspond to positive instances for a decision tree of $1652$ leaves that detects the digit $9$. Three instances are misclassified. Light pixels of the original image are depicted in grey, and the light pixels of the original image that are part of the minimum sufficient reason are colored white. Dark pixels that are part of the minimum sufficient reason are colored with red. Individual captions denote the size of the minimum sufficient reasons with $k^\star$.

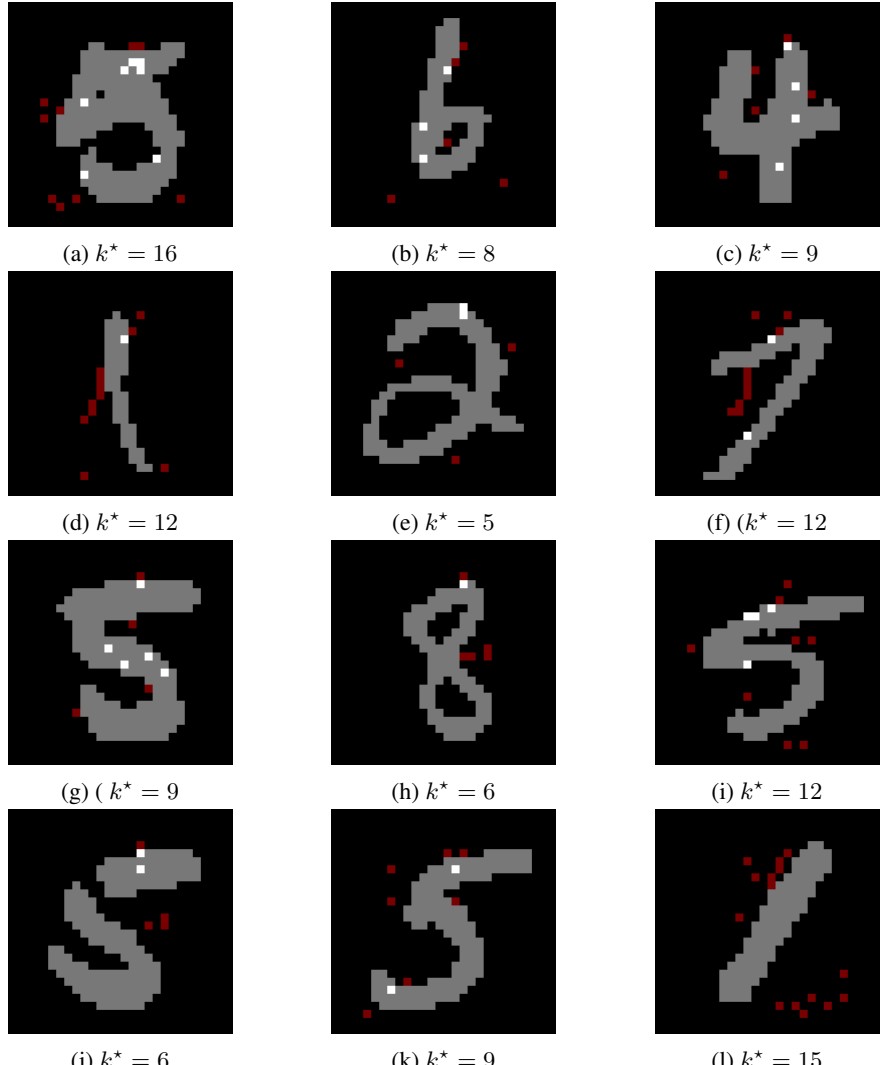

(a) $k^\star = 16$  (b) $k^\star = 8$  (c) $k^\star = 9$

(d) $k^\star = 12$  (e) $k^\star = 5$  (f) ($k^\star = 12$

(g) ( $k^\star = 9$  (h) $k^\star = 6$  (i) $k^\star = 12$

(j) $k^\star = 6$  (k) $k^\star = 9$  (l) $k^\star = 15$

Figure 16: Examples of *Minimum Sufficient Reasons* over the MNIST dataset. All images correspond to (correctly classified) negative instances for a decision tree of $1652$ leaves that detects the digit $9$. Light pixels of the original image are depicted in grey, and the light pixels of the original image that are part of the minimum sufficient reason are colored white. Dark pixels that are part of the minimum sufficient reason are colored with red. Individual captions denote the size of the minimum sufficient reasons with $k^\star$.

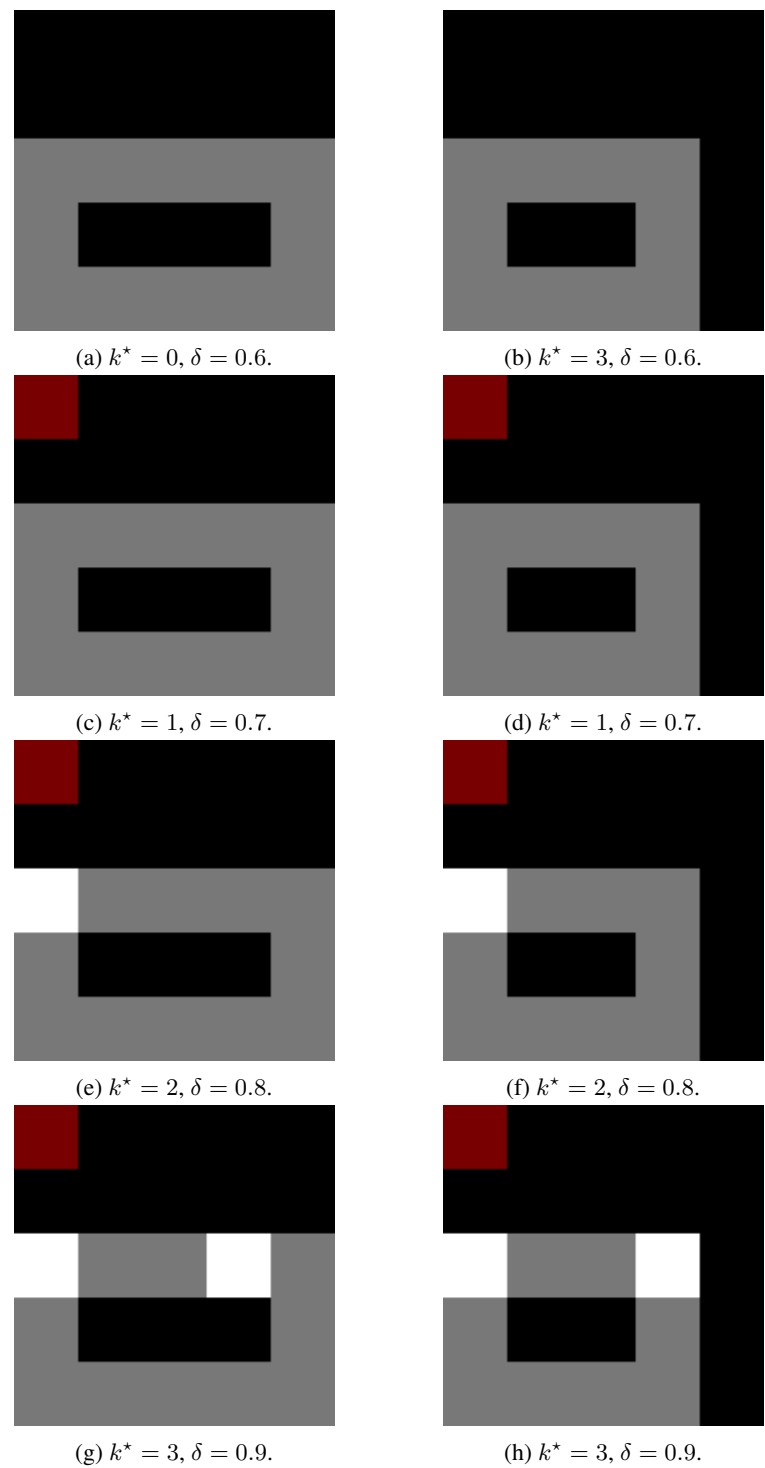

(a) $k^\star = 0, \delta = 0.6$.

(b) $k^\star = 3, \delta = 0.6$.

(c) $k^\star = 1, \delta = 0.7$.

(d) $k^\star = 1, \delta = 0.7$.

(e) $k^\star = 2, \delta = 0.8$.

(f) $k^\star = 2, \delta = 0.8$.

(g) $k^\star = 3, \delta = 0.9$.

(h) $k^\star = 3, \delta = 0.9$.

Figure 17: Examples of $\delta$-SRs over the synthetic rectangle dataset, where instances have been correctly classified as wide rectangles. In fact, the explanations depicted in (g) and (h) have probability 1, and thus no further changes are obtained when raising $\delta$.