# OpenReview forum: "On Computing Probabilistic Explanations for Decision Trees"
_NeurIPS.cc/2022/Conference — NeurIPS 2022 Accept_

### Official Review · Reviewer_megZ · 2022-07-01

**Rating:** 8
**Confidence:** 4
**Soundness:** 4 excellent
**Presentation:** 3 good
**Contribution:** 3 good

**Summary:**

A sufficient reason (SR) for a decision taken by a classifier on input x
is a subset e of the assignment x (when viewed as a set of literals)
which is sufficient to account for the decision. For obvious reasons
the user may be interested in minimum-cardinality or (more modestly)
subset-minimal SR's. SR's are a basic notion in the formal approach to
explaining decisions of classifiers, however it is a very strict notion,
leading to SR's which contain too many features for the user to grasp.
Thus, from a practical point of view, a more useful notion is a delta-SR
e which allows a certain degree of error, in that only a fraction delta<1 of
the completions of the partial assignment e need to be classified the same as x.

This paper settles the computational complexity of certain problems concerned with
finding delta-SR's over decision trees, showing notably that even finding a subset-minimal
delta-SR is intractable (assuming P not equal to NP).

They show that this intractability can be avoided if extra restrictions are applied
(such as a structural property of the decision tree or monotonicity of the classifier).
They also give a SAT encoding for finding smallest delta-SR's.

**Questions:**

(1) Can the notion of split number be generalised to DAGs (such as d-DNNFs)?

(2) Is bounded split number not the same as bounded treewidth? It is well known that
bounded treewidth allows polynomial-time counting for many combinatorial problems
(see, for example, Michael Jakl, Reinhard Pichler, Stefan Rümmele & Stefan Woltran,
Fast Counting with Bounded Treewidth, LPAR 2008, pp 436–450 or articles cited therein)

(3) Have you tested the loss in accuracy if you restrict yourself to learning
DT's with bounded split number?

**Strengths And Weaknesses:**

The main complexity results are negative but nonetheless very interesting since they
show intractability even for decision trees, which is one of the simplest possible
ML models. It is not too difficult to show that the simple greedy-deletion algorithm
which can be used to find SR's of decision trees in polynomial time no longer
works for delta-SR's due to the non-hereditary nature of delta-SR's (e.g. we can
have A \subset B \subset C with A,C both delta-SR's but not B). But, NP-hardness
is not at all obvious. The constructions required to prove this in the supplementary
material are far from straightforward.

There are also some positive results, concerning DT's with bounded split number or
monotone free BDD's (Corollary 1). The authors also give a SAT encoding which provides a
way of calculating delta-SR's, but as is shown in the supplementary material this is
only practical for decision trees of very modest size (20 leaves, 36 features).
This simply confirms, in practice, the theoretical intractability proved in the paper.

Minor points:
- perhaps reword 1st sentence of abstract since formal XAI is not just
concerned with explaining decisions of ML models.
- line 286: a citation is required so the reader can easily find out what free BDDs are
and find a proof that counting is polytime.
- line 308: in the output, don't you mean n-|..| \leq k instead of |..| \leq k?
- line 325: "its labeled" should be "it is labeled"
- line 342: "can be take safely" should be "can be taken safely"
- line 346: missing "be" before "a series"
- line 365: missing "and" before "clauses"?
- I am confused by Question 2 in Future Work which seems to have been answered by Corollary 1.
- line 376: delete "a" in "every fixed such a"
- use {} in latex to impose capitals in bibliography entries (e.g. {XAI})
- the arXiv paper by Izza et al occurs twice in the bibliography
- line 636: Do you mean \neg y_i instead of y_i ? Below you seem to be assuming that
these variables are negated.
- line 682: Let us say
- line 687: do you mean 4e instead of M?
- line 719: delete "that"

---

> ### Author Response · Authors · 2022-08-02
> **Response to megZ**
>
> We appreciate that the reviewer shares our excitement about these new theoretical results! We also thank the reviewer for the detailed minor comments; we will fix all typos in the updated version.
>
> > "Can the notion of split number be generalised to DAGs (such as d-DNNFs)?"
>
> This is an interesting question. It might be possible that the notion of split number and the tractability of bounded split number generalize to DAGs, but this would require a careful analysis of our dynamic programming approach. This is definitely a nice question for future work (actually it is aligned with question 5 on Section “Future Work”).
>
> > "Is bounded split number not the same as bounded treewidth? It is well known that bounded treewidth allows polynomial-time counting for many combinatorial problems (see, for example, Michael Jakl, Reinhard Pichler, Stefan Rümmele & Stefan Woltran, Fast Counting with Bounded Treewidth, LPAR 2008, pp 436–450 or articles cited therein)"
>
> We are not sure here about what a reasonable definition of treewidth for a decision tree would be. One possibility is to look at the standard definition of treewidth for a CNF and then look at the treewidth of the CNF translation of a decision tree. In general, the size of the clauses in this transition will be unbounded and so the treewidth. A second possibility is to look at the directed graph obtained from the decision tree after identifying nodes with the same feature. Under this definition bounded split number differs from bounded treewidth, as we can take decision trees with one root r and two left and right disjoint paths with features $x_1,..,x_n$ (we add leaves with any value to complete the tree). The split number is $n$ (unbounded) but the treewidth is 1 as we have a directed path as the underlying directed graph. It is a nice question whether there is a connection between treewidth and split number (for instance, whether bounded split number implies bounded treewidth), or whether (some notion of) bounded treewidth yields some tractability results.  If the question refers simply to the underlying graph of the decision tree, then its treewidth will always be 1 (as in any tree), but this might be an interesting parameter when considering more general DAGs as suggested in the previous question.
>
> > "Have you tested the loss in accuracy if you restrict yourself to learning DT's with bounded split number?"
>
> We have not tested this. We came to the notion of bounded split-number from a theoretical perspective: by thinking about what parameter would allow us to divide the problem into roughly independent sub-problems. We did test at some point how the split number of standard decision trees scaled with respect to their size, and even though the split number was much smaller, it still scaled roughly linearly with size. It is indeed a good idea to try to design a learning algorithm that keeps this number bounded and see how that impacts accuracy. We will include this as an open question in a camera-ready version.

---

### Official Review · Reviewer_BptY · 2022-07-09

**Rating:** 6
**Confidence:** 3
**Soundness:** 3 good
**Presentation:** 3 good
**Contribution:** 1 poor

**Summary:**

After rebuttal: The authors clarified my misunderstanding and I acknowledge that the problem seems to be practically harder than I thought. Some reservations about the relevance of decision trees for NeurIPS, the choice of the dataset and the speed of the SAT encodings remain, but I will change my overall evaluation to weak accept and lower my confidence.

---

The paper studies computing delta-sufficient reasons for decision trees. It shows that finding subset and cardinality minimal delta-sufficient reasons is not possible in polynomial time assuming P!=NP. A SAT-encoding is presented and the appendix shows some runtime results for trees with up to 200 leaves.

**Questions:**

Please correct me if I am wrong about the difficulty of the problem. Are the decision trees available for inspection in the code repository? I would be interested in looking at the decision trees of depth 20 in order to understand why the problem could be more complicated. It would also be interesting to know what the theoretical upper bound for the number of leaves compared to their actual number is. What is actually the background of the generated trees? Have they been generated randomly or do they achieve better performance with increasing number of leaves? Why has MNIST actually been chosen for the experiments? It does not really seem a natural choice for decision trees. Can we actually learn anything from the decision tree/ the explanations for MNIST? Why not consider a tabular dataset?

**Limitations:**

Yes.

**Strengths And Weaknesses:**

While this is a solid technical paper, I do not think that it is a good contribution for NeurIPS. It seems to me that the contribution is mainly interesting from a theoretical perspective. Please correct me if am wrong, but it seems to me that the complexity arises only from considering all possible extensions of the minimal sufficient reasons. However, considering all these candidates seems unnecessary in a decision tree because decision trees are typically not exponentially large. Indeed, all trees in the experiments have at most 200 leaves. This means that there are at most 200 decision paths that have to be analyzed. The SAT encodings need more than 4 seconds even if the trees have only 20 leaves, which seems rather slow. The runtime remains the same as the number of leaves increases, but this is probably because the encoding (exponentially) depends on the number of features (that presumably did not change in the experiments) and not on the number of decision paths (which is exponentially bounded by the number of features, but typically significantly smaller than the worst-case bound). It seems that the problem could be solved more efficiently by considering the decision nodes and their possible extensions in the tree. From an explanation perspective, it would also be nice to see some examples of explanations generated with this approach. Overall, I think that the current state of the paper could be a nice contribution for a smaller conference on theoretical aspects. However, for a major conference like NeurIPS, the significance is not sufficiently clear.

---

> ### Author Response · Authors · 2022-08-02
> **Response to BptY**
>
> > "While this is a solid technical paper, I do not think that it is a good contribution for NeurIPS. It seems to me that the contribution is mainly interesting from a theoretical perspective."
>
> While we agree with our paper being fundamentally theoretical, we believe that NeurIPS is a place that gathers both theoreticians and practitioners, and we are particularly interested in that exchange. For example, the paper “Provably efficient, succinct, and precise explanations”, published at NeurIPS 2021, is precisely about studying a model of probabilistic sufficient reasons that circumvents the lower bound presented in our paper by relaxing the minimality condition.
>
> > "Please correct me if am wrong [...] This means that there are at most 200 decision paths that have to be analyzed."
>
> This is indeed not correct; the main source of complexity is that sufficient reasons are about finding *a subset* of the input that is sufficient to explain its classification, and there are exponentially many such subsets. More precisely, our results imply that no algorithm can compute minimal delta-SR in time polynomial *in the number of leaves of the tree* unless P=NP. We acknowledge that our proofs require building very sophisticated decision trees and thus this fact gets obscured, but for a simpler example, the proof of NP-hardness for delta=1, that we cite from “P. Barceló, M. Monet, J. Pérez, and B. Subercaseaux. Model Interpretability through the lens of Computational Complexity. In NeurIPS, pages 15487–15498, 2020.” shows that computing a minimum sufficient reason in a decision tree with N leaves can be equivalent to computing a minimum dominating set for a graph with N nodes. Even for N=200 this is a highly non-trivial task considering no sub-exponential algorithm is known.
>
> > "The SAT encodings need more than 4 [...]"
>
> We agree with these statements. We have made things significantly more efficient both for deterministic and probabilistic explanations and uploaded a revised version. Nonetheless, even after our optimization, obtaining probabilistic explanations is still slow even for modest trees. We see this as a confirmation of the difficulty of the problem. Note that the size of the encoding depends both on the number of decision paths and the number of features, although you are right in that usually the number of paths is way smaller than the theoretical worst-case bound.
>
> > "From an explanation perspective, it would also be nice to see some examples of explanations generated with this approach."
>
> We have included examples and their discussion in the updated version (for the delta=1 case).
>
> > "Are the decision trees available for inspection in the code repository?"
>
> Yes, for example mnist.dt describes the tree trained for mnist. In the updated version we have uploaded this file directly for ease of check; it is written in a JSON format with a recursive description.
>
> > "I would be interested in looking at the decision trees of depth 20 in order to understand why the problem could be more complicated."
>
> We have also included one of these directly in the revised version for easy verification, under the name rectangles.dt
>
> > "It would also be interesting to know what the theoretical upper bound for the number of leaves compared to their actual number is."
>
> We are not sure to fully understand this question. We trained decision trees with Scikit learn, where one specifies the maximum number of leaves before training. Sometimes this limit is not achieved, when the training algorithm does not require any more splits. We have reported the actual number of leaves of the trees obtained, instead of the upper bound specified in Scikit learn.
>
> > "What is actually the background of the generated trees? Have they been generated randomly or do they achieve better performance with increasing number of leaves?"
>
> They have been trained with Scikit learn; we have now included more information about this in the updated version, including accuracy data.
>
> > "Why has MNIST actually been chosen for the experiments? [...] "
>
> These are all very good questions. MNIST has the advantage of having ~800 features, which serves to show that the delta=1 encoding is able to handle large feature spaces. We agree with you nonetheless in decision trees being an unnatural choice for this dataset (or more in general, vision tasks) in a general context. Another advantage of this dataset is that it has allowed us to present explanations visually (see revised version). Tabular datasets are definitely more natural, however, they usually include non-binary features, and thus their binarization becomes less trivial, so it was mostly a matter of convenience. For a camera-ready version of the paper, we will include probabilistic explanations on tabular datasets.
>
> On a general note, we are glad the reviewer values the theoretical aspects of the paper. We believe we have answered the reviewer's concerns and we kindly ask them to reevaluate their score.

---

> > ### Comment · Reviewer_BptY · 2022-08-08
> > **Acknowledgement**
> >
> > Thank you for the additional explanations. I will take them into account in the final discussion and will update my review accordingly.

---

### Official Review · Reviewer_A6C9 · 2022-07-09

**Rating:** 5
**Confidence:** 4
**Soundness:** 3 good
**Presentation:** 1 poor
**Contribution:** 2 fair

**Summary:**

In the setting of explainable artificial intelligence, this paper investigates probabilistic explanations for decision tree predictions. Namely, given a decision tree $T$, a data instance $e$ and a confidence parameter $\delta \in (0,1]$, a $\delta$-sufficient reason for $e$ given $T$ is a monomial that covers $e$ and for which the proportion of satisfying assignments $e'$ such that $T(e') = T(e)$ is at least $\delta$. A minimal $\delta$-sufficient reason is a  $\delta$-sufficient reason that is subset minimal, and a minimum $\delta$-sufficient reason is $\delta$-sufficient reason of minimum size. Based on these definitions, the authors prove that the problems of finding minimal/minimum $\delta$-sufficient reasons are NP-hard. They also consider tractable cases by considering restrictions on the structure (bounded split number) and the semantics (monotonicity) of decision trees. Finally, they propose a propositional SAT encoding for computing minimum $\delta$-sufficient reasons.


**Questions:**

Yes, I have a few questions about tractable cases:
* A natural class of shallow (and interpretable) models is the family of decision trees of bounded depth, say $d$. Are the problems of finding minimal/minimum $\delta$-sufficient reasons still intractable for fixed $d$? Are they fixed-parameter tractable with respect to $d$?
* Theorem 5 is interesting but leaves open a natural case: the well-studied class of monotone DNF. As we know, counting the number of satisfying assignments of a DNF admits an FPRAS. So, do we still have an FPRAS result for computing minimal/minimum $\delta$-sufficient reasons when the predictor is representable by a monotone DNF formula?

**Limitations:**

Overall, I think that the key results of this paper are of theoretical nature, and hence, this paper should stick to the theoretical side. Indeed, as mentioned above, the experimental study is still at an early stage. So, focusing on theoretical results (Section 4 and 5) should leave some room for presenting the proof of the main result (Theorem 2) in more detail, and for examining more tractable cases - or showing that natural subclasses remain intractable.


**Strengths And Weaknesses:**

Overall, I would say that the paper includes some good ideas, but it requires more polishing. For the moment, the paper is not self-contained and should be re-organized.

Let us start with the positive aspects.  Based on the recent work by Izza et al. (2022), the problem of finding a $\delta$-sufficient reason of size at most $k$ belongs to NP, and the optimization version of this problem belongs to $\Theta_2^P$. So, a key question is to determine whether such a problem is NP-hard, or not. This is arguably the main result of this paper (Section 4): the authors have proven that the problems of computing minimal/minimum $\delta$-sufficient reasons are NP-hard. The work spent on finding tractable subclasses (Section 5) is interesting but would deserve further investigation (see comments in the section "Limitations").

Unfortunately, the paper is not self-contained. In particular, the main result of this study (Theorem 3) includes a sketch of proof that could be improved. To this point, I was merely confused by the sentence “Finally, the reduction from k-Minimal-Expected-Clauses to Check-Sub-SR builds an instance $(T, e)$ from $(\phi, \sigma)$ in a way that there is a direct correspondence between partial assignments $\mu \subseteq \sigma$ and partial instances $e’ \subseteq e$, satisfying that [...]”. As we know, unless P = \#P, we cannot find a polynomial translation from a $k$-CNF formula $\phi$ to an equivalent decision tree $T$ (even for $k = 2$). In fact, the construction of $T$ is more sophisticated (notably, it is not equivalent to $\phi$), and the overall proof is clarified in the Supplementary Material.

My last major comment is about the SAT encoding of the (decision version of the) minimum $\delta$-sufficient reason problem (Section 6.2). While this encoding looks elegant, the experiments reported in the Supplementary Material are at a preliminary stage. Read again my comment below, but if the authors really want to provide a propositional SAT encoding, it should be tested on various datasets, and it should be compared with the SMT encoding proposed by Izza et al.

---

> ### Author Response · Authors · 2022-08-02
> **Response to AC69**
>
> > “Unfortunately, the paper is not self-contained. In particular, the main result of this study (Theorem 3) includes a sketch of proof that could be improved. To this point, I was merely confused by the sentence “Finally, the reduction from k-Minimal-Expected-Clauses to Check-Sub-SR builds an instance  from  in a way that there is a direct correspondence between partial assignments  and partial instances , satisfying that [...]”. As we know, unless P = #P, we cannot find a polynomial translation from a -CNF formula  to an equivalent decision tree  (even for ). In fact, the construction is more sophisticated (notably, it is not equivalent to ), and the overall proof is clarified in the Supplementary Material."
>
> We respectfully disagree about the paper not being self-contained. Due to the length of our proofs, we were forced to defer them to the supplementary material, but this is a common practice at NeurIPS. On the other hand, we agree with you in that claiming a translation from a 2-CNF formula to a decision tree would be wrong (modulo complexity assumptions); that sentence is meant to say that the correspondence between assignments is such that the aforementioned equation holds. Note that computing the expected number of satisfied clauses of a CNF formula can be trivially done by linearity of expectation, and thus this does not break any complexity assumptions. We appreciate the suggestion and will reformulate that sentence to avoid confusion.
>
> > “My last major comment is about the SAT encoding of the (decision version of the) minimum -sufficient reason problem (Section 6.2). While this encoding looks elegant, the experiments reported in the Supplementary Material are at a preliminary stage. Read again my comment below, but if the authors really want to provide a propositional SAT encoding, it should be tested on various datasets, and it should be compared with the SMT encoding proposed by Izza et al.”
>
> We completely agree with the presented experiments being only at a preliminary stage. We have included more experimentation now. Please check the revised supplementary material. Nonetheless, we agree with the reviewer that the main strength of the paper lies in its theoretical results.
>
> Please note that the SMT encoding proposed by Izza et al. under the title “Provably Precise, Succinct and Efficient Explanations for Decision Trees” appeared on arXiv on May 22nd, 2022, while the deadline for the paper submission this year was May 19th. We agree that this paper is completely relevant to our work, and we cite it in our revised version. Comparing our approaches is a definite part of our future work.
>
> > “A natural class of shallow (and interpretable) models is the family of decision trees of bounded depth, say d. Are the problems of finding minimal/minimum -sufficient reasons still intractable for fixed d? Are they fixed-parameter tractable with respect to d?”
>
> While we believe the abundance of shallow decision trees is good motivation for this question, the proposed formalization, unfortunately, becomes trivial as we describe next.  Note first that the input size is dominated by the size of the input decision trees, as if there are features in the instance that do not appear in any node of the tree, then it’s easy to observe that those cannot be part of any (delta)-minimal sufficient reason. Now, the size of a decision tree of depth at most d is at most $2^d$. If d is a constant, then this is also a constant and thus the problem can be solved in constant time. When parameterizing by d, as the input has size at most $2^d$, a naive exponential algorithm has complexity $2^{2^d}$ and is thus fixed-parameter tractable (FPT).
>
> > “Theorem 5 is interesting but leaves open a natural case: the well-studied class of monotone DNF. As we know, counting the number of satisfying assignments of a DNF admits an FPRAS. So, do we still have an FPRAS result for computing minimal/minimum -sufficient reasons when the predictor is representable by a monotone DNF formula?”
>
> We totally agree. This is an interesting and natural question for future work. We do not think this is trivial at all and will add a remark about this question in a camera-ready version. Thanks for the suggestion!
>
> On a general note, we are glad the reviewer values the theoretical aspects of the paper. We believe we have answered the reviewer's concerns and we kindly ask them to reevaluate their score.

---

> > ### Comment · Reviewer_A6C9 · 2022-08-09
> > **Re: Response to AC69**
> >
> > Thanks for your answers. For the moment, I have a mixed opinion. On the one hand, the proof of the main result is quite elegant, and I really would like to see it published. On the other hand, I believe the paper still requires some polishing for a conference such as NeurIPS.
> >
> > That being said, the effort spent by the authors in providing additional experimental results is appreciated. Yet, unless I missed something, it seems that for the MNIST dataset, all experiments have been performed using $\delta = 1$, right? If so, I would suggest making additional experiments using several choices for $\delta$ (and, like in Table 1, indicating the size of the obtained reason together with the time spent in computing that reason).

---

> > > ### Author Response · Authors · 2022-08-09
> > > **Response to AC69**
> > >
> > > Thanks for your response.
> > >
> > > Indeed we do not obtain probabilistic explanations for MNIST, as its ~800 features are too much for the probabilistic encoding. Hopefully this is clearer now in the supplementary material.
> > >
> > > Moreover, we have added some images of explanations for the dataset of rectangles. We plan to add more of these to better illustrate $\delta$-SRs.
> > >
> > > We appreciate your opinion on our theoretical results being elegant and valuable, and we would like to improve the polishing of the paper; note that in the final version of the paper we will address some typos and minor comments pointed out by other reviewers, as well as polish the sketch of proof discussed above. What other aspects of the paper could we polish? We are willing to make further changes to improve the presentation of our results.

---

> > > > ### Comment · Reviewer_A6C9 · 2022-08-09
> > > > **Re: Response to AC69**
> > > >
> > > > Okay, if your encoding for probabilistic explanations cannot handle MNIST, I would suggest considering some other real-world datasets which are commonly used in XAI (ex: "Compas", "Placement", etc.). By the way, "Gisette" includes a large number of features, but decision trees trained with CART are typically quite small (around 350 nodes). So, it might be a good alternative to MNIST for inferring probabilistic explanations.

---

> > > > > ### Author Response · Authors · 2022-08-09
> > > > > **Response to A6C9**
> > > > >
> > > > > Thanks for the suggestions!
> > > > >
> > > > > Given what you mention, it might be possible to improve our encoding for instances that have more features than nodes. Perhaps this could allow for probabilistic explanations for the Gisette dataset. We appreciate this idea.
> > > > >
> > > > > Indeed testing our encoding in more real world datasets, and optimize its performance, are natural steps for future work in a more practical paper.

---

### Meta-Review · Area_Chair_Z1ge · 2022-08-24

**Recommendation:** Accept
**Confidence:** Certain

**Metareview:**

The paper studies the complexity of discovering probabilistic explanations for decision trees. They show that computing minimum/minimal probabilistic explanations is an NP-hard problem. On the other hand, they also identify structural conditions that make the problem efficiently solvable.

The reviewers appreciated the challenges in proving the complexity results. Although the experimental results are not so convincing, we evaluated the paper mainly on its theoretical aspects and find it suitable for acceptance.

**Award:**

No

---

### Decision · Program_Chairs · 2022-09-14

Accept